# From Gradient Volume to Shapley Fairness: Towards Fair Multi-Task Learning

**Xiao Wang**[1], **Yuying Han**[1], **Dazi Li**[1], **Fei Zhang**[2], **Min Tang**[2] [*]

[1]Beijing University of Chemical Technology
[2]National Innovation Institute of Defense Technology
{w_xiao, han_yuying, lidz}@buct.edu.cn
feizhang100@sina.com, mtangcn@163.com

## Abstract

Multi-task learning often suffers from gradient conflicts, leading to task-level unfair optimization and degraded overall performance. To address this, we present SVFair, a Shapley value-based framework for fair gradient aggregation that explicitly targets task-level fairness under such conflicts. Unlike heuristic scalarization or pairwise conflict penalties, SVFair combines a geometric view of gradient interaction with a cooperative-game view of fair contribution. We propose two scalable geometric conflict metrics: VolDet, a gram determinant volume metric, and VolDetPro, its sign-aware extension distinguishing antagonistic gradients. By integrating these metrics into Shapley value computation, SVFair quantifies each task's deviation from the overall gradient and rebalances updates toward fairness. In parallel, our Shapley value computation admits controllable complexity. Extensive experiments show that SVFair achieves state-of-the-art results across diverse supervised and reinforcement learning benchmarks, and further improves existing methods when integrated as a fairness-enhancing module.

## 1 Introduction

Multi-task learning (MTL) (Caruana, 1997; Zhang & Yang, 2021; 2018) enhances learning efficiency and generalization by jointly training a model on multiple tasks. By leveraging shared representations, MTL can lead to better performance compared to training separate models for each task. Consequently, MTL has been widely applied in various fields like natural language processing (Chen et al., 2024; Radford et al., 2019), computer vision (Misra et al., 2016; Liu et al., 2019; He et al., 2020; Vandenhende et al., 2020), reinforcement learning (Oh et al., 2017; Yu et al., 2020b) and natural sciences (Tang et al., 2023; Yang, 2024; Eyuboglu et al., 2021). However, sharing a single model inevitably couples the optimization dynamics of all tasks: gradients from different objectives can conflict with each other, so that some tasks consistently dominate the update direction while others are repeatedly compromised. This phenomenon, often referred to as task-level optimization unfairness (Kendall et al., 2018; Lin et al., 2024; Sener & Koltun, 2018), not only degrades the worst-task performance but can also undermine the reliability of MTL in safety- or mission-critical applications (Ban & Ji, 2024; Qin et al., 2025b). To address this phenomenon, existing methods have adopted loss-based (Kendall et al., 2018; Liu et al., 2019; 2023; Shen et al., 2024; Xiao et al., 2025) and gradient-based (Sener & Koltun, 2018; Liu et al., 2021a; Navon et al., 2022; Ban & Ji, 2024; Qin et al., 2025b; Zhang et al., 2025) methods.

Although some gradient-based methods (Sener & Koltun, 2018; Navon et al., 2022) attempt to incorporate fairness, most still focus on final performance and fail to explicitly quantify how individual task gradients conflict with the overall gradient (Ban & Ji, 2024; Qin et al., 2025b). This leads to unfair allocation of optimization resources and limits overall performance improvement. While some studies use cosine similarity (Yu et al., 2020a; Liu et al., 2021b) or inner products (Liu et al., 2021a) to measure task conflicts, these metrics are restricted to pairwise analysis and fail to capture the global structure of gradient conflicts across the entire task set. In particular, they do not provide

---

[*]Corresponding author.

a coalition-level view of how a subset of tasks collectively interacts with the others, making it difficult to reason about which tasks contribute to, or suffer from, gradient interference at the level where fairness should be defined. This limitation becomes particularly severe in large-scale scenarios, such as CelebA (Liu et al., 2015) with 40 tasks, in which scalability and efficiency are critical challenges.

These observations motivate us to seek a framework that jointly (i) models gradient conflict from a geometric perspective at the level of arbitrary task coalitions, and (ii) links this coalition utility to a principled fair-division rule for distributing gradient updates. To systematically address this limitation, we propose two scalable conflict metrics centered on task subset gradient differences. Building upon this metric, we construct a fairness metric centered on the task set as a whole, effectively quantifying gradient deviation, thereby enhancing optimization balance across tasks and improving overall performance. Our main contributions are as follows:

- **We introduce Shapley value-based fairness measurement into MTL.** We are the first to apply the Shapley value (Shapley, 1953; Wang et al., 2022a; Chen et al., 2023) to the MTL optimization framework, enabling precise quantification of individual task gradient deviations from the overall gradient, thereby alleviating inter-task conflicts and enhancing fairness.

- **We propose VolDet and VolDetPro, efficient geometric gradient conflict metrics.** VolDet measures conflict as the volume of the parallelotope spanned by task gradients, computed from the determinant of their Gram matrix. VolDetPro preserves this intuition but is sign-aware, adding a lightweight penalty only when pairwise similarities are negative to fix the volume-only blind spot. Both metrics remain efficient and scalable, enabling single-pass Shapley value computation, reducing cost, and supporting large-scale MTL.

- **We establish the SVFair gradient aggregation framework with Shapley values.** SVFair leverages the Shapley values computed in a single pass to guide gradient aggregation. Both theoretical analysis and empirical results demonstrate convergence to Pareto stationary points, accompanied by significantly improved overall performance.

- **Extensive experiments validate the effectiveness and compatibility of SVFair.** SVFair consistently surpasses state-of-the-art methods across diverse supervised and reinforcement learning benchmarks. Moreover, the Shapley value can be seamlessly integrated into existing MTL algorithms, further enhancing overall performance.

## 2 RELATED WORK

**Multi-task learning.** Existing MTL optimization methods can be broadly categorized into loss-based (Liu et al., 2019; Dai et al., 2023) and gradient-based (Sener & Koltun, 2018; Chen et al., 2018; Yu et al., 2020a; Liu et al., 2021a) methods. Loss-based (Liu et al., 2023; Shen et al., 2024; Xiao et al., 2025) approaches adjust task weights at the loss level through heuristic strategies or uncertainty modeling. Although computationally efficient, they often suffer from performance bottlenecks in complex task scenarios. In contrast, gradient-based (Chen et al., 2020; Navon et al., 2022; Ban & Ji, 2024; Qin et al., 2025b; Zhou et al., 2025; Qin et al., 2025a) methods leverage task gradient information, employing techniques such as Pareto optimization (Sener & Koltun, 2018; Ban & Ji, 2024; Zhang et al., 2025), gradient projection (Yu et al., 2020a), or conflict avoidance (Liu et al., 2021a) to significantly enhance performance. Notably, these two paradigms are not mutually exclusive. Recent studies (Qin et al., 2025b) have begun to integrate both strategies, and even incorporate higher-order information (e.g., model performance) to better coordinate MTL optimization.

**Shapley value.** The Shapley value (Shapley, 1953), a foundational allocation principle from cooperative game theory, has been increasingly applied to machine learning (Rozemberczki et al., 2022) for quantifying the marginal contribution of individual components to the overall system. It has been widely used to evaluate feature importance in model interpretability (Wang et al., 2022a; Chen et al., 2023; Kumar et al., 2020; Tang et al., 2023), data point utility in data valuation (Ghorbani & Zou, 2019; Jia et al., 2019), client contribution in federated learning (Fan et al., 2022; Rafi et al., 2024), and policy value in multi-agent reinforcement learning (Li et al., 2024; Wang et al., 2022b). While some studies (Wang et al., 2022a; Li et al., 2024) have explored incorporating Shapley value into MTL, a deep integration between the two remains absent. Moreover, the high computational complexity of Shapley value has hindered its practical adoption. In this work, we establish a principled connection between Shapley value and MTL optimization.

**Gram matrix.** The Gram matrix has been widely used in machine learning. As an inner product matrix in kernel methods, it enables nonlinear mappings (Pennington & Worah, 2017), reveals structural patterns in attention mechanisms (Teo & Nguyen, 2024), supports alignment kernels for time series (Cuturi, 2011), detects out-of-distribution samples (Sastry & Oore, 2020), accelerates kernel-based computations (Achlioptas et al., 2001), enhances the expressive power of graph neural networks (Morris et al., 2019), facilitates multi-modal alignment (Cicchetti et al., 2025), models diversity (Kulesza et al., 2012), and promotes policy diversity (Perez-Nieves et al., 2021) in reinforcement learning via determinantal point processes (Kulesza et al., 2012).

## 3 PRELIMINARIES

**Multi-task Learning.** In MTL, there are $N$ tasks, each with a loss function $\ell_i(\theta)$ for $i \in \{1, \ldots, N\}$, and the overall loss is $\mathcal{L} = (\ell_1, \ldots, \ell_N)$. The gradient of task $i$ is $g_i = \nabla l_i$, and the gradient matrix is $G = [g_1, \ldots, g_N]$. To jointly optimize all tasks, a common gradient-based approach is to aggregate gradients into an update direction $d := \sum_{i=1}^N \alpha_i g_i$, followed by the update $\theta_{t+1} = \theta_t - \eta d$, where $\alpha := (\alpha_1, ..., \alpha_N)^\top \in \mathbb{R}^N$ denotes the weights and $\eta$ denotes the step size. Using a first-order Taylor expansion, the change in loss for task $i$ can be approximated as $\ell_i(\theta_{t+1}) - \ell_i(\theta_t) \approx -\eta\, g_i^\top d$, which provides a guide for designing the update direction. Recent methods (Ban & Ji, 2024; Qin et al., 2025b) for determining the aggregation direction $d$ primarily optimize the utility $g_i^\top d$:

$$\text{FairGrad} : \arg\max_d \sum_{i=1}^N \frac{(g_i^\top d)^{1-\gamma}}{1-\gamma}, \quad \text{PIVRG} : \arg\min_d \frac{1}{N}\sum_{i=1}^N \frac{\omega_i}{g_i^\top d} \quad s.t. \quad g_i^\top d > 0, \forall i. \quad (1)$$

Here, $\gamma \in [0, 1) \cup (1, +\infty)$, $\omega_i := \frac{N \cdot \exp(\Delta m_i/\tau)}{\sum_{j=1}^N \exp(\Delta m_j/\tau)}$ is the higher-order information weight, $\Delta m_i$ is the performance drop (see Eq.4 for more details), and $\tau$ is the temperature parameter.

**Pareto Optimality.** For two points $\theta_1, \theta_2 \in \mathbb{R}^m$, $\theta_1$ is said to dominate $\theta_2$ if $\ell_i(\theta_1) \leq \ell_i(\theta_2)$ for all $i \in \{1, \ldots, N\}$ and $\mathcal{L}(\theta_1) \neq \mathcal{L}(\theta_2)$. A point $\theta^* \in \mathbb{R}^m$ is Pareto optimal if there does not exist any other point dominates it. The Pareto front is the set of all Pareto optimal points. Furthermore, a point is Pareto stationary if a convex combination of the task gradients at that point equals zero; this is a first-order necessary condition for Pareto optimality.

**Shapley Value.** For a cooperative game with $N$ participants, the Shapley value quantifies the marginal contribution of each participant to the total utility. Let $v(S)$ denote the utility of a subset $S \subseteq \{1, 2, ..., N\}$. The Shapley value $\phi_i$ for participant $i$ is defined as:

$$\phi_i = \sum_{S \subseteq N \setminus \{i\}} \frac{|S|!(N - |S| - 1)!}{N!} \left[ v(S \cup \{i\}) - v(S) \right]. \quad (2)$$

We set the $v(S)$ in the Shapley value to a gradient conflict metric that is a proper set function and faithfully reflects antagonism across tasks, so the induced Shapley value weights steer updates toward balanced Pareto improvement.

## 4 SVFAIR: A SCALABLE SHAPLEY VALUE FRAMEWORK FOR FAIR MTL

### 4.1 EVOLVING NEEDS FOR GRADIENT CONFLICT METRICS

Existing MTL methods have made initial progress in fairness optimization but largely ignore the systematic quantification of individual task gradient deviations from the overall gradient, thus limiting conflict mitigation while hindering overall performance improvement. Fortunately, Shapley value, a general method for quantifying individual contributions, provides a new perspective for addressing this issue. By integrating Shapley value, we quantify each task gradient's deviation from the overall gradient, providing a more precise gradient conflict signal to replace $\omega_i$ in Eq.1. So it is essential to design a suitable utility function that quantifies the gradient conflict of task subsets.

However, existing conflict metrics, such as cosine similarity (Yu et al., 2020a; Liu et al., 2021b) or inner product (Liu et al., 2021a) only analyze pairwise relationships between tasks, without capturing the collaborative and conflicting structure among the entire task set. As the number of tasks

$N$ increases, such local metrics face severe challenges in scalability and efficiency, making them unsuitable as utility function $v(S)$.

$$\text{Cosine Similarity}: \cos(g_i, g_j) = g_i^\top g_j / \parallel g_i \parallel \cdot \parallel g_j \parallel, \quad \text{Inner Product}: g_i^\top g_j, \quad (3)$$

An alternative is to define the utility function $v(S)$ as the performance drop $\Delta m$ (Ban & Ji, 2024; Qin et al., 2025b; Xiao et al., 2025) which is commonly used in MTL, quantifying the impact of task subsets on overall performance:

$$\Delta m\% = \frac{1}{K} \sum_{k=1}^{K} (-1)^{\delta_k} (M_{m,k} - M_{b,k})/M_{b,k} \times 100, \quad (4)$$

where $M_{b,k}$ and $M_{m,k}$ denote the scores of the baseline and the compared method on the $k$-th evaluation metric, respectively, and $\delta_k = 1$ indicates that a higher value of the $k$-th metric is preferable. However, employing $\Delta m$ as $v(S)$ requires a complete MTL model training for each subset $S$, resulting in an exponential computational complexity $\mathcal{O}(C \cdot 2^N)$, where $C$ denotes the cost of a single training run (See the analysis in Appendix B.4). Therefore, it is imperative to design a new efficient and scalable conflict metric that enables the practical application of Shapley value in MTL.

## 4.2 TOWARDS SCALABLE GRADIENT CONFLICT METRICS WITH PARALLELOTOPE VOLUME

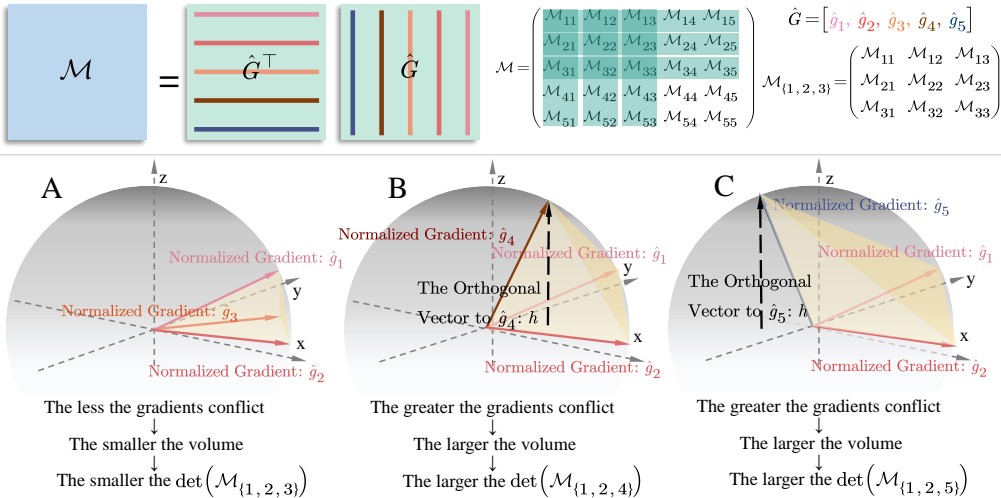

Figure 1: Overview of the VolDet metric. **Up:** Construction of the Gram matrix $\mathcal{M} = \hat{G}^\top \hat{G}$ from the normalized gradient matrix $\hat{G}$, and extraction of an example submatrix $\mathcal{M}_S$ for $S = \{1, 2, 3\}$ via simple indexing. **Down:** 3D geometric interpretation of partial parallelotope volume: (A) when task gradients are nearly aligned, the spanned volume and $\det(\mathcal{M}_S)$ are small; (B) when gradients diverge, both the volume and $\det(\mathcal{M}_S)$ grow; (C) given the same orthogonal vector $h$, B and C have identical volume and $\det(\mathcal{M}_S)$, yet C's obtuse (negative) alignments indicate stronger conflict.

To systematically quantify inter-task gradient conflict, we propose a gradient conflict metric as the squared volume of the parallelotope spanned by normalized task gradients. Geometrically, this volume reflects directional alignment: it approaches zero as gradients align and the shape collapses (Fig. 1-Down-A), and increases as directions diverge (Fig. 1-Down-B).

**Definition 1** *In MTL, the Gram matrix $\mathcal{M} \in \mathbb{R}^{N \times N}$ is defined as $\mathcal{M} = \hat{G}^\top \hat{G}$, where the normalized gradient matrix is $\hat{G} = [\hat{g}_1, \ldots, \hat{g}_N]$ and $\hat{g}_i = g_i / \|g_i\|$. According to the volume determinant identity, the squared volume of the parallelotope formed by these normalized gradients satisfies:* $\text{Vol}^2(\hat{g}_1, \ldots, \hat{g}_N) = \text{Vol}^2(\mathcal{M}) = \det(\mathcal{M})$. *The proof can be found in Appendix B.1.*

**Definition 2** *VolDet Metric (Volume-Determinant Metric). For any task subset $S \subseteq \{1, \ldots, N\}$, the squared volume of the corresponding subspace, as the utility function $v(S)$ of Shapley value, can be computed by extracting the submatrix $\mathcal{M}_S \in \mathbb{R}^{|S| \times |S|}$ from the global Gram matrix $\mathcal{M}$:*

$$v(S) := \text{Vol}^2(S) = \det(\mathcal{M}_S). \quad (5)$$

Remarkably, Definition 2 avoids the limitations of $\Delta m$ in Section 4.1. Instead, it directly captures the angular relationships among gradients and characterizes the overall distribution of gradient directions. The VolDet metric naturally satisfies the set function form required by the Shapley value (See the Appendix B.2 for more details). It does not depend on specific training outcomes or loss values but is solely determined by gradient geometry, which provides stronger generalization and fairness interpretability. However, VolDet is sign-agnostic. Since $\det(M_S) = \det\left(\hat{G}_S^\top \hat{G}_S\right) = \det\left(\hat{G}_S\right)^2$, it is invariant to any column sign flip $\hat{g}_i \to -\hat{g}_i$. Hence Fig. 1-Down-B and Fig. 1-Down-C yield the same $\det(M_S)$ when the third vector contributes the same orthogonal component, even though Fig. 1-Down-C contains obtuse (negative-cosine) pairs and thus stronger conflict.

To address the sign insensitivity of pure volumes while preserving the efficiency and Shapley-ready form of VolDet, we introduce a sign-aware refinement named VolDetPro. VolDetPro resolves this by augmenting VolDet with a lightweight penalty driven only by negative pairwise similarities in the Gram subspace, which activates in antagonistic regimes and remains zero otherwise.

**Definition 3** *VolDetPro. For any task subset $S \subseteq \{1, \ldots, N\}$, let $\mathcal{M}_S \in \mathbb{R}^{|S| \times |S|}$ be the Gram submatrix. Define the index set of antagonistic (negative-similarity) pairs on the strict upper triangle $\mathcal{I}(S) = \left\{ (i,j) : 1 \leq i < j \leq |S|, (\mathcal{M}_S)_{ij} < 0 \right\}$. Then VolDetPro can be defined as followed:*

$$v(S) = \det(\mathcal{M}_S) + \frac{\sqrt{|\mathcal{I}(S)|} + |S|}{|S|} \left| \sum_{(i,j) \in \mathcal{I}(S)} (\mathcal{M}_S)_{ij} \right|. \tag{6}$$

Here $\mathcal{I}(S)$ denotes the index set of antagonistic (negative-similarity) pairs on the strict upper triangle of $\mathcal{M}_S$. The term $\left| \sum_{(i,j) \in \mathcal{I}(S)} (\mathcal{M}_S)_{ij} \right|$ aggregates the conflict strength by accumulating the magnitudes of negative pairwise similarities, while the factor $\sqrt{|\mathcal{I}(S)|}$ encodes the conflict coverage (how many antagonistic pairs) with sublinear growth to avoid inflation as $|S|$ increases. VolDetPro equals VolDet when $\mathcal{I}(S) = \emptyset$ and otherwise increases smoothly with both the number and magnitude of antagonistic pairs, distinguishing configurations that share the same volume but exhibit stronger antagonism. More VolDetPro explanations are analyzed in Appendix B.3.

Most importantly, VolDet and VolDetPro scale efficiently via simple indexing on a single Gram matrix. After one round of gradient sampling and Gram construction, the $v(S)$ for all subsets $S$ is obtained without any per-subset retraining. Replacing the $\Delta m$ with the VolDet reduces Shapley value computation to one training pass while preserving conflict sensitivity. Computational complexity is analyzed in Appendix B.4, and a comparison of conflict metrics is provided in Appendix B.5.

### 4.3 Solve The Aggregated Gradient $d$ With Shapley Value Fairness

To apply the VolDet (Eq.5) or VolDetPro (Eq.6) metrics, we introduce a Shapley-Value-based gradient aggregation method.

**Shapley Value Fairness:** Based on Eq.2 and Eq.5 (or Eq.6), we can calculate the Shapley value $\phi_i$ for each task $i$, which quantifies the extent to which the task gradient $g_i$ deviates from the overall task gradient $G$. Intuitively, a higher $\phi_i$ indicates that task $i$ is more misaligned with others and should be assigned a higher influence. To achieve this, we incorporate $\phi_i$ into the optimization framework of Eq.1, and solve the following optimization problem to aggregate gradients:

$$\arg\min_d \ \frac{1}{N} \sum_{i=1}^N \omega_i / (g_i^\top d), \ \ \text{s.t. } g_i^\top d > 0, \ \forall i, \tag{7}$$

where $\omega_i = \frac{\exp(\phi_i/\tau)}{\sum_{j=1}^N \exp(\phi_j/\tau)}$. We refer to this Shapley-value-guided optimization framework as SVFair, which introduces a fairness-aware high-order information to the overall descent direction.

**Solve the Aggregated Gradient $d$:** To solve the Eq.7, we constrain $d$ on a sphere centered at the origin with radius $\epsilon$. The optimal $d^*$ can be solved with the Karush-Kuhn-Tucker (KKT) conditions: (1) The optimal solution lies exactly on the boundary of the Euclidean ball, i.e., $\|d^*\| = \epsilon$. (2) All the inner product constraints $g_i^\top d > 0$ are strictly satisfied. (3) Introducing Lagrange multiplier $\lambda$ to get the optimality condition: $\nabla f(d^*) = -\lambda d^*/\epsilon$. (4) Finally, the stationarity condition simplifies to Eq. 8.

$$\sum_{i=1}^N \frac{\epsilon N \omega_i}{\lambda (g_i^\top d)^2} g_i = d. \tag{8}$$

Following previous works (Ban & Ji, 2024) and (Qin et al., 2025b), we similarly assume that the gradients of tasks are linearly independent otherwise it would imply reaching a Pareto stationary point and $d$ can be represented as a linear combination of task gradients: $d = \sum_{i=1}^{N} \alpha_i g_i$, For simplicity, let constant item $\frac{\epsilon N}{\lambda} = 1$, we then obtain $\alpha_i = \omega_i/(g_i^\top d)^2$, and thus $(G^\top G \alpha)^2 = \omega/\alpha$,: where $\omega = [\omega_1, \ldots, \omega_N]^\top \in \mathbb{R}_+^N$, $\alpha = [\alpha_1, \ldots, \alpha_N]^\top \in \mathbb{R}_+^N$, and the square operation is element-wise. Algorithm 1 is the procedure of SVFair. See the Appendix D for proof details.

As mentioned earlier, we assume that the gradients of different tasks are linearly independent except at a Pareto stationary point. Formally, we adopt the following assumption, as also used in pior works (Navon et al., 2022; Ban & Ji, 2024; Qin et al., 2025b).

**Assumption 1** *For the output sequence $\{\theta_t\}$ generated by the proposed method, we make the following assumptions: 1) Each task loss function $l_i(\theta)$ is differentiable and L-smooth. 2) If $\theta_t$ is not a Pareto stationary point, the gradients $\{g_i(\theta_t)\}_{i=1}^N$ are linearly independent. 3) The update direction $d_t$ satisfies $\|d_t\| \le \epsilon$, and there exists a constant $m > 0$ such that $g_i(\theta_t)^\top d_t \ge m$.*

---

**Algorithm 1** SVFair for MTL

1: **Input:** model parameters $\theta_0$, temperature paramete $\tau$, learning rate $\{\eta_t\}$
2: **for** $t = 1$ **to** $T - 1$ **do**
3:     Compute normalized gradient matrix $\hat{G}(\theta_t)$
4:     Compute Gram matrix $\mathcal{M} = \hat{G}(\theta_t)^\top \hat{G}(\theta_t)$
5:     Compute Shapley values $\{\phi_i\}$ via Eq. 2 and utility function (Eq. 5:VolDet or Eq. 6:VolDetPro)
6:     Solve Eq. 7 to obtain gradient weights $\alpha_t$.
7:     Compute the aggregated gradient $d_t = G(\theta_t)\alpha_t$
8:     Update parameters $\theta_{t+1} = \theta_t - \eta_t d_t$
9: **end for**

---

**Theorem 1** *With the Assumption 1, there exists a subsequence $\{\theta_{t_j}\}$ of the output sequence $\{\theta_t\}$ that converges to a Pareto stationary point $\theta^*$. See the Appendix E.1 for more details.*

## 5 EXPERIMENTS

We evaluate SVFair on a variety of MTL benchmarks under toy example, supervised learning and reinforcement learning settings to show its effectiveness. Throughout, SVFair(VolDet) refers to SV-Fair using VolDet as the Shapley utility function $v(S)$, and SVFair(VolDetPro) refers to SVFair using VolDetPro as the $v(S)$. Furthermore, we compare our proposed SVFair method with the following methods in our experiments: Single-task learning (STL), DWA (Liu et al., 2019), UW (Kendall et al., 2018), MGDA (Sener & Koltun, 2018), RLW (Lin et al., 2021), PCGrad (Yu et al., 2020a), GradDrop (Chen et al., 2020), CAGrad (Liu et al., 2021a), IMTL-G (Liu et al., 2021b), Nash-MTL (Navon et al., 2022), FAMO (Liu et al., 2023), FairGrad (Ban & Ji, 2024) and PIVRG (Qin et al., 2025b). All experiments follow the settings of prior works (Navon et al., 2022; Liu et al., 2023; Ban & Ji, 2024; Qin et al., 2025b). Our code is available in the supplementary material.

### 5.1 TOY EXAMPLE

We adopt the 2-task toy example introduced in (Ban & Ji, 2024), where the objectives of the 2 tasks, denoted as $\mathcal{L}_1$ and $\mathcal{L}_2$. More details are provided in Appendix F.1. We select 5 starting points and illustrate the optimization trajectories with different MTL methods in Fig. 2.

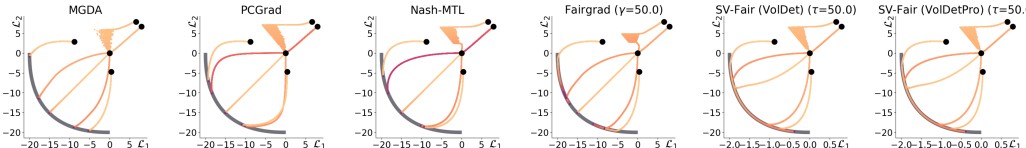

Figure 2: Comparison of MTL approaches on a challenging synthetic two-task benchmark. We visualize optimization trajectories w.r.t. objective values $(\mathcal{L}_1, \mathcal{L}_2)$. Starting points are indicated by black dots, and the Pareto front (see Section 3) is represented by thick gray lines.

Methods such as MGDA and Nash-MTL aim to find a Pareto stationary point, and their optimization typically halts once such a point is reached. As a result, when the gradient of a weak task is small,

Table 1: Results on NYU-v2 (3-task) and Cityscapes (2-task) datasets. Each experiment is repeated 3 times with different random seeds and the average is reported.

| Method | NYU-v2 | | Cityscapes | |
|---|---|---|---|---|
| | MR ↓ | $\Delta m\% \downarrow$ | MR ↓ | $\Delta m\% \downarrow$ |
| RLW | 13.33 | 7.78 | 11.00 | 24.37 |
| DWA | 10.89 | 3.57 | 9.00 | 21.43 |
| UW | 10.78 | 4.06 | 9.25 | 5.88 |
| MGDA | 8.78 | 1.38 | 12.25 | 44.14 |
| PCGrad | 11.22 | 3.97 | 9.12 | 18.21 |
| GradDrop | 10.56 | 3.58 | 8.25 | 23.67 |
| CAGrad | 8.22 | 0.19 | 7.88 | 11.58 |
| IMTL-G | 7.44 | -0.60 | 6.50 | 11.04 |
| Nash-MTL | 5.11 | -4.05 | 4.25 | 6.72 |
| FAMO | 6.67 | -4.10 | 8.75 | 8.13 |
| FairGrad | 4.78 | -4.66 | 3.25 | 5.12 |
| PIVRG | 3.56 | -6.50 | **2.50** | -0.45 |
| SVFair(VolDet) | **1.44** | **-8.81** | 6.88 | -2.08 |
| SVFair(VolDetPro) | 2.22 | -8.29 | 6.12 | **-2.40** |

these methods struggle to further improve its performance, leading to incomplete coverage of the Pareto front. FairGrad continues to move along the front beyond local stationary points and provides compensation for weaker tasks, but it exhibits noticeable bias depending on the initialization and fails to ensure uniform coverage. In contrast, SV-Fair enables smooth traversal along the Pareto front, allowing all trajectories to consistently reach more complete and balanced solutions. More experimental details and results are provided in Appendix F.1.

## 5.2 SUPERVISED LEARNING

We evaluate the performance of SVFair in 4 different supervised learning benchmarks, with experimental details and results provided in the Appendix F.2.

**Evaluation Metrics** Following prior works (Ban & Ji, 2024; Qin et al., 2025b), we consider two overall metrics to evaluate MTL performance of the MTL method: (1) $\Delta m\%$, (defined in Eq.4), the average per-task performance drop compared to the STL baseline ; (2) Mean Rank (MR), the average rank of each method across tasks (lower is better; MR = 1 means ranking first in all tasks).

**NYU-v2** (Silberman et al., 2012) contains 1,449 densely annotated images focused on indoor scene understanding. It supports one pixel-wise classification task (semantic segmentation) and two pixel-wise regression tasks (depth estimation and surface normal prediction). The result in Table 1 shows that SVFair achieves the best MR and overall performance drop on the NYU-V2 benchmark. As can be seen from Table 10 in Appendix F.2, an interesting observation is that previous methods tend to focus their improvements on Segmentation and Depth, while achieving little to no gains on the Surface Normal task—often even underperforming compared to STL—indicating a clear task imbalance. In contrast, both PIVRG and SVFair consistently outperform STL across all evaluation metrics. More importantly, SVFair outperforms PIVRG with near-optimal metric scores, achieving an almost perfect average rank and the best performance drop with a $\Delta m\%$ of -8.81.

**Cityscapes** (Cordts et al., 2016) contains 5,000 urban street view images, designed for two tasks: semantic segmentation and depth estimation. The result in Table 1 shows that SVFair also achieves the best overall performance drop on the Cityscapes benchmark. As can be seen from Table 11 in Appendix F.2, prior methods typically show strong improvements on Segmentation but struggle on the more challenging Depth task. SVFair places greater focus on Depth, effectively mitigating the negative impact of gradient conflicts. As a result, it not only achieves the best Depth metrics but also leads to an overall performance improvement over STL, indicated by a negative $\Delta m\%$.

**Office-31** (Saenko et al., 2010) contains 4,110 images spanning three domains (Amazon, DSLR, and Webcam), designed for image classification. The result in Table 2 shows that SVFair also achieves

Table 2: Results on Office-31 (3-task) and CelebA (40-task) datasets. Each experiment is repeated 3 times with different random seeds and the average is reported.

| Method | Office-31 | | | | | CelebA | |
| | Amazon ↑ | DSLR ↑ | Webcam ↑ | MR ↓ | $\Delta m\% \downarrow$ | MR ↓ | $\Delta m\% \downarrow$ |
|---|---|---|---|---|---|---|---|
| STL | 86.61 | 95.63 | 96.85 | | | | |
| RLW | 83.82 | 96.99 | 96.85 | 7.33 | 0.60 | 5.75 | 1.46 |
| DWA | 83.87 | 96.99 | 96.48 | 8.00 | 0.71 | 7.10 | 3.20 |
| UW | 83.82 | 97.27 | 96.67 | 7.17 | 0.56 | 6.00 | 3.23 |
| MGDA | 85.47 | 95.90 | 97.03 | 6.33 | 0.28 | 11.12 | 14.85 |
| PCGrad | 83.59 | 96.99 | 96.85 | 8.17 | 0.69 | 6.92 | 3.17 |
| GradDrop | 84.33 | 96.99 | 96.30 | 8.17 | 0.59 | 8.50 | 3.29 |
| CAGrad | 83.65 | 95.63 | 96.85 | 9.33 | 1.14 | 6.72 | 2.48 |
| IMTL-G | 83.41 | 96.72 | 96.48 | 10.50 | 0.98 | 4.90 | 0.84 |
| Nash-MTL | 85.01 | 97.54 | 97.41 | 3.83 | -0.24 | 5.32 | 2.84 |
| FairGrad | 82.11 | 97.54 | 97.96 | 6.17 | 0.68 | 5.78 | 0.37 |
| SVFair(VolDet) | 85.81 | **99.45** | **98.33** | 1.67 | -1.53 | **4.15** | **-0.63** |
| SVFair(VolDetPro) | **86.04** | **99.45** | **98.33** | **1.33** | **-1.62** | 5.72 | 0.11 |

the best overall performance drop on the Office-31 benchmark and attains the best accuracy across all three domains.

**CelebA** (Liu et al., 2015) is a large-scale facial attribute dataset containing over 200,000 images, each annotated with 40 attributes. It can therefore be formulated as a MTL problem with 40 image-level classification tasks. The result in Table 2 shows that SVFair also achieves the best overall performance drop on the CelebA benchmark.

## 5.3 REINFORCEMENT LEARNING

We evaluate SVFair on the MT10, which includes 10 robotic tasks from the MetaWorld environment (Yu et al., 2020b). The goal is to learn a policy that generalizes across diverse tasks. Following prior work, we adopt Soft Actor-Critic (SAC) (Haarnoja et al., 2018) as the base algorithm and train the model for 2 million steps with a batch size of 1280. We compare SVFair with Multi-task SAC (MTL SAC) (Yu et al., 2020b), CAGrad (Liu et al., 2021a), Nash-MTL (Navon et al., 2022), FAMO (Liu et al., 2023), FairGrad (Ban & Ji, 2024) and PIVRG (Qin et al., 2025b). The result is shown in Table 3. We evaluate each method every 10,000 steps and report the best average success rate achieved across 10 random seeds during the entire training process. The results indicate that SVFair achieves the best performance on the MT10.

Table 3: Results on MT10 benchmark. Average over 10 random seeds.

| Method | success rate (mean ± stderr) |
|---|---|
| STL | 0.90 ± 0.03 |
| MTL SAC | 0.49 ± 0.07 |
| CAGrad | 0.83 ± 0.05 |
| Nash-MTL | 0.91 ± 0.03 |
| FAMO | 0.83 ± 0.05 |
| FairGrad | 0.84 ± 0.07 |
| PIVRG | 0.96 ± 0.02 |
| SVFair(VolDet) | **0.97 ± 0.03** |
| SVFair(VolDetPro) | **0.97 ± 0.02** |

## 5.4 PRACTICAL IMPLEMENTATION AND INTEGRATION OF SHAPLEY VALUE

**Choice of Hyperparameter $\tau$.**

The Eq. 7 in SVFair relies on a temperature parameter $\tau$. To evaluate the effect of $\tau$ on the mean rank and overall performance drop, we conduct a small-scale hyperparameter sweep with $\tau \in \{0.5, 1.0, 2.0, 5.0\}$ on the Cityscapes, Office-31 and CelebA benchmarks. The result in Table 4

Table 4: $\Delta m\%$ of different $\tau$ on three benchmarks. SVFair uses VolDet for $v(S)$; see Appendix H for VolDetPro.

| Method | Cityscapes | Office-31 | CelebA |
|---|---|---|---|
| SVFair ($\tau = 0.5$) | -1.27 | **-1.49** | 1.20 |
| SVFair ($\tau = 1.0$) | 0.10 | -1.30 | 0.22 |
| SVFair ($\tau = 2.0$) | -0.78 | -1.44 | 0.23 |
| SVFair ($\tau = 5.0$) | **-2.16** | -0.97 | **-0.10** |

shows that the $\tau$ has different effects across benchmarks. On Cityscapes and CelebA, a larger $\tau$ (5.0) yields the greatest performance improvement, whereas on Office-31 only the smaller $\tau$ (0.5) achieves a slight positive gain. We recommend using a larger $\tau$ when Shapley values differ strongly and conflicts are sharp, and a smaller $\tau$ when tasks are more balanced; a detailed justification is provided in Appendix H.1.

**Compute Shapley Value.** For a small number of tasks, we directly compute the exact Shapley values. To further improve efficiency in large-task regimes (e.g., CelebA), we adopt *Monte Carlo (MC) subset sampling* (Rozemberczki et al., 2022): we randomly draw $K$ subsets $S$ to estimate Shapley values, reducing the enumeration cost. Table 5 shows that SVFair's single-epoch runtime is comparable to the baselines; on CelebA, even when evaluating many subsets, the one-time training cost remains dominant and the overall overhead is acceptable (the sample size $K$ in this work is 1000). Further experimental details and the MC procedure are provided in Appendix G.

Table 5: Single-epoch runtime (seconds) across benchmarks.

| Method | NYU-v2 | CityScapes | Office-31 | MT10 | CelebA (K=1k) | CelebA (K=2k) | CelebA (K=3k) | CelebA (K=5k) |
|---|---|---|---|---|---|---|---|---|
| FairGrad | 205.6 | 92.4 | 37.5 | 10.2 | 810 | 810 | 810 | 810 |
| SVFair | 209.0 | 94.8 | 38.1 | 10.3 | 870 | 936 | 980 | 1086 |

**Reinforcement Learning.** Following the implementation in FairGrad (Ban & Ji, 2024), we employ the SGD optimizer to solve the $(G^\top G\alpha)^2 = \omega/\alpha$ within the reinforcement learning setting. Specifically, we set the learning rate to 0.1, momentum to 0.5, and optimize over 20 epochs to effectively control computational costs. In addition, since MT10 involves 10 tasks, we also adopt the previously mentioned Monte Carlo sampling method to accelerate the computation of Shapley values.

**Integrate Shapley Value.** Previous MTL methods often overlook global gradient conflict information, leading to bad overall performance. We propose to integrate the Shapley Value-based global gradient conflict information $\omega_i$ (From Eq. 7) into these methods to enhance fairness in optimization. Specifically, we apply this strategy to a range of loss-based MTL methods, including RLW (Lin et al., 2021), DWA (Liu et al., 2019), and UW (Kendall et al., 2018). The loss is adjusted using the weight $\omega_i$, resulting in the modified loss $\mathcal{L}' = \omega \odot \mathcal{L}$. The result in Table 6 shows that incorporating global gradient conflict information effectively improves the performance of these methods.

Table 6: Results of integrating our Shapley value into existing methods on the NYU-v2 (3-task) dataset. Each experiment is repeated 3 times with different random seeds and the average is reported.

| Method | Segmentation | | Depth | | Surface Normal | | | | | $\Delta m\% \downarrow$ |
|---|---|---|---|---|---|---|---|---|---|---|
| | | | | | Angle $\downarrow$ | | Within $t° \uparrow$ | | | |
| | mIoU $\uparrow$ | PAcc $\uparrow$ | AErr $\downarrow$ | RErr $\downarrow$ | Mean | Median | 11.25 | 22.5 | 30 | |
| RLW | 37.17 | 63.77 | 0.5759 | 0.2410 | 28.27 | 24.18 | 22.26 | 47.05 | 60.62 | 7.78 |
| SV(VolDet)-RLW | 39.06 | 64.56 | 0.5551 | 0.2221 | 29.05 | 24.91 | 21.65 | 45.53 | 58.68 | **7.60** |
| SV(VolDetPro)-RLW | 39.14 | 64.85 | 0.56 | 0.22 | 29.04 | 24.99 | 21.53 | 45.37 | 58.59 | 7.71 |
| DWA | 39.11 | 65.31 | 0.5510 | 0.2285 | 27.61 | 23.18 | 24.17 | 50.18 | 62.39 | 3.57 |
| SV(VolDet)-DWA | 41.59 | 66.37 | 0.5482 | 0.2286 | 27.32 | 22.60 | 24.00 | 50.00 | 63.15 | **2.17** |
| SV(VolDetPro)-DWA | 41.88 | 66.48 | 0.54 | 0.22 | 27.48 | 22.85 | 23.75 | 49.31 | 62.66 | 2.20 |
| UW | 36.87 | 63.17 | 0.5446 | 0.2260 | 27.04 | 22.61 | 23.54 | 49.05 | 63.65 | 4.05 |
| SV(VolDet)-UW | 40.79 | 65.6 | 0.5416 | 0.2238 | 26.58 | 21.65 | 25.75 | 51.69 | 64.79 | 0.09 |
| SV(VolDetPro)-UW | 39.67 | 65.39 | 0.53 | 0.22 | 26.17 | 20.90 | 26.67 | 53.20 | 66.08 | **-1.50** |

## 5.5 VolDet VS VolDetPro

VolDetPro extends VolDet by penalizing both the number and the magnitude of negative-cosine pairs, making the metric more sensitive to obtuse interactions. When the **P**roportion of **N**egative **C**osine **P**airs (PNCP) in the normalized gradient matrix $\hat{G}$ is high, VolDetPro amplifies true conflict signals, shifts the Shapley weights away from opposing tasks, alleviates the tug-of-war, and improves the weakest task; it therefore tends to outperform VolDet. When PNCP is low, the additional penalty may overcorrect and suppress beneficial diversity, so VolDet is usually more stable.

As shown in Fig. 3, training-stage statistics indicate that PNCP on Office-31 stays above 0.25 for most of the training process, whereas PNCP on CelebA and Cityscapes remains below 0.25, which is consistent with the observed results. We thus conclude that VolDetPro is preferable when PNCS is high, whereas VolDet is preferable when PNCS is low. More details are provided in Appendix I.

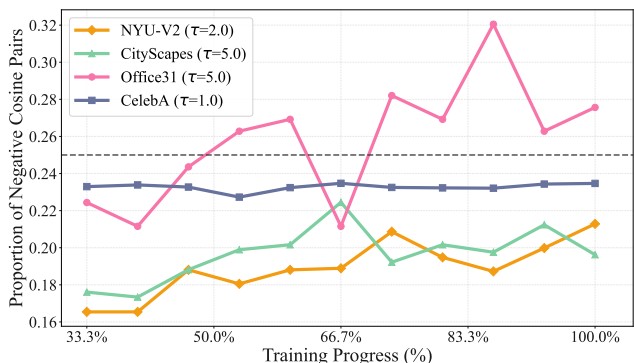

Figure 3: Proportion of Negative Cosine Pairs (PNCS) during training.

## 6 CONCLUSION

We presented SVFair, a Shapley-value framework that measures and mitigates gradient conflicts in multi-task learning through a subset-level geometric utility. At its core are two scalable metrics: VolDet, the Gram-determinant volume of normalized task gradients, and VolDetPro, a sign-aware refinement that penalizes negative cosine pairs to disambiguate obtuse interactions with identical volumes. Plugging these utilities into a single-pass Shapley computation yields fairness-aware weights that guide a provably convergent aggregation toward Pareto-stationary solutions. Empirically, SV-Fair achieves state-of-the-art results across supervised and reinforcement learning benchmarks, and further improves existing MTL methods when used as a drop-in fairness module. In practice, exact Shapley values are feasible for few tasks, while Monte-Carlo subset sampling keeps large-task regimes efficient without additional training runs. We also find VolDetPro preferable when negative cosine pairs are prevalent, whereas VolDet remains robust when conflicts are mostly acute.

### STATEMENT

**Ethics Statement** This paper raises no ethical concerns. Our study does not involve human subjects, sensitive personal data, or experiments that could potentially lead to ethical issues. All experiments are conducted on publicly available datasets, and the research process fully adheres to the ICLR Code of Ethics.

**Reproducibility Statement** To ensure reproducibility, we provide the complete source code in the supplementary materials. In addition, all theoretical results and detailed derivations are included in the appendix. Together, these resources enable our work to be independently verified and fully reproduced by the community.

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

APPENDIX CONTENTS

## A   USE OF LLMS

We made limited use of large language models (LLMs) during manuscript preparation, specifically to:

1. polish writing and check grammar;
2. assist with consistency and coherence checks of the text's logical flow.

All usage was conducted under author supervision. LLMs did not contribute to research design, experiment execution, result analysis, or the generation of substantive conclusions. The authors are solely responsible for all technical content, data, and conclusions.

## B   THE FULL DETAILS OF VOLDRT AND VOLDETPRO METRIC (I.E. SECTION 4.2)

### B.1   PROOF OF $\det(\mathcal{M}) = \mathrm{Vol}^2(\hat{g}_1, \ldots, \hat{g}_N)$

**Definition 1:** In MTL, the Gram matrix $\mathcal{M} \in \mathbb{R}^{N \times N}$ is defined as $\mathcal{M} = \hat{G}^\top \hat{G}$, where the normalized gradient matrix is $\hat{G} = [\hat{g}_1, \ldots, \hat{g}_N] \in \mathbb{R}^{d \times N}$ and $\hat{g}_i = g_i / \|g_i\|$. According to the volume determinant identity, the squared volume of the parallelotope formed by these normalized gradients satisfies: $\mathrm{Vol}^2(\hat{g}_1, \ldots, \hat{g}_N) = \mathrm{Vol}^2(\mathcal{M}) = \det(\mathcal{M})$.

We analyze the identity $\det(\mathcal{M}) = \mathrm{Vol}^2(\hat{g}_1, \ldots, \hat{g}_N)$ under the three cases of $d > N$, $d = N$, and $d < N$.

***Proof:***

**Case 1:** $d > N$

- **Linearly independent case:** If the column vectors $\{\hat{g}_i\}$ are linearly independent, then $\mathrm{rank}(\hat{G}) = N$, and $\mathcal{M}$ is positive definite. Performing a QR decomposition on $\hat{G}$ gives:

  $$\hat{G} = QR, \quad Q \in \mathbb{R}^{d \times N}, \; Q^\top Q = I_N, \; R \in \mathbb{R}^{N \times N} \text{ upper triangular and invertible.}$$

  Then

  $$\mathcal{M} = \hat{G}^\top \hat{G} = R^\top Q^\top Q R = R^\top R, \quad \det(\mathcal{M}) = (\det R)^2 = \left( \prod_{i=1}^{N} R_{ii} \right)^2 .$$

  The entries $R_{ii}$ correspond to the norms of the orthogonal vectors in the Gram–Schmidt process, so:

  $$\mathrm{Vol}(\hat{g}_1, \ldots, \hat{g}_N) = \prod_{i=1}^{N} \|u_i\| = \prod_{i=1}^{N} R_{ii},$$

  which implies

  $$\det(\mathcal{M}) = \mathrm{Vol}^2(\hat{g}_1, \ldots, \hat{g}_N).$$

- **Linearly dependent case:** If the vectors are linearly dependent, then $\mathrm{rank}(\hat{G}) < N$ and $\det(\mathcal{M}) = 0$. The parallelepiped degenerates and its volume is zero, so the identity still holds.

**Case 2:** $d = N$

Here $\hat{G} \in \mathbb{R}^{N \times N}$ is a square matrix.

- If $\hat{G}$ is invertible, then:

  $$\det(\mathcal{M}) = \det(\hat{G}^\top \hat{G}) = (\det \hat{G})^2.$$

  Since the volume of the parallelepiped spanned by the vectors in $\mathbb{R}^N$ is $\mathrm{Vol} = |\det \hat{G}|$, we have:

  $$\det(\mathcal{M}) = \mathrm{Vol}^2.$$

- If $\hat{G}$ is singular, then $\det(\mathcal{M}) = 0$ and $\mathrm{Vol} = 0$, and the identity holds trivially.

**Case 3:** $d < N$

In this case, at most $d$ vectors can be linearly independent, so $\mathrm{rank}(\hat{G}) \leq d < N$, and thus $\det(\mathcal{M}) = 0$. The $N$-dimensional parallelepiped in $\mathbb{R}^d$ collapses, yielding zero volume:

$$\mathrm{Vol}(\hat{g}_1, \ldots, \hat{g}_N) = 0, \quad \Rightarrow \quad \mathrm{Vol}^2 = 0 = \det(\mathcal{M}).$$

**Conclusion**

In all cases, we have:

$$\boxed{\det(\mathcal{M}) = \mathrm{Vol}^2(\hat{g}_1, \ldots, \hat{g}_N)}.$$

### B.2 MULTILINEARITY OF VOLDET METRIC AND COMPATIBILITY WITH SUBSET $v(S)$ OF SHAPLEY VALUE

**Proposition 1** *The VolDet metric $v(S) = \det(\mathcal{M}_S)$ is multilinear with respect to the input task gradients, and hence naturally satisfies the set function requirements needed for Shapley value computation.*

**Definition 4** *Let $\mathbb{V} = \mathbb{R}^N$ be a vector space. The determinant*

$$\det : \underbrace{\mathbb{V} \times \cdots \times \mathbb{V}}_{N \text{ times}} \to \mathbb{R}, \quad (v_1, \ldots, v_N) \mapsto \det[v_1, \ldots, v_N]$$

*is called* multilinear *if for any fixed position $i \in \{1, \ldots, N\}$, and for any vectors $v_i, w_i \in \mathbb{V}$ and scalars $\alpha, \beta \in \mathbb{R}$, the following holds:*

$$\det[\ldots, v_{i-1}, \alpha v_i + \beta w_i, v_{i+1}, \ldots]$$
$$= \alpha \cdot \det[\ldots, v_{i-1}, v_i, v_{i+1}, \ldots] + \beta \cdot \det[\ldots, v_{i-1}, w_i, v_{i+1}, \ldots].$$

**Definition 5** *Given a task subset $S \subseteq \{1, \ldots, N\}$, let $\{\hat{g}_i\}_{i \in S}$ denote the normalized task gradients. Then the Gram submatrix is defined as*

$$\mathcal{M}_S = \hat{G}_S^\top \hat{G}_S \in \mathbb{R}^{|S| \times |S|},$$

*where $\hat{G}_S = [\hat{g}_i]_{i \in S} \in \mathbb{R}^{d \times |S|}$. The $k$-th column of $\mathcal{M}_S$ is*

$$C_k = \left[\langle \hat{g}_{i_j}, \hat{g}_{i_k} \rangle\right]_{j=1}^{|S|}, \quad \text{where } S = \{i_1, \ldots, i_{|S|}\}.$$

**Definition 6** *Suppose that for some $i \in S$ (corresponding to column $k$), we have:*

$$\hat{g}_i = \alpha u + \beta w, \quad \text{with } u, w \in \mathbb{R}^d, \ \alpha, \beta \in \mathbb{R}.$$

*Then the $k$-th column becomes:*

$$C_k = \left[\langle \hat{g}_{i_j}, \alpha u + \beta w \rangle\right]_j = \alpha C_k^u + \beta C_k^w,$$

*where $C_k^u = \left[\langle \hat{g}_{i_j}, u \rangle\right]_j$ and similarly for $C_k^w$.*

***Proof of Proposition 1:***

By applying the Definitions 4, 5, 6 of the determinant function to the $k$-th column:

$$\det[C_1, \ldots, C_{k-1}, \alpha C_k^u + \beta C_k^w, C_{k+1}, \ldots, C_{|S|}]$$
$$= \alpha \det[C_1, \ldots, C_{k-1}, C_k^u, \ldots] + \beta \det[C_1, \ldots, C_{k-1}, C_k^w, \ldots].$$

That is, let:

$$v(S) = \det(\mathcal{M}_S)$$

Hence, for the subset utility with $\hat{g}_i = \alpha u + \beta w$:

$$v(S) = \alpha \cdot v(S)(\hat{g}_i = u) + \beta \cdot v(S)(\hat{g}_i = w).$$

This precisely aligns with the Shapley value's requirement that the subset utility function be linearly decomposable with respect to individual task gradient, thereby demonstrating that the utility function constructed by VolDet inherently satisfies the formal condition of marginal contribution additivity.

### B.3    VOLDET VS VOLDETPRO

**Volume base term** $\det(M_S)$**.** The determinant of the Gram submatrix $\det(M_S)$ measures the squared volume spanned by the normalized task gradients in subset $S$, capturing directional diversity and independence. A small determinant indicates that gradients are nearly collinear, leading to stronger risks of mutual interference, whereas a larger determinant reflects more diverse descent directions. By retaining this base term, VolDetPro can still faithfully reflect structural correlation conflicts even without antagonistic pairs. This choice also ensures consistency with the classical volume-based metric, allowing for intuitive interpretation and comparison. In particular, when $I(S) = \varnothing$, $\det(M_S)$ serves as the sole contribution, providing a natural degenerate case.

**Conflict strength term** $\left|\sum_{(i,j)\in I(S)}(M_S)_{ij}\right|$**.** This term accumulates only the negative cosine similarities in the strict upper triangle, directly quantifying the overall magnitude of antagonism. Whenever some task pairs form obtuse angles, these entries become strongly negative and thus significantly increase the score, reflecting the inherent optimization difficulty. Unlike the volume term, this component can discriminate between cases with the same $\det(M_S)$ but stronger antagonism, thereby making VolDetPro more sensitive to genuine conflict. When no negative pairs exist, the term naturally vanishes, avoiding spurious contributions. Hence, the measure jointly covers both "crowding" due to correlation and "collision" due to antagonism.

**Coverage and scaling factor** $\sqrt{|I(S)| + |S|}/|S|$**.** The square-root dependence on the number of antagonistic pairs $|I(S)|$ provides sublinear growth, preventing uncontrolled escalation when the task number increases. The denominator $|S|$ normalizes the scale, ensuring comparability across subsets of different sizes and suppressing combinatorial explosion. This factor balances the contribution of coverage (more pairs involved) and per-pair intensity, without being dominated by either extreme. As the number of antagonistic pairs grows, the amplification factor increases moderately, reflecting the intuition that broader coverage implies more substantial conflict. Overall, this design guarantees stability under varying task scales and sampling noise.

**Natural degeneration and computational feasibility.** When $I(S) = \varnothing$, VolDetPro reduces exactly to VolDet, i.e., $v(S) = \det(M_S)$, maintaining interpretability consistent with the volume metric. This avoids generating artificial penalties during stages with no antagonism, such as early training or near convergence. The formulation only requires constructing the normalized gradient Gram matrix once, after which all subset values $v(S)$ can be indexed directly. No extra retraining or complex optimization is needed, resulting in low computational and memory overhead. This property also facilitates integration with Shapley-value-based aggregation, since both rely on the same Gram structure.

**Sensitivity to both quantity and intensity of antagonism.** Unlike metrics that consider only the number or the strength of conflicts, VolDetPro couples these two aspects in its second term. The more negative pairs exist and the closer each is to $-1$, the larger the cumulative penalty becomes, signaling more severe optimization difficulties. This enables meaningful comparison between scenarios such as "few strong conflicts" versus "many mild conflicts." When volumes are similar, the configuration with stronger or denser antagonism always yields higher $v(S)$. Therefore, VolDetPro provides stronger discriminative power during model selection and training.

**Resolving misjudgment in obtuse-collinear scenarios.** Classical volume can approach zero when two gradients are nearly opposite, mistakenly suggesting "low conflict." VolDetPro corrects this by assigning a large penalty through its second term, which is consistent with the intuition that opposing gradients induce severe conflict. This eliminates the paradox that $\theta$ and $180° - \theta$ yield the same determinant but represent very different levels of antagonism. For multi-task optimization, this correction exposes hard-to-align task pairs earlier in training. Hence, the monotonicity of the measure is strengthened around critical boundary cases.

### B.4    COMPUTATIONAL COMPLEXITY ANALYSIS.

In many cooperative game scenarios, computing the Shapley value has been proven to be *NP-Hard*, and even *#P-Complete* (Michalak et al., 2013). Therefore, we will analyze the computational complexity in Definition 2.

We provide a detailed comparison of the computational complexity analysis for four different methods, aiming to highlight the efficiency advantages of VolDet under large-scale tasks (Such as CelebA):

**1. $\Delta m\%$ (Exhaustive Enumeration)** This baseline method uses the performance drop metric $\Delta m\%$ to estimate the marginal contribution of each task subset. It requires training a separate MTL model for every one of the $2^N$ possible task subsets, leading to an exponential computational complexity:

$$\mathcal{O}(C \cdot 2^N),$$

where $C$ denotes the cost of a single MTL model training. While feasible for small $N$, this approach becomes computationally prohibitive as the number of tasks grows.

**2. $\Delta m\%$ + Monte Carlo Sampling** To mitigate the exponential cost, a Monte Carlo strategy can be employed, where only $K \ll 2^N$ subsets are randomly sampled for training and evaluation. This reduces the overall complexity to:

$$\mathcal{O}(K \cdot C).$$

However, since each subset still requires a full training run, the method remains expensive in practice, especially for deep networks with high $C$. The procedure of Monte Carlo sampling (Rozemberczki et al., 2022) is summarized in Algorithm 2.

**3. VolDet Metric (Exhaustive Enumeration)** This approach eliminates the need for retraining by computing gradient-based cooperative utility after a single full training with cost $\mathcal{O}(C)$. It includes gradient normalization for $N$ task gradients of dimension $d$ with complexity $\mathcal{O}(Nd)$, construction of the Gram matrix from normalized gradients with complexity $\mathcal{O}(N^2d)$, and determinant computation for all $2^N$ subsets, each costing $\mathcal{O}(N^3)$, resulting in $\mathcal{O}(2^N \cdot N^3)$ in total. The overall complexity is:

$$\mathcal{O}(C + Nd + N^2d + 2^N N^3).$$

This is acceptable for small $N$, where matrix computations are negligible compared to the training cost. If the number of tasks N is small, then:

$$\mathcal{O}(C + Nd + N^2d + 2^N N^3) \approx \mathcal{O}(C).$$

**4. VolDet Metric + Monte Carlo Sampling** To enable scalability in large-task settings such as CelebA with 40 tasks, we apply Monte Carlo sampling to the VolDet framework. After a single full training pass with cost $\mathcal{O}(C)$, we perform gradient normalization for $N$ tasks at $\mathcal{O}(Nd)$ and construct the Gram matrix at $\mathcal{O}(N^2d)$. We then randomly sample $K$ subsets and compute the determinant for each, costing $\mathcal{O}(K \cdot N^3)$ in total. The overall complexity is thus:

$$\mathcal{O}(C + Nd + N^2d + KN^3) \approx \mathcal{O}(C),$$

It is important to note that in real-world MTL scenarios, model parameters often reach tens of millions, making the training cost $\mathcal{O}(C)$ (ranging from hours to days) significantly larger than the determinant computation cost $\mathcal{O}(N^3)$, i,e, $\mathcal{O}(N^3) \ll \mathcal{O}(C)$. This further highlights the efficiency advantage of our approach.

**5. VolDetPro Metric (Exhaustive Enumeration)** Compared with VolDet, VolDetPro adds a lightweight penalty computed from the strict upper-triangular negative entries of the Gram submatrix $\mathcal{M}_S$. After one full training with cost $\mathcal{O}(C)$, we normalize $N$ task gradients at $\mathcal{O}(Nd)$ and build the Gram matrix at $\mathcal{O}(N^2d)$. For each of the $2^N$ subsets, we (i) compute $\det(\mathcal{M}_S)$ at $\mathcal{O}(|S|^3)$ and (ii) scan and accumulate negative entries to form the penalty at $\mathcal{O}(|S|^2)$, which is asymptotically dominated by the determinant cost. Hence the overall complexity remains

$$\mathcal{O}(C + Nd + N^2d + 2^N N^3).$$

This is acceptable for small $N$, and when $N$ is small relative to model size,

$$\mathcal{O}(C + Nd + N^2d + 2^N N^3) \approx \mathcal{O}(C).$$

**6. VolDetPro Metric + Monte Carlo Sampling** To scale to large-task regimes, we adopt Monte Carlo sampling with $K \ll 2^N$ subsets. After a single full training pass $\mathcal{O}(C)$, gradient normalization $\mathcal{O}(Nd)$, and Gram construction $\mathcal{O}(N^2d)$, we evaluate $K$ subsets. For each subset, $\det(\mathcal{M}_S)$

Table 7: Comparison of Training Runs (TR), Determinant Calculations (DC), and Overall Computational Complexity for Six Methods

| Method | TR | DC | Overall Complexity |
|---|---|---|---|
| $\Delta m\%$ (Exhaustive) | $2^N$ | 0 | $O(C \cdot 2^N)$ |
| $\Delta m\%$ + Monte Carlo | $K$ | 0 | $O(K \cdot C)$ |
| VolDet (Exhaustive) | 1 | $2^N \cdot O(N^3)$ | $O(C + Nd + N^2d + 2^N N^3)$ |
| VolDet + Monte Carlo | 1 | $K \cdot O(N^3)$ | $O(C + Nd + N^2d + KN^3) \approx O(C)$ |
| VolDetPro (Exhaustive) | 1 | $2^N \cdot O(N^3)$ | $O(C + Nd + N^2d + 2^N N^3)$ |
| VolDetPro + Monte Carlo | 1 | $K \cdot O(N^3)$ | $O(C + Nd + N^2d + KN^3) \approx O(C)$ |

costs $\mathcal{O}(|S|^3)$ and the penalty aggregation costs $\mathcal{O}(|S|^2)$ (negligible in order). Therefore, the overall complexity is

$$\mathcal{O}(C + Nd + N^2d + KN^3) \approx \mathcal{O}(C),$$

since in practice $\mathcal{O}(N^3) \ll \mathcal{O}(C)$ for deep MTL models, and $K$ is modest.

---

**Algorithm 2** Monte Carlo Estimation of Shapley Values

---

1: **Input:** number of players $N$, utility function $v(\cdot)$, sample size $K$
2: **Output:** estimated Shapley values $\{\phi_i\}_{i=1}^N$

3: *Initialize:* $\phi[i] \leftarrow 0$ for all $i = 1, \ldots, N$
4: **for** $t = 1$ **to** $K$ **do**
5:     $\pi \leftarrow \text{RandomPermutation}(\{1, \ldots, N\})$
6:     $S \leftarrow \emptyset$
7:     **for** $j = 1$ **to** $N$ **do**
8:         $i \leftarrow \pi[j]$
9:         $\Delta \leftarrow v(S \cup \{i\}) - v(S)$
10:       $\phi[i] \leftarrow \phi[i] + \Delta$
11:       $S \leftarrow S \cup \{i\}$
12:     **end for**
13: **end for**
14: **for** $i = 1$ **to** $N$ **do**
15:     $\phi[i] \leftarrow \phi[i]/K$
16: **end for**
17: **Return** $\{\phi_i\}_{i=1}^N$

---

### B.5 COMPARISON OF COMMON MTL CONFLICT METRICS

We will present the formal definitions of four widely used conflict metrics in MTL, along with a comparative analysis of their advantages and limitations.

**1: Cosine Similarity**

    **Definition:** $\cos(g_i, g_j) = \frac{g_i^\top g_j}{\|g_i\| \cdot \|g_j\|}$.

    **Advantages:**

- Depends solely on direction; insensitive to gradient magnitude.
- Computationally efficient: $\mathcal{O}(d)$.

    **Limitations:**

- Measures only pairwise angular relations; unable to capture higher-order task interactions.
- Requires $\mathcal{O}(N^2)$ comparisons as the number of tasks increases.
- Cannot directly reflect the collective conflict within a task subset.

**2: Inner Product**

> **Definition:** $g_i^\top g_j$.
>
> **Advantages:**
>
> - Directly quantifies gradient projection.
> - More efficient than cosine similarity due to no normalization overhead.
>
> **Limitations:**
>
> - Also limited to pairwise relations.
> - Sensitive to gradient norm differences; requires normalization or rescaling.
> - Lacks correspondence to global task conflict strength.

**3: Performance Drop ($\Delta m$)**

> **Definition:** $\Delta m\% = \frac{1}{K} \sum_{k=1}^{K} (-1)^{\delta_k} \cdot \frac{M_{m,k} - M_{b,k}}{M_{b,k}} \times 100 \qquad (i.e. Eq.\ 4)$.
>
> **Advantages:**
>
> - Reflects the actual performance impact of a task subset.
> - Semantically interpretable; aligns well with business objectives.
>
> **Limitations:**
>
> - Sensitive to evaluation metrics, task scale, and other non-structural factors.
> - Requires training a separate model for each subset; complexity $\mathcal{O}(C \cdot 2^N)$.
> - Non-differentiable; unsuitable for end-to-end learning integration.

**4: VolDet**

> **Definition:** Let $S \subseteq \{1, \ldots, N\}$ be a task subset. Define the normalized gradient matrix $\hat{G}_S$ and the corresponding Gram submatrix $\mathcal{M}_S$ (i.e. Definition 2):
>
> $$\mathcal{M}_S = [\hat{g}_i^\top \hat{g}_j]_{i,j \in S}, \quad \text{where} \quad v(S) = \det(\mathcal{M}_S).$$
>
> **Advantages:**
>
> - Captures global geometric conflict among multiple tasks, beyond pairwise relations.
> - Requires only a single round of gradient sampling and matrix construction; all subset utilities can be computed via index-based operations.
> - Continuous, differentiable, and scale-invariant; solely depends on gradient directions.
>
> **Limitations:**
>
> - Exact evaluation for all subsets still requires $\mathcal{O}(2^N \cdot D)$; must be approximated via sampling (See the Appendix B.4).

**5: VolDetPro**

> **Definition:** Let $S \subseteq \{1, \ldots, N\}$ be a task subset. Define the index set of antagonistic pairs on the strict upper triangle $\mathcal{I}(S) = \{(i,j) \mid 1 \le i < j \le |S|,\ (\mathcal{M}_S)_{ij} < 0\}$ (i.e. Definition 3):
>
> $$v_{\mathrm{pro}}(S) = \det(\mathcal{M}_S) + \frac{\sqrt{|\mathcal{I}(S)|} + |S|}{|S|} \left| \sum_{(i,j) \in \mathcal{I}(S)} (\mathcal{M}_S)_{ij} \right|.$$
>
> **Advantages:**

- *Sign-aware*: distinguishes acute vs. obtuse configurations that share the same volume (resolves VolDet's sign insensitivity).

- Preserves VolDet's practicality: single-pass gradient sampling and Gram construction; all subset utilities via index-based operations; Shapley-ready.

- Scale-invariant (uses normalized gradients) and almost everywhere differentiable; monotone w.r.t. antagonism (and strength of negative pairs).

- Lightweight overhead: per-subset penalty aggregation costs $\mathcal{O}(|S|^2)$, dominated by the determinant $\mathcal{O}(|S|^3)$; same big-$\mathcal{O}$ as VolDet.

**Limitations:**

- The absolute-value term is non-smooth at $0$ (subgradients apply), though this rarely affects optimization in practice.

## 6: Summary:

- *Local vs Global*: Cosine and inner product are limited to pairwise relations, while the Gram determinant captures full-set geometry.

- *Efficiency & Single-pass*: Replacing outcome-dependent $\Delta m$ with Gram-based volumes (VolDet/VolDetPro) enables computing $v(S)$ for all subsets from a single round of gradient sampling and Gram construction via indexing operations, greatly reducing the practical overhead of Shapley-value computation.

- *Robustness & Differentiability*: VolDet is norm-invariant and (almost everywhere) differentiable, making it suitable for end-to-end integration; it is also stable to gradient rescaling and rotations.

- *Sign-awareness (VolDetPro)*: VolDetPro resolves the sign insensitivity of pure volumes, distinguishing acute vs. obtuse configurations that share the same volume; it equals VolDet when $I(S) = \varnothing$ and increases with both the number and magnitude of negative similarities, while keeping the same big-$\mathcal{O}$ complexity as VolDet.

## C EXAMPLE: SHAPLEY VALUE COMPUTATION

We provide an example of Shapley value computation based on the VolDet metric, where the task number $N$ is 4. Let gradient matrix $G^\top$ is:

$$G^\top = [g_1, ..., g_N]^\top = \begin{pmatrix} 1 & 0 & 0 & 0 \\ 1 & 1 & 0 & 0 \\ 1 & 0 & 1 & 0 \\ 1 & 1 & 1 & 1 \end{pmatrix}, \quad N = 4.$$

We define the utility function of Shapely value is $v(S) := \text{Vol}^2(S) = \det(\mathcal{M}_S)$, (i.e.Eq. 5), where $g_i$ is the gradient of task $i$, and $\mathcal{M}$ is the Gram matrix of the normalized gradient matrix. ***Note.*** The Shapley-Value computation is identical for VolDetPro; one only needs to replace Definition 2 with Definition 3. Since the set-function form and indexing operations remain unchanged and only the utility values differ by a sign-aware additive term, we present the worked example using VolDet for brevity.

### C.1 NORMALIZE GRADIENTS

The normalized gradient matrix $\hat{G}^\top$ is:

$$\hat{G}^\top = [\hat{g}_1, ..., \hat{g}_N]^\top = \begin{pmatrix} 1.0000 & 0.0000 & 0.0000 & 0.0000 \\ 0.7071 & 0.7071 & 0.0000 & 0.0000 \\ 0.7071 & 0.0000 & 0.7071 & 0.0000 \\ 0.5000 & 0.5000 & 0.5000 & 0.5000 \end{pmatrix}$$

where $\hat{g}_i = g_i / \|g_i\|$.

### C.2 GRAM MATRIX $\mathcal{M}$

The Gram matrix $\mathcal{M}$ is:

$$\mathcal{M} = \hat{G}^\top \hat{G} = \begin{pmatrix} 1.0000 & 0.7071 & 0.7071 & 0.5000 \\ 0.7071 & 1.0000 & 0.5000 & 0.7071 \\ 0.7071 & 0.5000 & 1.0000 & 0.7071 \\ 0.5000 & 0.7071 & 0.7071 & 1.0000 \end{pmatrix}$$

### C.3 SHAPLEY VALUE FORMULA

For each task $i$, the Shapley value $\phi_i$ is computed as (i.e. 2):

$$\phi_i = \sum_{S \subseteq \{0,...,3\} \setminus \{i\}} \frac{|S|!(N - |S| - 1)!}{N!} [v(S \cup \{i\}) - v(S)]. \tag{9}$$

Since $N = 4$, the weight function is:

$$w(k) = \frac{k!(3-k)!}{4!}, \quad w(0) = 0.25, \quad w(1) = 0.08333, \quad w(2) = 0.08333, \quad w(3) = 0.25.$$

We compute the determinant $v(S)$ of some subsets as examples:

1. Subset $S = \{1\}$

$$v(\{1\}) = \det \left( \mathcal{M}_{\{1\},\{1\}} \right) = \det \left( [\mathcal{M}_{11}] \right) = \mathcal{M}_{11} = 1.$$

2. Subset $S = \{2, 3\}$

$$v(\{2,3\}) = \det \begin{bmatrix} \mathcal{M}_{22} & \mathcal{M}_{23} \\ \mathcal{M}_{32} & \mathcal{M}_{33} \end{bmatrix} = \det \begin{bmatrix} 1 & 0.7071 \\ 0.7071 & 1 \end{bmatrix} = 1 - (0.7071)^2 \approx 0.50.$$

Table 8: Shapley Value Contributions for $|S| = 1$

| $S$ | $v(S)$ | $v(S \cup \{i\})$ | $\Delta = v(S \cup \{0\}) - v(S)$ | Contribution $= w(1) \cdot \Delta$ |
|---|---|---|---|---|
| $\{1\}$ | 1 | 0.5000 | $-0.5000$ | $-0.04167$ |
| $\{2\}$ | 1 | 0.5000 | $-0.5000$ | $-0.04167$ |
| $\{3\}$ | 1 | 0.7500 | $-0.2500$ | $-0.02083$ |

Table 9: Shapley Value Contributions for $|S| = 2$

| $S$ | $v(S)$ | $v(S \cup \{i\})$ | $\Delta$ | Contribution $= w(2) \cdot \Delta$ |
|---|---|---|---|---|
| $\{1, 2\}$ | 0.7500 | 0.2500 | $-0.5000$ | $-0.04167$ |
| $\{1, 3\}$ | 0.5000 | 0.2500 | $-0.2500$ | $-0.02083$ |
| $\{2, 3\}$ | 0.5000 | 0.2500 | $-0.2500$ | $-0.02083$ |

3. Subset $S = \{1, 2, 3\}$

$$v(\{1, 2, 3\}) = \det \begin{bmatrix} \mathcal{M}_{11} & \mathcal{M}_{12} & \mathcal{M}_{13} \\ \mathcal{M}_{21} & \mathcal{M}_{22} & \mathcal{M}_{23} \\ \mathcal{M}_{31} & \mathcal{M}_{32} & \mathcal{M}_{33} \end{bmatrix} = \det \begin{bmatrix} 1 & 0.5000 & 0.7071 \\ 0.5000 & 1 & 0.7071 \\ 0.7071 & 0.7071 & 1 \end{bmatrix} \approx 0.25.$$

## C.4 Contributions for $\phi_0$

Let the "contribution" be denoted as $w(|S|) * [v(S \cup \{i\}) - v(S)]$, and $\Delta = v(S \cup \{i\}) - v(S)$.

1. $|S| = 0$
   - **Weight:** $w(0) = 0.25$.
   - **Difference:** $\Delta = v(\{0\}) - v(\varnothing) = 1 - 0 = 1$.
   - **Contribution:** $w(0) \times \Delta = 0.25 \times 1 = 0.25$.

2. $|S| = 1$
   - **Weight:** $w(1) = \frac{1!2!}{4!} = \frac{1}{12} \approx 0.08333$.
   - **Difference:** See the Table 8.
   - **Contribution:** $\sum w(1)\Delta = -0.04167 - 0.04167 - 0.02083 = -0.10417$.

3. $|S| = 2$
   - **Weight:** $w(2) = \frac{2!1!}{4!} = \frac{1}{12} \approx 0.08333$.
   - **Difference:** See the Table 9
   - **Contribution:** $\sum w(2)\Delta = -0.04167 - 0.02083 - 0.02083 = -0.08333$.

4. $|S| = 3$
   - **Weight:** $w(3) == 0.25$.
   - **Difference:** $v(\{0, 1, 2, 3\}) - v(\{1, 2, 3\}) = 0.1250 - 0.2500 = -0.1250$
   - **Contribution:** $w(3)\Delta = 0.25 \times (-0.1250) = -0.03125$.

## C.5 Summary of $\phi_0$

$$\phi_0 = 0.25 + (-0.10417) + (-0.08333) + (-0.03125) = 0.03125.$$

Other $\phi_1$, $\phi_2$, and $\phi_3$ follow the same procedure, and can be computed as:

> **Final Shapley Value:** $[\, 0.03125, \ 0.03125, \ 0.03125, \ 0.03125 \,]$

## C.6  SOFTMAX NORMALIZATION

According to Eq. 1, with emperature paramete$\tau = 1$, we have:

$$\omega_i = \frac{e^{\phi_i}}{\sum_{j=0}^{3} e^{\phi_j}} = \frac{e^{0.03125}}{4\,e^{0.03125}} = 0.25.$$

**Final** $\omega_i$ **Value:** $\begin{bmatrix} 0.25, & 0.25, & 0.25, & 0.25 \end{bmatrix}$

# D  PROOF OF SVFAIR OPTIMIZATION PROBLEM

OPTIMIZATION OBJECTIVE

We consider the following SVFair gradient aggregation problem (See Eq. 7 in Section 4.3 for more details):

$$\min_d \ \frac{1}{N} \sum_{i=1}^{N} \frac{\omega_i}{g_i^\top d} \quad \text{s.t.} \ g_i^\top d > 0, \ \forall i,$$

We constrain $d$ on the surface of an $\ell_2$-ball of radius $\epsilon$, i.e. $\|d\| = \epsilon$. By applying the Karush–Kuhn–Tucker (KKT) conditions, one obtains:

**Step 1:** The optimum lies on the boundary: $\|d^*\| = \epsilon$.

**Step 2:** All inner-product constraints are strictly satisfied: $g_i^\top d^* > 0$, $\forall i$.

**Step 3:** Introducing a Lagrange multiplier $\lambda$ yields the stationarity condition $\nabla f(d^*) = -\lambda \frac{d^*}{\epsilon}$.

**Step 4:** Finally, the stationarity condition simplifies to Eq. 10.

$$\sum_{i=1}^{N} \frac{\epsilon N \omega_i}{\lambda (g_i^\top d)^2} g_i = d. \tag{10}$$

*Proof of Step 1*

Proof by contradiction. Assume that $\|d^*\| < \epsilon$. Since $g_i^T d^* > 0$, for any $\eta > 1$ and $\eta \|d^*\| \leq \epsilon$, define

$$\tilde{d} = \eta d^*.$$

Thus, it is evident that:

$$\|\tilde{d}\| = \eta \|d^*\| \leq \epsilon, \quad g_i^T \tilde{d} = \eta (g_i^T d^*) > 0,$$

indicating that $\tilde{d}$ is still feasible.

By scaling, we obtain:

$$f(\tilde{d}) = \frac{1}{N} \sum_{i=1}^{N} \frac{\omega_i}{g_i^T (\eta d^*)} = \frac{1}{\eta} \frac{1}{N} \sum_{i=1}^{N} \frac{\omega_i}{g_i^T d^*} = \frac{1}{\eta} f(d^*) < f(d^*),$$

which contradicts the optimality of $d^*$.

Therefore, the assumption does not hold, we must have:

$$\|d^*\| = \epsilon.$$

*Proof of Step 2*

Proof by contradiction. Assume that there exists some index $j$ such that:

$$g_j^T d^* = 0.$$

Then, we have:

$$f(d^*) = \frac{1}{N} \sum_{i=1}^{N} \frac{\omega_i}{g_i^T d^*} \geq \frac{\omega_j}{N \cdot 0} = +\infty,$$

which contradicts the finiteness of the objective value at the optimal solution.

Therefore, the assumption does not hold, we must have

$$g_i^T d^* > 0, \quad \forall i.$$

*Proof of Step 3*

1. KKT Preliminaries

The objective function $f(d)$ is convex over the feasible region $\{d : \|d\| \leq \epsilon, \ g_i^T d > 0\}$.

There exists a strictly feasible point (e.g., taking $\|d\| \ll \epsilon$ and all $g_i^T d > 0$) satisfying Slater's condition, implying that the KKT conditions are both necessary and sufficient for optimality.

2. Lagrangian Function

Define the inequality constraints as:

$$h_0(d) = \|d\| - \epsilon \leq 0, \quad h_i(d) = -g_i^T d \leq 0 \ (i = 1, \ldots, N).$$

The Lagrangian function is then given by:

$$\mathcal{L}(d, \lambda, \mu) = f(d) + \lambda h_0(d) + \sum_{i=1}^{N} \mu_i h_i(d),$$

where $\lambda, \mu_i \geq 0$.

3. Stationarity Condition

$$\nabla_d \mathcal{L}(d^*, \lambda, \mu) = \nabla f(d^*) + \lambda \nabla \|d^*\| + \sum_{i=1}^{N} \mu_i(-g_i) = 0.$$

Note that $\nabla \|d^*\| = d^*/\|d^*\|$, and $\nabla(-g_i^T d) = -g_i$.

4. Complementary Slackness and Feasibility

Since $\|d^*\| = \epsilon$ (from Step 1), it follows that $\lambda \geq 0$ and $\lambda(\|d^*\| - \epsilon) = 0 \implies \lambda$ is feasible.

Since $g_i^T d^* > 0$ (from Step 2), $h_i(d^*) < 0 \implies \mu_i = 0 \ \forall i$.

5. Simplifying Stationarity

Substituting $\|d^*\| = \epsilon$ and $\mu_i = 0$, we obtain:

$$\nabla f(d^*) + \lambda \frac{d^*}{\epsilon} = 0 \implies \nabla f(d^*) = -\lambda \frac{d^*}{\epsilon}.$$

6. $\lambda > 0$

If $\lambda = 0$, then $\nabla f(d^*) = 0$, indicating that the unconstrained optimal $d^*$ is a stationary point. However, since $\|d^*\| \leq \epsilon$ is not enforced, we can arbitrarily increase $\|d\|$, making each term $\omega_i/(g_i^T d)$ smaller and unbounded, contradicting the finite optimal objective value. Hence, $\lambda > 0$.

*Proof of Step 4*

For the function $h_i(d) = \frac{\omega_i}{g_i^T d}$, the gradient is given by:

$$\nabla h_i(d) = -\frac{\omega_i}{(g_i^T d)^2} g_i.$$

Thus, the gradient of $f(d)$ can be expressed as:

$$\nabla f(d) = \frac{1}{N} \sum_{i=1}^{N} \nabla h_i(d) = -\frac{1}{N} \sum_{i=1}^{N} \frac{\omega_i}{(g_i^T d)^2} g_i.$$

From Step 3, we have:

$$\nabla f(d^*) = -\lambda \frac{d^*}{\epsilon}.$$

Substituting the gradient expression above, we obtain:

$$\frac{1}{N} \sum_{i=1}^{N} \frac{\omega_i}{(g_i^T d^*)^2} g_i = \frac{\lambda}{\epsilon} d^*.$$

Multiplying both sides by $\epsilon/(\lambda N)$, we get:

$$d^* = \frac{\epsilon}{\lambda N} \sum_{i=1}^{N} \frac{\omega_i}{(g_i^T d^*)^2} g_i = \sum_{i=1}^{N} \alpha_i g_i,$$

where

$$\alpha_i = \frac{\epsilon \omega_i}{\lambda N (g_i^T d^*)^2}.$$

Since $\omega_i > 0$ and $g_i^T d^* > 0$, we have $\alpha_i > 0$. Thus, $d^*$ is located within the convex hull formed by $\{g_i\}$.

# E  SVFAIR ANALYSIS

## E.1  SVFAIR CONVERGENCE ANALYSIS

**Theorem 1:** Suppose Assumption 1 is satisfied. There exists a subsequence $\{\theta_{t_j}\}$ of the output sequence $\{\theta_t\}$ that converges to a Pareto stationary point $\theta^*$, with a convergence rate of $\mathcal{O}(1/\varepsilon^2)$.

*Proof:* **Step 1**

According to Assumptions 1, suppose that $\mathcal{L}$ is $L$-smooth. We have:

$$\mathcal{L}(\theta_{t+1}) \leq \mathcal{L}(\theta_t) + \nabla\mathcal{L}(\theta_t)^\top(\theta_{t+1} - \theta_t) + \frac{L}{2}\|\theta_{t+1} - \theta_t\|^2.$$

Since $\theta_{t+1} = \theta_t - \eta_t d_t$, we have:

$$\mathcal{L}(\theta_{t+1}) - \mathcal{L}(\theta_t) \leq -\eta_t \nabla\mathcal{L}(\theta_t)^\top d_t + \frac{L}{2}\eta_t^2\|d_t\|^2.$$

To guarantee

$$\mathcal{L}(\theta_{t+1}) \leq \mathcal{L}(\theta_t), \tag{11}$$

the $\eta_t$ should be satisfied:

$$0 < \eta_t \leq \frac{2\nabla\mathcal{L}(\theta_t)^\top d_t}{L\|d_t\|^2}. \tag{12}$$

Meanwhile, we compute:

$$\nabla\mathcal{L}(\theta_t)^\top d_t = \frac{1}{N}\sum_{i=1}^{N} g_i(\theta_t)^\top d_t = \frac{1}{N}\sum_{i=1}^{N}\sqrt{\frac{\omega_{i,t}}{\alpha_{i,t}}}, \quad \|d_t\|^2 = \sum_{i=1}^{N}\sqrt{\omega_{i,t}\alpha_{i,t}}.$$

Thus, we obtain the upper bound:

$$\eta_t \leq \frac{\frac{2}{N}\sum_{i=1}^{N}\sqrt{\omega_{i,t}/\alpha_{i,t}}}{L\sum_{i=1}^{N}\sqrt{\omega_{i,t}\alpha_{i,t}}}.$$

That is, selecting any value within the interval $(0, \frac{2\sum_{i=1}^{N}\sqrt{\omega_{i,t}/\alpha_{i,t}}}{LN\sum_{i=1}^{N}\sqrt{\omega_{i,t}\alpha_{i,t}}})$ ensures Eq. 11 holds. For example, taking $\eta_t = \frac{2\sum_{i=1}^{N}\sqrt{\omega_{i,t}/\alpha_{i,t}}}{LN\sum_{i=1}^{N}\sqrt{\omega_{i,t}\alpha_{i,t}}}$ halved, i.e.

$$\eta_t = \frac{\sum_{i=1}^{N}\sqrt{\omega_{i,t}/\alpha_{i,t}}}{LN\sum_{i=1}^{N}\sqrt{\omega_{i,t}\alpha_{i,t}}} = \frac{\nabla\mathcal{L}(\theta_t)^\top d_t}{L\|d_t\|^2}. \tag{13}$$

*Step 2: Convergence*

According to Eq. 11 and Eq. 13, we have:

$$\mathcal{L}(\theta_{t+1}) \leq \mathcal{L}(\theta_t) \implies \{\mathcal{L}(\theta_t)\}_{t\geq 0}$$

is monotonically non-increasing.

Since each individual loss $\ell_i(\theta)$ is bounded below, so:

$$\mathcal{L}(\theta) = \frac{1}{N}\sum_{i=1}^{N}\ell_i(\theta) \geq L^* > -\infty.$$

We conclude:

$$\lim_{t\to\infty}\mathcal{L}(\theta_t) =: \mathcal{L}^* \quad \text{(a finite value)},$$

With Eq. 13, the Eq. 11 becomes:

$$\mathcal{L}(\theta_t) - \mathcal{L}(\theta_{t+1}) \geq \frac{3(\nabla\mathcal{L}(\theta_t)^\top d_t)^2}{2L\|d_t\|^2}.$$

Suppose $\|\nabla\mathcal{L}(\theta_t)\| \nrightarrow 0$. Then there exists $\varepsilon > 0$ and a subsequence $\{t_j\}$ such that $\|\nabla\mathcal{L}(\theta_{t_j})\| \geq \varepsilon$ for all $j$. By Assumption 1 (boundedness of $\|d_t\|$), assume:

$$\nabla\mathcal{L}(\theta_{t_j})^\top d_{t_j} \geq m > 0, \quad \|d_{t_j}\| \leq \epsilon.$$

Let $\Delta_t := \mathcal{L}(\theta_t) - \mathcal{L}(\theta_{t+1})$, then we have:

$$\Delta_{t_j} \geq \frac{3m^2}{2L\epsilon^2} =: \delta > 0.$$

Infinitely many positive lower bounds $\delta$ would imply that $\mathcal{L}(\theta_t)$ continues decreasing without bound, contradicting convergence to $\mathcal{L}^*$. Therefore,

$$\lim_{t\to\infty} \|\nabla\mathcal{L}(\theta_t)\| = 0.$$

Since $\nabla\mathcal{L}(\theta_t) = \frac{1}{N}\sum_{i=1}^{N} g_i(\theta_t)$, if there exists any task $j$ such that $\|g_j(\theta_t)\| \nrightarrow 0$, then the individual loss $\ell_j(\theta_t)$ would continue decreasing, contradicting the boundedness of $\mathcal{L}(\theta_t)$. Hence,

$$\lim_{t\to\infty} \|g_i(\theta_t)\| = 0, \quad \forall i.$$

Since $\|g_i(\theta_t)\| \to 0$, by continuity we conclude $g_i(\theta^*) = 0$. Hence the gradient matrix at $\theta^*$ ($\theta^*$ is a Pareto stationary point):
$$G(\theta^*) = [g_1(\theta^*), ..., g_N(\theta^*)] = 0,$$

is the zero matrix (See the Preliminaries Section 3 for the definition of Pareto stationary point).

### Step 3: Iteration Complexity for Gradient Norm Accuracy

We aim to find the minimum iteration index $T$ such that the gradient norm satisfies the target accuracy $\varepsilon > 0$:
$$\|\nabla\mathcal{L}(\theta_T)\| \leq \varepsilon.$$

When $\|\nabla\mathcal{L}(\theta_t)\| > \varepsilon$, by the Cauchy–Schwarz inequality, we have:

$$\nabla\mathcal{L}(\theta_t)^\top d_t \leq \|\nabla\mathcal{L}(\theta_t)\|\|d_t\|,$$

and therefore,
$$\Delta_t \geq \frac{3\|\nabla\mathcal{L}(\theta_t)\|^2\|d_t\|^2}{2L\|d_t\|^2} = \frac{3\|\nabla\mathcal{L}(\theta_t)\|^2}{2L} \geq \frac{3\varepsilon^2}{2L} =: \delta > 0.$$

Summing this lower bound from $t = 0$ to $T - 1$, we get:

$$\mathcal{L}(\theta_0) - \mathcal{L}^* = \sum_{t=0}^{T-1} \Delta_t \geq T \cdot \delta,$$

which implies:

$$T \leq \frac{\mathcal{L}(\theta_0) - \mathcal{L}^*}{\delta} = \frac{2L(\mathcal{L}(\theta_0) - \mathcal{L}^*)}{3\varepsilon^2} = \mathcal{O}(1/\varepsilon^2).$$

Hence, to achieve a gradient norm smaller than $\varepsilon$, the number of iterations is at most $\mathcal{O}(1/\varepsilon^2)$, which matches the classical convergence rate of first-order methods under non-convex $L$-smooth settings.

In non-convex $L$-smooth settings, gradient descent with a fixed step size $\eta = \frac{1}{L}$ achieves a classical convergence rate of $\mathcal{O}(1/\varepsilon^2)$ for reaching gradient norm accuracy $\varepsilon > 0$. Denote the full objective as $\mathcal{L}(\theta)$ and the update rule as:

$$\eta = \frac{1}{L}.$$

With the update $\theta_{t+1} = \theta_t - \frac{1}{L}\nabla\mathcal{L}(\theta_t)$, the descent lemma implies:

$$\mathcal{L}(\theta_{t+1}) \leq \mathcal{L}(\theta_t) - \frac{1}{L}\|\nabla\mathcal{L}(\theta_t)\|^2 + \frac{L}{2}\cdot\frac{1}{L^2}\|\nabla\mathcal{L}(\theta_t)\|^2 \qquad (14)$$

$$= \mathcal{L}(\theta_t) - \frac{1}{2L}\|\nabla\mathcal{L}(\theta_t)\|^2. \qquad (15)$$

Let $\Delta_t := \mathcal{L}(\theta_t) - \mathcal{L}(\theta_{t+1})$, then:

$$\Delta_t \geq \frac{1}{2L}\|\nabla\mathcal{L}(\theta_t)\|^2.$$

Summing from $t = 0$ to $T - 1$, and using monotonicity and boundedness, with $L^* = \inf_\theta \mathcal{L}(\theta)$, we obtain:

$$\sum_{t=0}^{T-1}\|\nabla\mathcal{L}(\theta_t)\|^2 \leq 2L\sum_{t=0}^{T-1}\Delta_t \leq 2L[\mathcal{L}(\theta_0) - L^*].$$

Let $i$ be the index such that $\|\nabla\mathcal{L}(\theta_i)\|$ is minimized. Then:

$$T\cdot\|\nabla\mathcal{L}(\theta_i)\|^2 \leq \sum_{t=0}^{T-1}\|\nabla\mathcal{L}(\theta_t)\|^2 \leq 2L[\mathcal{L}(\theta_0) - L^*].$$

To ensure $\|\nabla\mathcal{L}(\theta_i)\| \leq \varepsilon$, it suffices that:

$$T \geq \frac{2L[\mathcal{L}(\theta_0) - L^*]}{\varepsilon^2} = \mathcal{O}(1/\varepsilon^2).$$

### E.2 VolDet-based Fairness of SVFair

#### E.2.1 Feasible sets and regularity

Per-task improvement for direction $d$ is $u_i(d) = g_i^\top d$, $i = 1, \ldots, N$.

Fix $\varepsilon > 0$ and assume:

**Assumption 2 (Interior feasible direction)** *There exists $\bar{d} \in \mathbb{R}^p$ such that*

$$\|\bar{d}\| \leq \varepsilon, \qquad g_i^\top \bar{d} > 0, \quad i = 1, \ldots, N.$$

Pick any $\delta \in \left(0, \ \min_i g_i^\top \bar{d}\right)$.

**Definition 7 (Feasible direction and improvement sets)** *Define*

$$\mathcal{D} = \left\{ d \in \mathbb{R}^p : \|d\| \leq \varepsilon, \ g_i^\top d \geq \delta, \ i = 1, \ldots, N \right\}, \tag{16}$$

$$\mathcal{U} = \left\{ u \in \mathbb{R}^N : \exists d \in \mathcal{D}, \ u_i = g_i^\top d, \ i = 1, \ldots, N \right\}. \tag{17}$$

Then:

**Definition 8 (Geometry of $\mathcal{D}$ and $\mathcal{U}$)** *Under Assumption 2:*

1. *$\mathcal{D}$ is non-empty, compact, and convex.*

2. *Let $G = [g_1, \ldots, g_N] \in \mathbb{R}^{p \times N}$. Then $\mathcal{U} = G^\top \mathcal{D}$, hence $\mathcal{U}$ is non-empty, compact, and convex.*

3. *For all $u \in \mathcal{U}$, $u_i \geq \delta > 0$.*

**Proof** (1) $\mathcal{D}$ is the intersection of the closed ball $\{d : \|d\| \leq \varepsilon\}$ and finitely many closed half-spaces $\{d : g_i^\top d \geq \delta\}$, hence closed and bounded (compact) and convex. Assumption 2 and the choice of $\delta$ imply $\bar{d} \in \mathcal{D}$, so $\mathcal{D} \neq \emptyset$.

(2) $G^\top$ is linear; linear images of convex (resp. compact, non-empty) sets are convex (resp. compact, non-empty), hence $\mathcal{U}$ has the desired properties.

(3) From Definition 7, $u_i = g_i^\top d \geq \delta$ for all $d \in \mathcal{D}$.

#### E.2.2 VolDet fairness functional

**Definition 9 (VolDet fairness functional)** *For $u \in \mathbb{R}_{++}^N$,*

$$\mathcal{F}(u) = \sum_{i=1}^N \frac{\omega_i}{u_i}.$$

**Definition 10 (VolDet-fair allocation)** *A vector $u^\star \in \mathcal{U}$ is* VolDet-fair *if*

$$\mathcal{F}(u^\star) = \min_{u \in \mathcal{U}} \mathcal{F}(u).$$

*If the minimizer is unique, $u^\star$ is the* unique *VolDet-fair allocation.*

**Definition 11 (Strict convexity)** *On $\mathbb{R}_{++}^N$, $u \mapsto \mathcal{F}(u)$ is strictly convex.*

**Proof** For each $i$, $f_i(u_i) = \frac{\omega_i}{u_i}, u_i > 0$ satisfies

$$f_i''(u_i) = \frac{2\omega_i}{u_i^3} > 0.$$

Thus $f_i$ is strictly convex and so is the sum $\mathcal{F}(u) = \sum_i f_i(u_i)$.

### E.2.3  UNCONSTRAINED STRUCTURE OF $\sum_i \omega_i / u_i$

**Proposition 2 (Closed form under linear sum constraint)** *Let $U > 0$, $\omega_i > 0$. Consider*

$$\min_{u \in \mathbb{R}_{++}^N} \sum_{i=1}^{N} \frac{\omega_i}{u_i} \quad s.t. \quad \sum_{i=1}^{N} u_i = U. \tag{$\mathcal{P}_U$}$$

*Then the unique solution is*

$$u_i^\star = c\sqrt{\omega_i}, \qquad c = \frac{U}{\sum_{j=1}^{N} \sqrt{\omega_j}},$$

*and hence*

$$\frac{u_i^\star}{\sqrt{\omega_i}} \equiv c, \quad \forall i.$$

**Proof** Lagrangian:

$$\mathcal{L}(u, \lambda) = \sum_{i=1}^{N} \frac{\omega_i}{u_i} + \lambda \Big( \sum_{i=1}^{N} u_i - U \Big).$$

KKT:

$$-\frac{\omega_i}{u_i^2} + \lambda = 0 \ \Rightarrow \ u_i = \sqrt{\frac{\omega_i}{\lambda}}.$$

Sum constraint gives

$$\frac{1}{\sqrt{\lambda}} \sum_{i=1}^{N} \sqrt{\omega_i} = U \ \Rightarrow \ \sqrt{\lambda} = \frac{1}{U} \sum_{i=1}^{N} \sqrt{\omega_i}.$$

Strict convexity (Definition 11) implies uniqueness.

### E.2.4  SVFAIR AS UNIQUE VOLDET-FAIR DIRECTION

SVFair direction at this iteration solves

$$\min_{d \in \mathcal{D}} \sum_{i=1}^{N} \frac{\omega_i}{g_i^\top d}. \tag{$\mathcal{P}_{\text{SV}}$}$$

Note that $u_i = g_i^\top d$ maps equation $\mathcal{P}_{\text{SV}}$ to

$$\min_{u \in \mathcal{U}} \mathcal{F}(u).$$

**Theorem 2 (SVFair $\Rightarrow$ unique VolDet-fair allocation)** *Under Assumption 2 and Definition 8, problem equation $\mathcal{P}_{\text{SV}}$ admits at least one solution $d^{\text{SV}}$ and the induced vector*

$$u^{\text{SV}} = u(d^{\text{SV}})$$

*satisfies*

$$\mathcal{F}(u^{\text{SV}}) = \min_{u \in \mathcal{U}} \mathcal{F}(u).$$

*Moreover, $u^{\text{SV}}$ is unique, hence the unique VolDet-fair allocation.*

**Proof** By Definition 8, $\mathcal{U}$ is non-empty and compact. By Definition 11, $\mathcal{F}$ is continuous and strictly convex on $\mathbb{R}_{++}^N$, in particular on $\mathcal{U}$. Thus $\mathcal{F}$ attains a unique minimizer $u^{\text{SV}}$ over $\mathcal{U}$. Existence of at least one $d^{\text{SV}}$ with $u(d^{\text{SV}}) = u^{\text{SV}}$ follows from the definition of $\mathcal{U}$.

(If needed, an additional non-degeneracy assumption on the linear map $d \mapsto G^\top d$ can be imposed to ensure uniqueness of $d^{\text{SV}}$ itself.)

### E.2.5 FAIRGRAD UNDER VOLDET FAIRNESS

FairGrad-type methods solve, for some $\gamma > 0$,

$$\max_{d \in \mathcal{D}} \sum_{i=1}^{N} \frac{(g_i^\top d)^{1-\gamma}}{1-\gamma} \quad \Longleftrightarrow \quad \max_{u \in \mathcal{U}} \sum_{i=1}^{N} \frac{u_i^{1-\gamma}}{1-\gamma}. \tag{$\mathcal{P}_{\text{FG}}$}$$

Let $u^{\text{FG}}$ be any optimizer of equation $\mathcal{P}_{\text{FG}}$ in $\mathcal{U}$.

**Assumption 3 (Non-uniform VolDet–Shapley weights)** *There exist $i, j$ such that $\omega_i \neq \omega_j$.*

**Theorem 3 (SVFair vs. FairGrad under VolDet fairness)** *Under Assumptions 2 and 3, let $u^{\text{SV}}$ be as in Theorem 2 and let $u^{\text{FG}}$ solve equation $\mathcal{P}_{\text{FG}}$. Then*

$$\mathcal{F}(u^{\text{SV}}) < \mathcal{F}(u^{\text{FG}})$$

*unless $u^{\text{FG}} = u^{\text{SV}}$ (degenerate coincidence).*

**proof** By Theorem 2, $u^{\text{SV}}$ is the unique minimizer of $\mathcal{F}$ on $\mathcal{U}$. Thus, for any $u \in \mathcal{U} \setminus \{u^{\text{SV}}\}$,

$$\mathcal{F}(u^{\text{SV}}) < \mathcal{F}(u).$$

Apply this to $u^{\text{FG}}$.

## F EXPERIMENTS DETAILS

### F.1 EXPERIMENTS DETAILS FOR TOY EXAMPLE (DETAILS FOR SECTION 5.1).

Following the setup in prior works (Liu et al., 2023; Ban & Ji, 2024; Qin et al., 2025b), we employ three distinct two-task toy examples as presented in CAGrad (Liu et al., 2021a). These examples encompass varying levels of gradient conflict and optimization difficulty. Each two-task scenario ($\mathcal{L}_1(\theta)$ and $\mathcal{L}_2(\theta)$) is defined on $\theta = (\theta_1, \theta_2)^\top \in \mathbb{R}^2$, allowing for a comprehensive comparison of MTL approaches across different task pairs. We use sky blue to highlight the differences among the three two-task settings.

Following the setup in prior works (Liu et al., 2023; Ban & Ji, 2024; Qin et al., 2025b), we use five distinct starting points $\{(-8.5, 7.5), (0, 0), (9.0, 9.0), (-7.5, -0.5), (9.0, -1.0)\}$. The Adam optimizer is employed with a $1 \times 10^{-3}$ learning rate. The result is shown in the Figs 4-6. Furthermore, we compare our proposed SVFair method with the following methods in our experiments: UW (Kendall et al., 2018), RLW (Lin et al., 2021), DWA (Liu et al., 2019), IMTL-G (Liu et al., 2021b), MGDA (Sener & Koltun, 2018), PCGrad (Yu et al., 2020a), CAGrad (Liu et al., 2021a), Nash-MTL (Navon et al., 2022), FAMO (Liu et al., 2023), and FairGrad (Ban & Ji, 2024). All experiments follow the settings of prior works (Liu et al., 2021a; Navon et al., 2022; Liu et al., 2023; Ban & Ji, 2024; Qin et al., 2025b). Our code is available in the supplementary material. Here, $\gamma$ and $\tau$ are the hyperparameters of their respective algorithms.

In all three two-task toy examples, SV-Fair requires only a single hyperparameter $\tau \approx 0.1$–$50$ to smoothly and trace entirely out the Pareto front between Task A and Task C. Its convergence is stable, exhibiting neither extreme oscillations nor bias toward any single task. By contrast, FAMO/FairGrad are highly sensitive to their trade-off parameter $\gamma$ and demand extensive tuning, while conventional baselines often oscillate in conflict regions or cover only one end of the front. Consequently, SV-Fair outperforms these methods regarding solution diversity, stability, and ease of use.

**Two-Task A From FAMO (Liu et al., 2023)**

$$\mathcal{L}_1(\theta) = 0.1 \cdot f_1(\theta)g_1(\theta) + f_2(\theta)h_1(\theta),$$
$$\mathcal{L}_2(\theta) = f_1(\theta)g_2(\theta) + f_2(\theta)h_2(\theta),$$

where the functions are defined as follows:

$$f_1(\theta) = \max(\tanh(0.5\theta_2), 0),$$
$$f_2(\theta) = \max(\tanh(-0.5\theta_2), 0),$$
$$g_1(\theta) = \log\left(\max(|0.5(-\theta_1 - 7) - \tanh(-\theta_2)|, 0.000005)\right) + 6,$$
$$g_2(\theta) = \log\left(\max(|0.5(-\theta_1 + 3) - \tanh(-\theta_2) + 2|, 0.000005)\right) + 6,$$
$$h_1(\theta) = \frac{((-\theta_1 + 7)^2 + 0.1(-\theta_2 - 8)^2)}{10} - 20,$$
$$h_2(\theta) = \frac{((-\theta_1 - 7)^2 + 0.1(-\theta_2 - 8)^2)}{10} - 20.$$

**Two-Task B From FairGrad (Ban & Ji, 2024)**

$$\mathcal{L}_1(\theta) = 0.1 \cdot f_1(\theta)g_1(\theta) + f_2(\theta)h_1(\theta),$$
$$\mathcal{L}_2(\theta) = f_1(\theta)g_2(\theta) + f_2(\theta)h_2(\theta),$$

where the functions are defined as follows:

$$f_1(\theta) = \max(\tanh(0.5\theta_2), 0),$$
$$f_2(\theta) = \max(\tanh(-0.5\theta_2), 0),$$
$$g_1(\theta) = \log\left(\max(|0.5(-\theta_1 - 7) - \tanh(-\theta_2)|, 0.000005)\right) + 6,$$
$$g_2(\theta) = \log\left(\max(|0.5(-\theta_1 + 3) - \tanh(-\theta_2) + 2|, 0.000005)\right) + 6,$$
$$h_1(\theta) = \frac{((-\theta_1 + 7)^2 + 0.1(-\theta_1 - 8)^2)}{10} - 20,$$

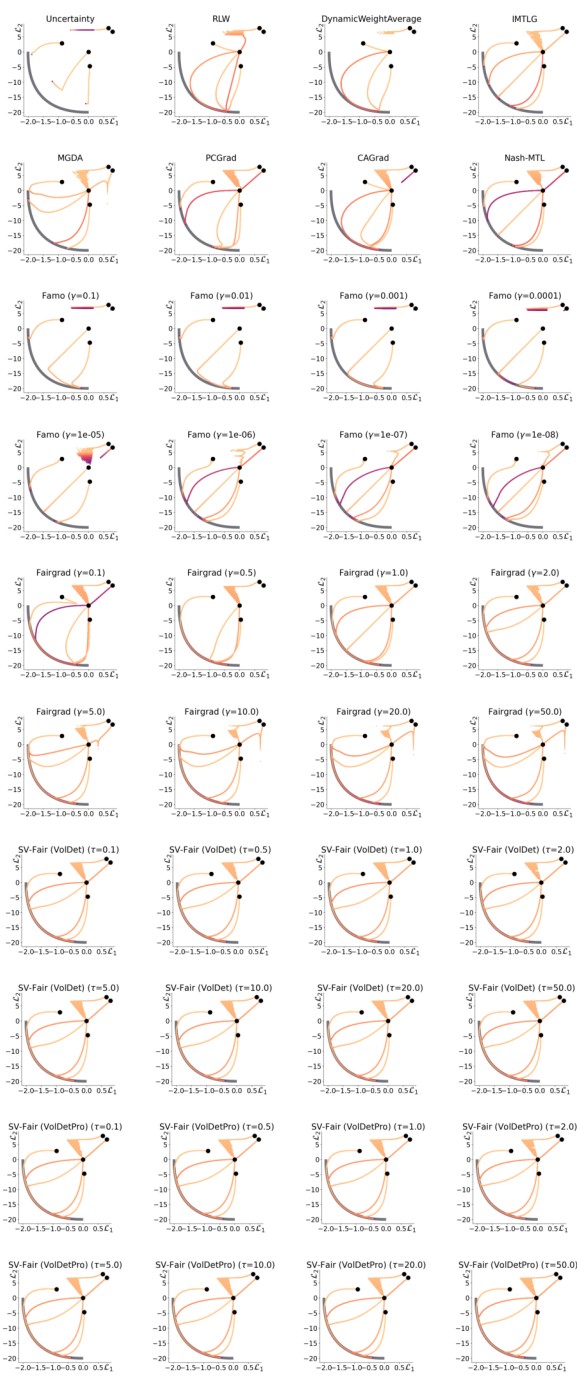

Figure 4: Comparison of MTL approaches on a challenging synthetic Two-Task A benchmark (From FAMO (Liu et al., 2023)). We visualize optimization trajectories w.r.t. objective values $(\mathcal{L}_1, \mathcal{L}_2)$. Black dots indicate starting points, and the Pareto front (see Section 3) is represented by thick gray lines.

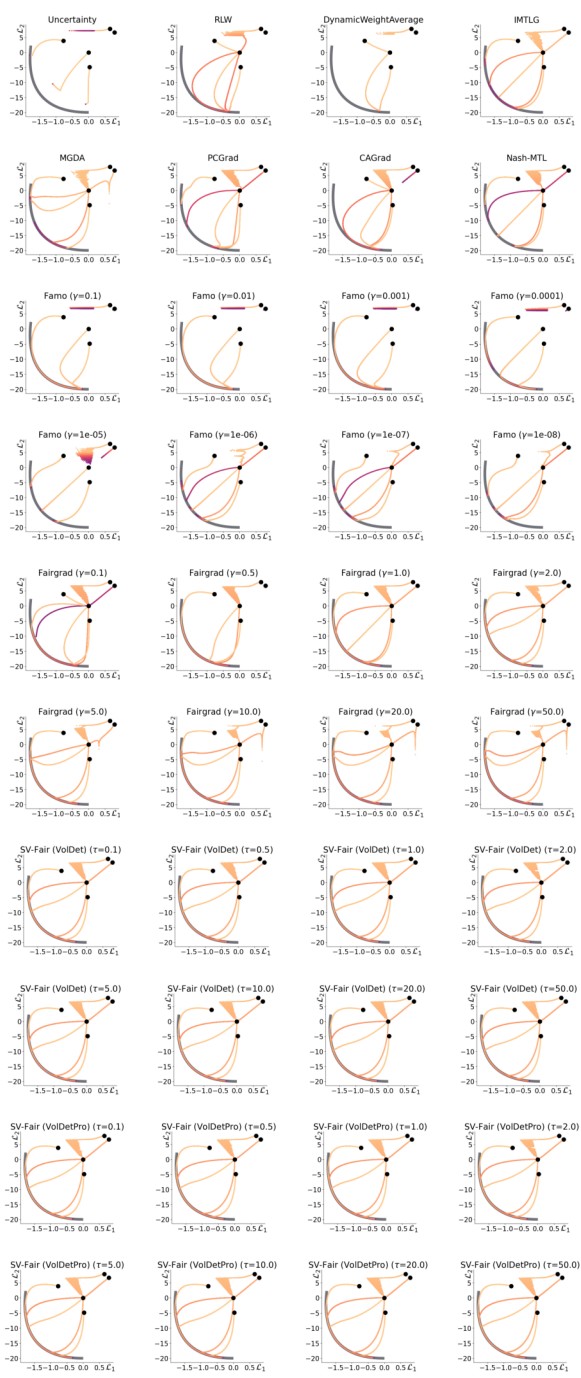

Figure 5: Comparison of MTL approaches on a challenging synthetic Two-Task B benchmark (From FairGrad (Ban & Ji, 2024)). We visualize optimization trajectories w.r.t. objective values $(\mathcal{L}_1, \mathcal{L}_2)$.

$$h_2(\theta) = \frac{((-\theta_1 - 7)^2 + 0.1(-\theta_1 - 8)^2)}{10} - 20.$$

**Two-Task C From PIVRG (Qin et al., 2025b)**

$$\mathcal{L}_1(\theta) = f_1(\theta)g_1(\theta) + f_2(\theta)h_1(\theta),$$
$$\mathcal{L}_2(\theta) = f_1(\theta)g_2(\theta) + f_2(\theta)h_2(\theta),$$

where the functions are defined as follows:

$$f_1(\theta) = \max(\tanh(0.5\theta_2), 0),$$
$$f_2(\theta) = \max(\tanh(-0.5\theta_2), 0),$$
$$g_1(\theta) = \log\left(\max(|0.5(-\theta_1 - 7) - \tanh(-\theta_2)|, 0.000005)\right) + 6,$$
$$g_2(\theta) = \log\left(\max(|0.5(-\theta_1 + 3) - \tanh(-\theta_2) + 2|, 0.000005)\right) + 6,$$
$$h_1(\theta) = \frac{((-\theta_1 + 7)^2 + 0.1(-\theta_2 - 8)^2)}{10} - 20,$$
$$h_2(\theta) = \frac{((-\theta_1 - 7)^2 + 0.1(-\theta_2 - 8)^2)}{10} - 20.$$

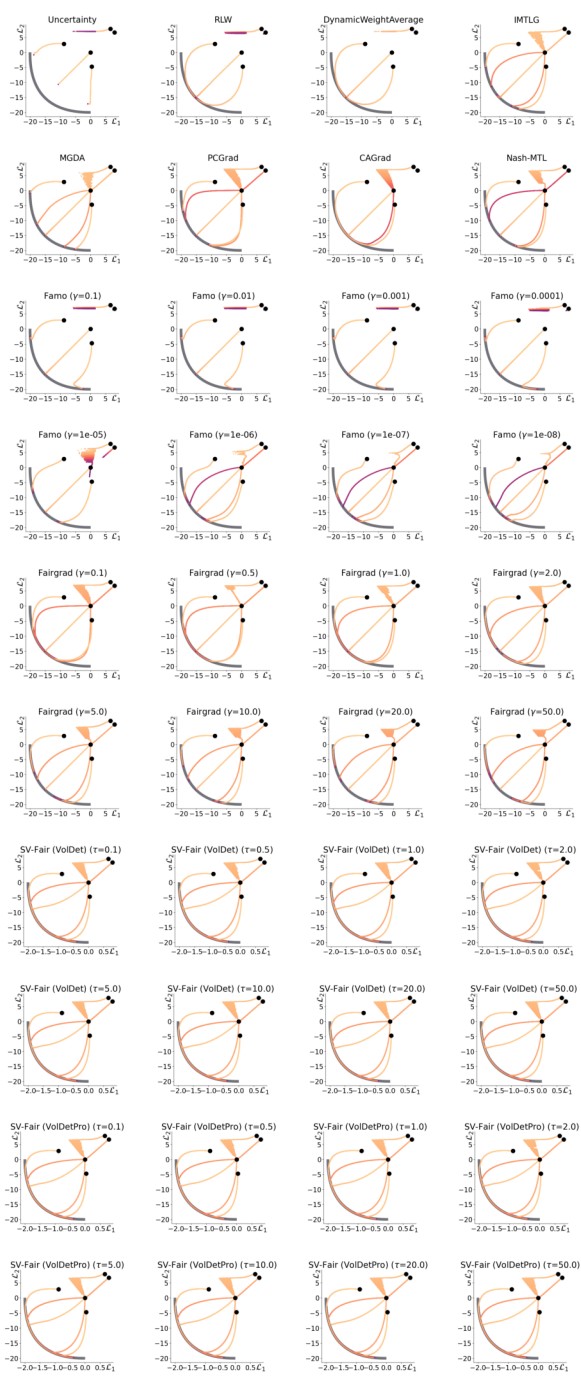

Figure 6: Comparison of MTL approaches on a challenging synthetic Two-Task C benchmark (From PIVRG (Qin et al., 2025b)). We visualize optimization trajectories w.r.t. objective values $(\mathcal{L}_1, \mathcal{L}_2)$.

## F.2 BENCHMARK DESCRIPTIONS AND TRAINING DETAILS

All our experiments were conducted on RTX 3090 and RTX 4090 GPUs. Our code is available in the supplementary material.

**NYU-v2** (Silberman et al., 2012) contains 1,449 densely annotated images focused on indoor scene understanding. It supports one pixel-wise classification task (semantic segmentation) and two pixel-wise regression tasks (depth estimation and surface normal prediction), with depth estimation covering 13 semantic categories. Following prior work, we adopt the MTAN backbone based on SegNet, integrating task-specific attention modules. Our method is trained on NYU-v2 for 200 epochs with a batch size of 2; the learning rate is set to 1e-4 for the first 100 epochs and then decayed by half for the remaining epochs.

**Cityscapes** (Cordts et al., 2016) contains 5,000 urban street view images, designed for two tasks: 7-class semantic segmentation (classification) and depth estimation (regression). We adopt the same training setup as NYU-v2, but the batch size is 8.

**CelebA** (Liu et al., 2015) is a large-scale facial attribute dataset containing over 200,000 images, each annotated with 40 attributes. It can therefore be formulated as a MTL problem with 40 image-level classification tasks. Following prior work, we adopt a CNN backbone with 9 convolutional layers and attach a separate linear output layer for each task. The model is trained using the Adam optimizer with a learning rate of 3e-4 for 15 epochs and a batch size of 256.

**Office-31** (Saenko et al., 2010) contains 4,110 images spanning three domains (Amazon, DSLR, and Webcam), designed for image classification. Following prior work, we adopt a pre-trained ResNet-18 as the shared feature extractor, with task-specific linear heads. Input images are resized to 224×224, and the model is trained for 100 epochs using the Adam optimizer with a learning rate of 1e-4 and weight decay of 1e-5.

## F.3 EXPERIMENTS RESULT DETAILS (DETAILS FOR SECTION 5.2)

We present the detailed experimental results corresponding to Section 5.2, Table 10, and Table 11.

Table 10: Results on NYU-v2 (3-task) dataset. Each experiment is repeated 3 times with different random seeds and the average is reported.

| Method | Segmentation | | Depth | | Surface Normal | | | | | MR ↓ | Δm% ↓ |
|---|---|---|---|---|---|---|---|---|---|---|---|
| | | | | | Angle ↓ | | Within $t°$ ↑ | | | | |
| | mIoU ↑ | PAcc ↑ | AErr ↓ | RErr ↓ | Mean | Median | 11.25 | 22.5 | 30 | | |
| STL | 38.30 | 63.76 | 0.6754 | 0.2780 | 25.01 | 19.21 | 30.14 | 57.20 | 69.15 | | |
| RLW | 37.17 | 63.77 | 0.5759 | 0.2410 | 28.27 | 24.18 | 22.26 | 47.05 | 60.62 | 13.33 | 7.78 |
| DWA | 39.11 | 65.31 | 0.5510 | 0.2285 | 27.61 | 23.18 | 24.17 | 50.18 | 62.39 | 10.89 | 3.57 |
| UW | 36.87 | 63.17 | 0.5446 | 0.2260 | 27.04 | 22.61 | 23.54 | 49.05 | 63.65 | 10.78 | 4.06 |
| MGDA | 30.47 | 59.90 | 0.6070 | 0.2555 | 24.88 | 19.45 | 29.18 | 56.88 | 69.36 | 8.78 | 1.38 |
| PCGrad | 38.06 | 64.64 | 0.5550 | 0.2325 | 27.41 | 22.80 | 23.86 | 49.83 | 63.14 | 11.22 | 3.97 |
| GradDrop | 39.39 | 65.12 | 0.5455 | 0.2279 | 27.48 | 22.96 | 23.38 | 49.44 | 62.87 | 10.56 | 3.58 |
| CAGrad | 39.79 | 65.49 | 0.5486 | 0.2250 | 26.31 | 21.58 | 25.61 | 52.36 | 65.58 | 8.22 | 0.19 |
| IMTL-G | 39.35 | 65.60 | 0.5426 | 0.2256 | 26.02 | 21.19 | 26.20 | 53.13 | 66.24 | 7.44 | -0.60 |
| Nash-MTL | 40.13 | 65.93 | 0.5261 | 0.2171 | 25.26 | 20.08 | 28.40 | 55.47 | 68.15 | 5.11 | -4.05 |
| FAMO | 38.88 | 64.90 | 0.5474 | 0.2194 | 25.06 | 19.57 | 29.21 | 56.61 | 68.98 | 6.67 | -4.10 |
| FairGrad | 39.74 | 66.01 | 0.5377 | 0.2236 | 24.84 | 19.60 | 29.26 | 56.58 | 69.16 | 4.78 | -4.66 |
| PIVRG | 39.90 | 65.74 | 0.5365 | 0.2243 | 24.30 | 18.80 | 30.95 | 58.26 | 70.38 | 3.56 | -6.50 |
| SVFair(VolDet) | 39.89 | **66.14** | **0.5195** | 0.2118 | **23.93** | **17.92** | **32.02** | **59.50** | **71.36** | **1.44** | **-8.81** |
| SVFair(VolDetPro) | **40.23** | 65.56 | 0.5245 | **0.2112** | 24.16 | 18.09 | 31.68 | 59.13 | 70.99 | 2.22 | -8.29 |

Table 11: Results on Cityscapes (2-task) dataset. Each experiment is repeated 3 times with different random seeds and the average is reported.

| Method | Segmentation | | Depth | | MR ↓ | Δm% ↓ |
|---|---|---|---|---|---|---|
| | mIoU ↑ | Pix Acc ↑ | Abs Err ↓ | Rel Err ↓ | | |
| STL | 74.01 | 93.16 | 0.0125 | 27.77 | | |
| RLW | 74.57 | 93.41 | 0.0158 | 47.79 | 11.00 | 24.37 |
| DWA | 75.24 | 93.52 | 0.0160 | 44.37 | 9.00 | 21.43 |
| UW | 72.02 | 92.85 | 0.0140 | 30.13 | 9.25 | 5.88 |
| MGDA | 68.84 | 91.54 | 0.0309 | 33.50 | 12.25 | 44.14 |
| PCGrad | 75.13 | 93.48 | 0.0154 | 42.07 | 9.12 | 18.21 |
| GradDrop | 75.27 | 93.53 | 0.0157 | 47.54 | 8.25 | 23.67 |
| CAGrad | 75.16 | 93.48 | 0.0141 | 37.60 | 7.88 | 11.58 |
| IMTL-G | 75.33 | 93.49 | 0.0135 | 38.41 | 6.50 | 11.04 |
| Nash-MTL | 75.41 | 93.66 | 0.0129 | 35.02 | 4.25 | 6.72 |
| FAMO | 74.54 | 93.29 | 0.0145 | 32.59 | 8.75 | 8.13 |
| FairGrad | 75.72 | **93.68** | 0.0134 | 32.25 | 3.25 | 5.12 |
| PIVRG | **75.82** | 93.65 | 0.0126 | 27.87 | **2.50** | -0.45 |
| SVFair(VolDet) | 73.42 | 92.92 | **0.0123** | 25.61 | 6.88 | -2.08 |
| SVFair(VolDetPro) | 74.16 | 93.13 | **0.0123** | **25.60** | 6.12 | **-2.40** |

### F.4 Experiments Details In PINN: Burgers Equation

To further validate the convergence and applicability of SVFair in broader multi-task scenarios, we conduct an additional experiment on physics-informed neural networks (PINNs). Compared with the vision and tabular tasks in the main paper, PINN training is typically more non-convex and exhibits stronger gradient stiffness: the gradients and convergence rates of the PDE residual, initial condition, and boundary condition losses are highly imbalanced.

**1: Experimental Setup: Multi-task PINN on the Burgers Equation**

We adopt the time-dependent Burgers equation as our test case. Let the scalar field $u(x, t)$ satisfy

$$\frac{\partial u}{\partial t} + u\frac{\partial u}{\partial x} = \nu\frac{\partial^2 u}{\partial x^2},$$

where the viscosity coefficient is $\nu = 0.01/\pi$, and the space–time domain is $x \in [-1, 1]$, $t \in [0, 1]$. The initial and boundary conditions follow the classical setting:

$$u(x, 0) = -\sin(\pi x), \quad u(-1, t) = u(1, t) = 0.$$

Within the PINN framework, we use a fully connected neural network $u_\theta(x, t)$ to approximate the solution $u(x, t)$. Following the configuration in the ConFIG (Liu et al., 2024) work, we define three loss terms for the PDE residual, boundary condition, and initial condition, denoted by $L_N, L_B, L_I$, and obtain their corresponding gradients $g_N, g_B, g_I$ via automatic differentiation. To align with the two-task formulation used by existing gradient-based methods, we merge the boundary and initial losses into a single term

$$L_{BI} = L_B + L_I,$$

which yields a two-task loss pair $(L_N, L_{BI})$ on which we apply different multi-task optimization methods. All methods share the same PINN architecture, sampling strategy, and training budget; only the *gradient aggregation / weight assignment* mechanism is changed, ensuring a fair comparison.

In SVFair, we treat $(L_N, L_{BI})$ as two "tasks", define the utility $v(S)$ of any task subset $S$ via VolDet/VolDetPro, and compute task-level Shapley values $\{\phi_i\}$ based on this utility. We then obtain per-task weights through

$$\omega_i(\tau) \propto \exp(\phi_i/\tau),$$

and use them to construct the final update direction $d = \sum_i \omega_i g_i$. Other implementation details (network depth, activation functions, number of collocation points, learning-rate schedule, etc.) follow the main paper and the ConFIG paper, and are omitted here for brevity.

**2: Baselines and Evaluation Metric**

We compare SVFair against the following representative methods: Adam (Kingma, 2014), PCGrad (Yu et al., 2020a), IMTL-G (Liu et al., 2021b), MinMax (Liu & Wang, 2021), ReLoBRaLo (Bischof & Kraus, 2025), ConFIG (Liu et al., 2024), M-ConFIG (Liu et al., 2024), ParetoMTL (Lin et al., 2019), MGDA (Sener & Koltun, 2018),

All methods are trained with the same set of random seeds, and we report the mean $\pm$ standard deviation of the test error over three runs. The evaluation metric is consistent with ConFIG (Liu et al., 2024) and measures the numerical error of the Burgers solution at test points (lower is better). The results are summarized in Table 12, where the last row corresponds to our newly added SVFair.

From Table 12, we observe that: Among all methods, SVFair achieves the smallest test error, consistently improving over Adam as well as generic MTL baselines. Notably, SVFair also outperforms ConFIG and M-ConFIG, which were specifically designed for PINNs and have been shown to exhibit strong convergence and numerical performance on Burgers in the original work. Under the same PINN architecture and training budget, using VolDet with Shapley-based weights to construct the SVFair update direction further alleviates the gradient conflict between $L_N$ and $L_{BI}$ and leads to a better multi-objective trade-off.

Overall, this additional experiment shows that SVFair is not merely "tuned to work" on standard vision MTL benchmarks, but also demonstrates stable convergence behavior and performance gains on highly stiff, strongly conflicting PINN training problems.

Table 12: Results on Burgers.

| Method | ↓ |
|---|---|
| Adam | 1.484 ± 0.061 |
| PCGrad | 1.344 ± 0.019 |
| IMTL-G | 1.339 ± 0.024 |
| MinMax | 1.889 ± 0.143 |
| ReLoBRaLo | 1.419 ± 0.053 |
| ConFIG | 1.308 ± 0.008 |
| M-ConFIG | 1.277 ± 0.035 |
| ParetoMTL | 1.603 ± 0.030 |
| MGDA | 1.706 ± 0.121 |
| **SVFair** | **1.271 ± 0.055** |

# G   DETAILS ABOUT THE MC SHAPLEY ESTIMATION

We provide additional details of the computation of the Shapley value discussed in Section 5.4. The analysis details of computational complexity can be found in Appendix B.4. Appendix B.4 states that, as the number of tasks $N$ grows, although VolDet (VolDetPro) can greatly reduce training costs, the overhead of computing determinants still increases exponentially—each determinant evaluation has a time complexity of $O(N^3)$. Therefore, we introduce Monte Carlo sampling to reduce the overall computational cost from exponential to an acceptable linear scale. The procedure of Monte Carlo sampling is summarized in Algorithm 3.

---

**Algorithm 3** Monte Carlo Estimation of Shapley Values

---

1: **Input:** number of players $N$, utility function $v(\cdot)$, sample size $K$
2: **Output:** estimated Shapley values $\{\phi_i\}_{i=1}^{N}$

3: *Initialize:* $\phi[i] \leftarrow 0$ for all $i = 1, \ldots, N$
4: **for** $t = 1$ **to** $K$ **do**
5: $\quad \pi \leftarrow \text{RandomPermutation}(\{1, \ldots, N\})$
6: $\quad S \leftarrow \emptyset$
7: $\quad$ **for** $j = 1$ **to** $N$ **do**
8: $\quad\quad i \leftarrow \pi[j]$
9: $\quad\quad \Delta \leftarrow v(S \cup \{i\}) - v(S)$
10: $\quad\quad \phi[i] \leftarrow \phi[i] + \Delta$
11: $\quad\quad S \leftarrow S \cup \{i\}$
12: $\quad$ **end for**
13: **end for**
14: **for** $i = 1$ **to** $N$ **do**
15: $\quad \phi[i] \leftarrow \phi[i]/K$
16: **end for**
17: **Return** $\{\phi_i\}_{i=1}^{N}$

---

## G.1   DETAILS ABOUT THE EFFECTIVENESS, CONVERGENCE, AND COST–PERFORMANCE TRADE-OFF OF MC SHAPLEY ESTIMATION

**1. Analysis of the Convergence of Monte Carlo Estimation for Shapley Values**   For any task subset $S$, the VolDet utility satisfies $0 \le v(S) = \det(\mathcal{M}_S) \le 1$. Hence each marginal contribution in the permutation-based MC estimator, $\Delta_i(S) = v(S \cup \{i\}) - v(S)$, is bounded in an interval of width $B \le 2$. Together with uniform sampling over permutations, this yields an **unbiased** and **bounded** MC estimator of the Shapley values. In Appendix G.2 we derive the following Hoeffding-

style concentration bound:

$$\Pr\left(\left\|\widehat{\phi} - \phi\right\|_\infty > \varepsilon\right) \;\leq\; 2N\exp\!\left(-\frac{2K\varepsilon^2}{B^2}\right). \tag{18}$$

This inequality shows that the deviation $\left\|\widehat{\phi} - \phi\right\|_\infty$ shrinks at rate $\mathcal{O}(1/\sqrt{K})$ as the number of MC samples $K$ increases. We provide more detail in Appendix G.2.

**2. Empirical results validate the effectiveness and convergence of MC Shapley Value**  Table 13 in Appendix G.1 reports an experiment to empirically validate the effectiveness and convergence of MC Shapley. We collect gradient Gram matrices from Office-31 ($N = 3$) and from CelebA ($N = 40$). For CelebA, we randomly sample task subsets of size $N_{\text{sub}} \in \{10, 15, 20\}$ and extract the corresponding VolDet submatrices. For each matrix, we compute both exact Shapley values and their MC approximation using our permutation-based estimator, for MC samples $K \in \{100, 250, 500, 1000, 2000\}$. We then measure the MSE between exact and MC Shapley values and average over 10 matrices.

Table 13: Results on Office-31 and CelebA. Mean squared error between exact and Monte Carlo Shapley values under the VolDet utility on Office-31 ($N = 3$) and CelebA subsets ($N_{\text{sub}} = 10, 15, 20$) for different numbers of MC samples $K$. Results are averaged over 10 gradient Gram matrices for each setting.

| Sample Size $K$ | Office-31($N = 3$) | CelebA($N = 10$) | CelebA($N = 15$) | CelebA($N = 20$) |
|---|---|---|---|---|
| $K = 100$ | $9.18 \times 10^{-5}$ | $1.96 \times 10^{-5}$ | $5.95 \times 10^{-6}$ | $3.19 \times 10^{-6}$ |
| $K = 250$ | $1.05 \times 10^{-4}$ | $2.13 \times 10^{-5}$ | $3.07 \times 10^{-6}$ | $1.48 \times 10^{-6}$ |
| $K = 500$ | $5.84 \times 10^{-5}$ | $1.38 \times 10^{-5}$ | $2.49 \times 10^{-6}$ | $1.77 \times 10^{-6}$ |
| $K = 1000$ | $0.671 \times 10^{-5}$ | $1.28 \times 10^{-5}$ | $2.26 \times 10^{-6}$ | $1.33 \times 10^{-6}$ |
| $K = 2000$ | $5.36 \times 10^{-5}$ | $1.25 \times 10^{-5}$ | $1.87 \times 10^{-6}$ | $1.28 \times 10^{-6}$ |

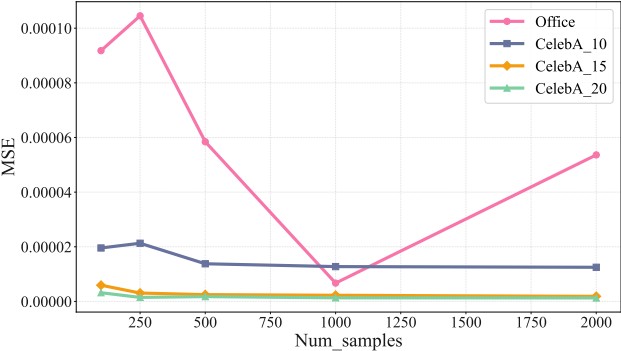

Figure 7: Visualization of Table 13.

We make two observations from the Table 13:

A: for all large-task settings with task number $N_{\text{sub}} \geq 10$, the MSE rapidly decreases as $K$ grows and essentially plateaus once $K \geq 1000$. This confirms that our choice of $K = 1000$ in the CelebA experiments is well justified: further increasing $K$ to 2000 yields almost no reduction in error, while nearly doubling the Shapley-related computation.

B: for the small-task case (Office-31, $N = 3$), computing exact Shapley values is already cheap and MC is not needed in practice. The slight non-monotonic "rebound" at $K = 2000$ is a typical Monte Carlo fluctuation at the $10^{-5}$ level rather than a systematic issue, and it has no impact on our method because we never rely on MC in such small-$N$ settings. Overall, these results directly support the *effectiveness* and *convergence* of the MC Shapley estimator.

**3. Empirical results validate the MC samples $K$ can improve the final performance**   Table 14 in Appendix G.1 reports an experiment to empirically validate that increasing the MC samples $K$ can improve the final performance. We run full training experiments on Office-31 and CelebA with different MC samples $K$. The results are summarized in the following table:

Table 14: Results on Office-31 (3-task) and CelebA (40-task) datasets. End-to-end results of SVFair with different Monte Carlo sample sizes $K$ on Office-31 and CelebA benchmarks.

| Sample Size $K$ | Office-31 | | | | CelebA |
|---|---|---|---|---|---|
| | Amazon ↑ | DSLR ↑ | Webcam ↑ | $\Delta m\% \downarrow$ | $\Delta m\% \downarrow$ |
| SVFair ($K = 100$) | 85.30 | 99.18 | 98.87 | -1.43 | 7.12 |
| SVFair ($K = 250$) | 84.84 | 99.45 | **99.07** | -1.42 | 3.83 |
| SVFair ($K = 500$) | 85.58 | 99.45 | 98.52 | -1.51 | 2.12 |
| SVFair ($K = 1000$) | 85.41 | **99.73** | 98.70 | -1.60 | 0.11 |
| SVFair ($K = 2000$) | 85.64 | **99.73** | 98.33 | -1.56 | **0.06** |
| SVFair (Precise) | **86.04** | 99.45 | 98.33 | **-1.62** | - |

We make observation from the Table 14:

**CelebA** Increasing $K$ from 100 to 1000 dramatically improves fairness, with $\Delta m\%$ dropping from 7.12 to 0.11. Further increasing $K$ to 2000 yields only a very small improvement ($0.11 \rightarrow 0.06$), while almost doubling the Shapley-related computation. This shows that $K = 1000$ already lies on a cost-effective performance plateau, consistent with the MSE results in point 2 (i.e. 2.Empirical results validate the effectiveness and convergence of MC Shapley value). The Office-31 result also exhibits a similar experimental phenomenon. Overall, these results directly support that MC samples $K$ can improve the final performance.

In summary, our analysis and experiments jointly demonstrate that the permutation-based MC Shapley estimator is theoretically well-founded, empirically accurate, and practically efficient.

### G.2   MONTE CARLO ESTIMATION OF SHAPLEY VALUES FOR VOLDET

In this section we show that the Monte Carlo estimator (Rozemberczki et al., 2022; Mitchell et al., 2022) used in SVFair provides an unbiased estimate of the task-wise Shapley Values and enjoys a standard Hoeffding-type concentration bound when the utility is defined by VolDet. The procedure of Monte Carlo sampling is summarized in Algorithm 2.

**Setup:** Let $\mathcal{T} = \{1, \ldots, N\}$ denote the task set and $v(S)$ be the utility of any subset $S \subseteq \mathcal{T}$. For VolDet, $v(S)$ is defined as the determinant of the Gram matrix constructed from the $\ell_2$-normalized gradients of tasks in $S$, $v(S) = \det(\mathcal{M}_S)$. By construction $M_S$ is positive semidefinite and has unit diagonal. The Shapley value of task $i$ is

$$\phi_i = \sum_{S \subseteq N \setminus \{i\}} \frac{|S|!(N - |S| - 1)!}{N!} \left[ v(S \cup \{i\}) - v(S) \right]. \tag{19}$$

For convenience in writing, we rewrite Eq.19 as the following Eq.20:

$$\phi_i = \frac{1}{N!} \sum_{\pi \in S_N} \left[ v\big(P_i(\pi) \cup \{i\}\big) - v\big(P_i(\pi)\big) \right], \tag{20}$$

where $S_N$ is the set of all permutations of $\mathcal{T}$ and $P_i(\pi)$ denotes the set of tasks that appear before $i$ under the permutation $\pi$.

**Definition 12 (Boundedness of VolDet utility)** *For any subset $S \subseteq \mathcal{T}$, the VolDet utility satisfies:*

$$0 \leq v(S) = \det(\mathcal{M}_S) \leq 1. \tag{21}$$

**Proof:** Since $\mathcal{M}_S$ is a Gram matrix of $\ell_2$-normalized vectors, it is positive semidefinite with diagonal entries $(\mathcal{M}_S)_{ii} = 1$. By Hadamard's inequality, the determinant of a positive semidefinite matrix is upper bounded by the product of its diagonal entries, hence $\det(\mathcal{M}_S) \leq \prod_{i \in S} (\mathcal{M}_S)_{ii} = 1$. Positive semidefiniteness also implies $\det(\mathcal{M}_S) \geq 0$.

As a direct corollary, for any task $i$ and subset $S \subseteq \mathcal{T} \setminus \{i\}$, we have:

$$v\big(S \cup \{i\}\big),\, v(S) \in [0,1] \quad \Rightarrow \quad \Delta_i(S) := v\big(S \cup \{i\}\big) - v(S) \in [-1,1]. \tag{22}$$

**Permutation-based MC estimator.** We approximate Eq.20 by averaging marginal contributions over $K$ i.i.d. random permutations. Specifically, let $\pi^{(1)}, \ldots, \pi^{(K)}$ be i.i.d. samples from the uniform distribution over $S_N$, and define

$$\Delta_i^{(t)} := v\big(P_i(\pi^{(t)}) \cup \{i\}\big) - v\big(P_i(\pi^{(t)})\big), \quad t = 1, \ldots, K, \tag{23}$$

and the MC estimator

$$\widehat{\phi}_i := \frac{1}{K} \sum_{t=1}^{K} \Delta_i^{(t)}. \tag{24}$$

**Proposition 3 (Unbiasedness)** *For each task $i$, the MC estimator $\widehat{\phi}_i$ is an unbiased estimator of the Shapley value $\phi_i$, i.e.,*

$$\mathbb{E}\, \widehat{\phi}_i \;=\; \phi_i. \tag{25}$$

**Proof:** The permutations $\pi^{(t)}$ are i.i.d. and uniformly distributed over $S_N$, and $\Delta_i^{(t)}$ is a measurable function of $\pi^{(t)}$. Therefore,

$$\mathbb{E}\, \widehat{\phi}_i = \frac{1}{K} \sum_{t=1}^{K} \mathbb{E}\, \Delta_i^{(t)} = \mathbb{E}_{\pi \sim \mathrm{Unif}(S_N)}\Big[ v\big(P_i(\pi) \cup \{i\}\big) - v\big(P_i(\pi)\big)\Big]. \tag{26}$$

By the permutation definition of the Shapley value in Eq.20, the right-hand side equals $\phi_i$, which proves the claim.

**Concentration bound.** From Definition 12 we know that $v(S) \in [0,1]$, hence each marginal contribution satisfies $\Delta_i^{(t)} \in [-1,1]$ as in Eq.22. Let $B$ denote the width of this bounded interval,

$$B := \max_{i,t} \big( \sup \Delta_i^{(t)} - \inf \Delta_i^{(t)} \big) \le 2. \tag{27}$$

**Theorem 4 (Hoeffding-style concentration for VolDet)** *Let $\widehat{\phi} = (\widehat{\phi}_1, \ldots, \widehat{\phi}_N)$ be the MC estimator and $\phi = (\phi_1, \ldots, \phi_N)$ be the true Shapley values. Assume the marginal contributions are bounded with width $B$ as above. Then for any $\varepsilon > 0$,*

$$\Pr\left( \|\widehat{\phi} - \phi\|_\infty > \varepsilon \right) \;\le\; 2N \exp\!\Big( -\frac{2K\varepsilon^2}{B^2} \Big). \tag{28}$$

*In particular, when $v(\cdot)$ is VolDet, we can take $B \le 2$.*

**Proof:** Fix a task $i \in \mathcal{T}$. The random variables $\Delta_i^{(1)}, \ldots, \Delta_i^{(K)}$ are independent and bounded in an interval of width at most $B$. By Hoeffding's inequality,

$$\Pr\left( |\widehat{\phi}_i - \phi_i| > \varepsilon \right) \;\le\; 2 \exp\!\Big( -\frac{2K\varepsilon^2}{B^2} \Big). \tag{29}$$

Applying the union bound over all $i = 1, \ldots, N$ yields

$$\Pr\left( \|\widehat{\phi} - \phi\|_\infty > \varepsilon \right) \;\le\; \sum_{i=1}^{N} \Pr\left( |\widehat{\phi}_i - \phi_i| > \varepsilon \right) \;\le\; 2N \exp\!\Big( -\frac{2K\varepsilon^2}{B^2} \Big), \tag{30}$$

which is exactly Eq.28.

The bound in Theorem 4 shows that the deviation $\|\widehat{\phi} - \phi\|_\infty$ shrinks at rate $\mathcal{O}(1/\sqrt{K})$ as the number of MC samples $K$ increases, and provides a conservative lower bound on the number of samples required to achieve a desired accuracy level.

# H   DETAILS FOR CHOICE OF $\tau$. (DETAILS FOR SECTION 5.4)

## H.1   THEORETICAL PERSPECTIVE TO GUIDE THE CHOICE OF $\tau$.

In SVFair, $\phi_i$ denote the Shapley value. The task weights are computed as

$$\omega_i(\tau) = \frac{\exp(\phi_i/\tau)}{\sum_j \exp(\phi_j/\tau)}. \tag{31}$$

It can be shown (a formal derivation will be given in the Appendix H.2) that this softmax mapping is exactly the solution of the following *entropy-regularized* optimization problem:

$$\omega(\tau) = \arg\max_{\omega \in \Delta} \Big( \langle \omega, \phi \rangle + \tau\, H(\omega) \Big), \tag{32}$$

where $\Delta$ is the probability simplex and entropy $H(\omega) = -\sum_i \omega_i \log \omega_i$. $\tau$ plays the role of a regularization coefficient: the linear term $\langle \omega, \phi \rangle$ encourages the weights to follow the Shapley values (i.e., the "ideal fairness compensation"), while the term $\tau H(\omega)$ encourages a high-entropy weight distribution, preventing over-concentration on a few tasks and improving training stability.

From this formulation, we obtain two key monotonicity properties:

1: If $\tau_1 < \tau_2$, then

$$H(\omega(\tau_1)) \le H(\omega(\tau_2)), \qquad \max_i \omega_i(\tau_1) \ge \max_i \omega_i(\tau_2), \tag{33}$$

i.e., smaller $\tau$ yields sharper, more concentrated weights, whereas larger $\tau$ makes the weights closer to uniform.

2: For any pair of tasks $(i, j)$,

$$\log \frac{\omega_i(\tau)}{\omega_j(\tau)} = \frac{\phi_i - \phi_j}{\tau}, \tag{34}$$

so, for a given gap $\phi_i - \phi_j$, $\tau$ precisely controls how strongly this gap is amplified in the weight space.

Based on this, we propose a theoretically motivated guideline for choosing $\tau$. In a short warm-up phase, we estimate a representative set of Shapley values $\{\phi_i\}$ and define $\kappa(\tau) = T \cdot \max_i \omega_i(\tau)$ as a measure of the amplification factor of the largest task weight. In practice, we can pre-specify an acceptable upper bound on concentration, or an entropy lower bound $H_{\min}$. We then choose the *smallest* $\tau$ that satisfies $\kappa(\tau) \le \kappa_{\max}$  or  $H(\omega(\tau)) \ge H_{\min}$, so that we maximize the Shapley-driven fairness adjustment while maintaining sufficient stability.

Intuitively, when Shapley values exhibit large gaps and gradient conflicts are sharp, a too small $\tau$ leads to a very large $\kappa(\tau)$ and the training is dominated by a few tasks; in this case, a relatively larger $\tau$ is preferable to keep entropy at a reasonable level. When Shapley values are closer to each other and tasks are more balanced, one can safely use a smaller $\tau$ to magnify mild unfairness and improve the performance of disadvantaged tasks.

## H.2   SOFTMAX AS THE OPTIMAL SOLUTION OF ENTROPY-REGULARIZED LINEAR UTILITY.

**Proposition 4 (Softmax as the optimal solution of entropy-regularized linear utility)** *Let* $\phi = (\phi_1, \ldots, \phi_T) \in \mathbb{R}^T$ *be a given vector of Shapley values, and let*

$$\Delta = \left\{ \omega \in \mathbb{R}^T \,\middle|\, \omega_i \ge 0,\ \sum_{i=1}^T \omega_i = 1 \right\}$$

*be the probability simplex. For any temperature parameter $\tau > 0$, consider the optimization problem*

$$\max_{\omega \in \Delta} F(\omega) \quad where \quad F(\omega) = \langle \omega, \phi \rangle + \tau H(\omega),$$

*with Shannon entropy*

$$H(\omega) = -\sum_{i=1}^T \omega_i \log \omega_i.$$

*Then this problem admits a unique maximizer, and the optimal solution is exactly the softmax*

$$\omega_i^\star(\tau) = \frac{\exp(\phi_i/\tau)}{\sum_{j=1}^T \exp(\phi_j/\tau)}, \quad i = 1, \dots, T.$$

**proof:**

**1. Concavity and uniqueness.** Note that

$$H(\omega) = -\sum_{i=1}^T \omega_i \log \omega_i$$

is strictly concave on the relative interior of $\Delta$, and the linear term $\langle \omega, \phi \rangle$ is concave as well. Hence

$$F(\omega) = \langle \omega, \phi \rangle + \tau H(\omega)$$

is strictly concave on $\Delta$. Since $\Delta$ is compact and convex, a strictly concave function on $\Delta$ has at most one global maximizer, and the maximizer exists and is unique. We now derive its closed-form expression.

**2. Lagrangian and KKT conditions.** Consider the constrained optimization problem

$$\max_{\omega \in \mathbb{R}^T} \sum_{i=1}^T \Big( \omega_i \phi_i - \tau \omega_i \log \omega_i \Big) \quad \text{s.t.} \quad \sum_{i=1}^T \omega_i = 1, \ \omega_i \geq 0.$$

We form the Lagrangian (let $\lambda$ be the multiplier for the equality constraint; the non-negativity constraints will be automatically satisfied at the interior optimum):

$$\mathcal{L}(\omega, \lambda) = \sum_{i=1}^T \Big( \omega_i \phi_i - \tau \omega_i \log \omega_i \Big) + \lambda \Big( 1 - \sum_{i=1}^T \omega_i \Big).$$

Taking the partial derivative with respect to each $\omega_i$ and setting it to zero yields the first-order optimality condition:

$$\frac{\partial \mathcal{L}}{\partial \omega_i} = \phi_i - \tau(1 + \log \omega_i) - \lambda = 0, \quad i = 1, \dots, T.$$

Equivalently,

$$\phi_i - \lambda - \tau = \tau \log \omega_i,$$

so

$$\log \omega_i = \frac{\phi_i - \lambda - \tau}{\tau} \quad \implies \quad \omega_i = \exp\Big(\frac{\phi_i}{\tau}\Big) \cdot \exp\Big(-\frac{\lambda + \tau}{\tau}\Big).$$

Define

$$C = \exp\Big(-\frac{\lambda + \tau}{\tau}\Big),$$

then we can write

$$\omega_i = C \exp\Big(\frac{\phi_i}{\tau}\Big), \quad i = 1, \dots, T.$$

**3. Normalization and softmax form.** Using the normalization constraint $\sum_{i=1}^T \omega_i = 1$, we obtain

$$1 = \sum_{i=1}^T \omega_i = C \sum_{i=1}^T \exp\Big(\frac{\phi_i}{\tau}\Big),$$

which implies

$$C = \Big( \sum_{j=1}^T \exp\Big(\frac{\phi_j}{\tau}\Big) \Big)^{-1}.$$

Substituting this back into the expression for $\omega_i$, we get

$$\omega_i^\star(\tau) = \frac{\exp(\phi_i/\tau)}{\sum_{j=1}^{T} \exp(\phi_j/\tau)}, \quad i = 1, \ldots, T.$$

**4. Non-negativity and feasibility.** Since the exponential function is strictly positive, for any $\tau > 0$ and any $\phi_i \in \mathbb{R}$ we have $\omega_i^\star(\tau) > 0$ and $\sum_i \omega_i^\star(\tau) = 1$. Hence $\omega^\star(\tau) \in \Delta$, and by Step 1 it must be the unique global maximizer.

Therefore, the unique optimal solution of the entropy-regularized problem coincides with the softmax weights.

### H.3 Experiments about the Choice of $\tau$.

The Eq. 7 in SVFair relies on a temperature parameter $\tau$:

$$\omega_i = \frac{\exp(\phi_i/\tau)}{\sum_{j=1}^{N} \exp(\phi_j/\tau)}$$

We provide additional results of different $\tau$ discussed in Section 5.4 in Table 4. To evaluate the effect of $\tau$ on the mean rank and overall performance drop ($\Delta m\%$), we conduct a small-scale hyperparameter sweep with $\tau \in \{0.5, 1.0, 2.0, 5.0\}$ on the Cityscapes, Office-31, and CelebA benchmarks. The detailed results are Table 15 and Table 16. The result shows that different values of $\tau$ have varying effects on different benchmarks. It can be observed that as $\tau$ changes, the algorithm prioritizes certain tasks, leading to better average rankings and performance improvements.

Table 15: Results on Cityscapes (2-task) dataset. Each experiment is repeated 3 times with different random seeds and the average is reported.

| Method | Segmentation | | Depth | | MR ↓ | $\Delta m\%$ ↓ |
|---|---|---|---|---|---|---|
| | mIoU ↑ | Pix Acc ↑ | Abs Err ↓ | Rel Err ↓ | | |
| SVFair(VolDet) ($\tau$ =0.5) | 73.75 | 93.07 | 0.0126 | 26.01 | 7.88 | -1.27 |
| SVFair(VolDet) ($\tau$ =1.0) | 74.50 | 92.90 | 0.0127 | 27.54 | 8.75 | 0.10 |
| SVFair(VolDet) ($\tau$ =2.0) | **74.76** | **93.36** | 0.0125 | 27.24 | **6.00** | -0.78 |
| SVFair(VolDet) ($\tau$ =5.0) | 74.27 | 93.19 | **0.0121** | 26.37 | 7.00 | **-2.16** |
| SVFair(VolDetPro) ($\tau$ =0.5) | 74.22 | 93.11 | 0.0126 | 26.93 | 11.12 | -0.61 |
| SVFair(VolDetPro) ($\tau$ =1.0) | 73.90 | 93.11 | 0.0127 | **25.52** | 11.00 | -1.57 |
| SVFair(VolDetPro) ($\tau$ =2.0) | 74.04 | 93.05 | 0.0125 | 26.32 | 11.12 | -1.29 |
| SVFair(VolDetPro) ($\tau$ =5.0) | 73.95 | 92.99 | 0.0124 | 26.26 | 11.00 | -1.49 |

Table 16: Results on Office-31 (3-task) and CelebA (40-task) datasets. Each experiment is repeated 3 times with different random seeds and the average is reported.

| Method | Office-31 | | | | | CelebA | |
|---|---|---|---|---|---|---|---|
| | Amazon ↑ | DSLR ↑ | Webcam ↑ | MR ↓ | $\Delta m\%$ ↓ | MR ↓ | $\Delta m\%$ ↓ |
| SVFair(VolDet) ($\tau$ =0.5) | 85.13 | **99.19** | 98.70 | 4.83 | -1.30 | 7.50 | 1.20 |
| SVFair(VolDet) ($\tau$ =1.0) | 85.70 | 98.91 | **98.89** | **3.67** | **-1.49** | 7.68 | 0.22 |
| SVFair(VolDet) ($\tau$ =2.0) | **85.99** | 99.18 | 98.15 | 4.00 | -1.44 | **7.00** | **-0.10** |
| SVFair(VolDet) ($\tau$ =5.0) | 84.50 | 98.91 | 98.71 | 7.00 | -0.97 | 7.82 | 0.23 |
| SVFair(VolDetPro) ($\tau$ =0.5) | 85.64 | 98.91 | 98.70 | 5.17 | -1.41 | 12.62 | 6.34 |
| SVFair(VolDetPro) ($\tau$ =1.0) | 84.96 | 99.18 | **98.89** | 5.00 | -1.30 | 10.60 | 3.31 |
| SVFair(VolDetPro) ($\tau$ =2.0) | 85.53 | 98.09 | 99.26 | 4.33 | -1.27 | 8.52 | 1.10 |
| SVFair(VolDetPro) ($\tau$ =5.0) | 85.35 | 99.18 | **98.89** | 4.00 | -1.45 | 8.85 | 0.27 |

# I    EXPERIMENTS DETAILS FOR VOLDET VS VOLDETPRO (DETAILS FOR SECTION 5.5)

**Experiments setting.**    We run SVFair with the different temperature values shown in the legend ($\tau = 5.0$ for Office-31 and Cityscapes, $\tau = 2.0$ for NYU-V2, $\tau = 1.0$ for CelebA). At eight evenly spaced checkpoints from $33.3\%$ to $100\%$ training progress, we compute Proportion of Negative Cosine Pairs (PNCS) from the current task gradients by normalizing per-task gradients $\{\hat{g}_i\}$ and normalized gradient matrix $\hat{G}$, forming all pairwise cosines, and reporting the fraction with cosine $< 0$. Curves show the mean PNCS over runs; the dashed line marks the 0.25 threshold used in our analysis. Office-31 stays mostly above 0.25, whereas NYU-V2, CelebA and Cityscapes remain below.

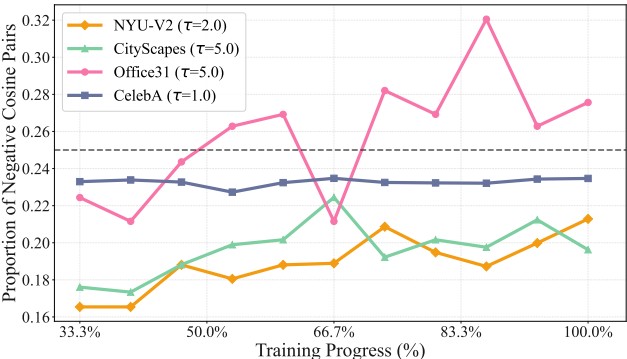

Figure 8: Proportion of Negative Cosine Pairs (PNCS) during training.

**Metric mechanism.**    VolDetPro extends VolDet by adding a penalty on the *number* and *magnitude* of negative–cosine pairs. The penalty activates only when obtuse relations appear, making the metric more responsive to directional opposition and allowing SVFair to expose conflicts earlier and adjust Shapley weights.

**When VolDetPro is better.**    When tasks exhibit pronounced negative correlations, the volume term alone cannot distinguish configurations with the same volume but more obtuse angles. The penalty in VolDetPro increases monotonically with both the count and the absolute sum of negative cosines, thereby increasing sensitivity to true conflicts. During weight allocation, SVFair moves weight away from opposing tasks, reduces tug–of–war, and improves the weakest task. Hence VolDetPro is more likely to achieve lower MR and better $\Delta m\%$ in such regimes.

**When VolDet is better.**    When overall correlations are positive or conflicts are weak, a sign–sensitive penalty may induce slight over–correction and prematurely suppress useful directional diversity. In this case the volume–only VolDet is more neutral and tends to yield steadier aggregate gains.

**Experiments Diagnostic.**    We compute at multiple training stages the PNCS statistic, see Fig. A1. On Office–31, PNCS stays above 0.25 for most of the later stages, indicating that oppositional relations dominate. Using VolDetPro in this regime amplifies true conflict signals and yields better MR and $\Delta m\%$. On NYU-V2, CelebA and Cityscapes, PNCS remains below 0.25 throughout training, which suggests mainly positive or weakly conflicting relations; the volume term already captures the global geometry well, and VolDet is more stable and often better. This observation is fully consistent with the tables in the main text and provides a practical diagnostic for selecting the metric.

## J  THE LINK BETWEEN SHAPLEY VALUE WEIGHTING AND MTL OPTIMIZATION

SVFair builds on the same fairness-aware aggregation principle introduced in Eq. 1 ($\alpha$-fair utility maximization) by FairGrad (Ban & Ji, 2024) and PIVRG (Qin et al., 2025b), which generalizes network $\alpha$-fairness to gradient combination. In that framework, each task's gradient is treated as a "utility," and the algorithm seeks the descent direction that maximizes *global fairness* subject to remaining a valid descent for all tasks.

In SVFair, we replace the original heuristic weights $\omega_i$ in Eq. 1 with Shapley values $\phi_i$ computed from our VolDet (i.e. Definition 5) conflict metric, the determinant of the gradient Gram matrix. VolDet measures the multi-dimensional volume spanned by normalized gradients, flagging both overt and subtle conflicts that pairwise cosines can miss. A task whose gradient contributes most to inflating this volume (i.e., injects new conflicting directions) receives a higher $\phi_i$, whereas gradients that align well receive lower weight.

Beyond VolDet, we introduce a sign-sensitive extension, VolDetPro (i.e. Definition 6), which preserves the volume term while explicitly accounting for negative cosine relations. When $\mathcal{I}(S) = \varnothing$, $v(S)$ reduces to VolDet; when obtuse alignments exist, the second term grows with both the number and the magnitude of negative cosines, correcting the sign-agnosticity of VolDet and yielding fairer attributions to under-represented or opposed tasks. We compute Shapley values $\phi_i$ on $v(\cdot)$, so that tasks that newly introduce strong disagreement receive larger $\phi_i$, while already represented directions receive smaller ones.

Solving Eq. 7 with $\omega_i \propto \exp(\phi_i/\tau)$ yields a Shapley-balanced update: under-represented or heavily opposed tasks pull $d$ toward themselves, while naturally aligned tasks have less influence. Under Assumption 1 (Appendix E.1), iterating this update converges to a Pareto-stationary point that is feasible for all tasks and realizes a more balanced compromise on the Pareto front, rather than collapsing to an extreme corner.

From a cooperative-game perspective, Shapley allocations for convex games lie in the core, meaning no subset of players (tasks) can improve their outcome by deviating. Analogously, our Shapley-weighted gradient direction is stable: no group of tasks can find a better joint descent direction that helps themselves without harming others. This direct parallel to Pareto efficiency and coalition stability, together with the sign-aware design of VolDetPro, provides a clear theoretical justification for why Shapley–VolDet/VolDetPro weighting leads to more balanced and robust Pareto solutions, not merely as an empirical artifact but grounded in fair-division principles (Ban & Ji, 2024; Qin et al., 2025b).

# K  POSITIONING OF SVFAIR IN THE LITERATURE FROM GEOMETRIC AND GAME-THEORETIC PERSPECTIVES

This appendix further clarifies where SVFair lies in the landscape of existing MTL optimization methods. We focus on two complementary viewpoints — the *gradient geometry perspective* and the *cooperative game-theoretic perspective* — and highlight the conceptual advances of our work beyond prior approaches.

## K.1  GEOMETRIC PERSPECTIVE

From a geometric viewpoint, mainstream MTL optimization methods are typically designed around the *relative configuration of task gradients in vector space*. Early approaches such as the MGDA family search, at each step, for a direction in the convex hull of task gradients that minimizes the norm or satisfies certain constraints, thereby obtaining an update direction lying on (or approximating) the Pareto front (Sener & Koltun, 2018; Zhang et al., 2025). PCGrad (Yu et al., 2020a) instead relies on pairwise cosine similarities between task gradients, projecting out components that are mutually conflicting, so as to mitigate local conflicts. Subsequent methods such as FAMO (Liu et al., 2023) similarly build on pairwise angles, gradient-norm ratios, or simple geometric regularizers to measure and alleviate task interference.

A common feature of this line of work is that the geometric quantities are mostly restricted to *pairwise relationships between tasks* (e.g., cosine similarity or angle deviations). Algorithmically, the focus is on locally modifying gradients at the current iteration, without explicitly constructing a set function $v(S)$ defined over *all* task subsets $S$. As a consequence, these geometric signals are not elevated to a global object that can be directly coupled with a principled notion of "fair allocation".

On top of this, SVFair introduces VolDet and VolDetPro as *coalition-level geometric conflict measures*. For any task subset $S$, VolDet interprets the determinant of the Gram matrix of task gradients in $S$ as the squared oriented volume of the corresponding gradient parallelotope, yielding a utility function $v(S)$ defined on all coalitions. This quantity jointly encodes the conflict structure among all tasks in $S$, rather than only pairwise information, and is invariant to task ordering and to orthogonal reparameterizations.

Furthermore, VolDetPro recognizes that the standard volumetric measure is *sign-agnostic* with respect to the gradient angle: acute and obtuse angles with the same absolute cosine yield the same volume. To remedy this, VolDetPro augments VolDet with an additional penalty term that is only activated when negative cosine similarities are present, assigning larger conflict scores to coalitions whose gradients form strongly antagonistic (obtuse) angles, while exactly reducing to VolDet when all pairwise cosines are non-negative. In this way, VolDet/VolDetPro are not merely a collection of heuristic pairwise metrics, but a mathematically well-defined, coalition-level volumetric function that enjoys both geometric invariance and sign sensitivity, and that serves as a unified foundation for the subsequent game-theoretic modeling.

## K.2  COOPERATIVE GAME-THEORETIC PERSPECTIVE

From the perspective of cooperative games and fairness-aware optimization, existing work has primarily focused on designing reasonable trade-offs *in loss space*. A canonical example is Nash-MTL (Navon et al., 2022), which formulates MTL as a Nash bargaining problem and derives task weights by solving for a Nash solution that balances task losses. Rafii et al. (Rafi et al., 2024) provide a comprehensive survey of fairness-constrained optimization in federated and multi-task settings. In these approaches, the notion of fairness mainly applies to task losses or high-level utilities, and the weighting rules are determined by the specific bargaining model or constraints, with relatively indirect connections to gradient geometry.

On the other hand, Shapley-related work typically treats *data points or features* as players and studies their contribution to single-task performance, as in Data Shapley (Ghorbani & Zou, 2019) and subsequent gradient-based approximations. In such methods, the utility function $v(S)$ is usually defined as the performance obtainable when training only on the subset $S$, and Shapley values are used for offline data/feature valuation or interpretability, rather than for real-time weighting of task gradients during training.

SVFair's key innovation on the cooperative game-theoretic side is to construct a *task-level cooperative game directly in gradient space*. Here, the players are the tasks, and the utility function $v(S)$ is the "gradient volume conflict" of the subset $S$ as measured by VolDet/VolDetPro. Based on this, we apply Shapley values to allocate this utility in a way that is *uniquely determined by the standard fairness axioms* (efficiency, symmetry, dummy, additivity), resulting in task contributions $\{\phi_i\}$ that are then mapped via a softmax into aggregation weights $\omega_i(\tau)$ at each step.

Unlike Nash-MTL, which negotiates at the level of task losses, SVFair directly imposes fairness on the *responsibility for gradient conflict*: "how much a task contributes to conflict" and "how large an update step it should receive" are coupled within the same formal framework. At the same time, SVFair is clearly distinguished from data Shapley methods that treat samples or features as players and are mainly used for offline valuation.

In summary, SVFair is not a simple combination of existing geometric or game-theoretic methods. Instead, it upgrades gradient geometry into a coalition-level set function $v(S)$ via VolDet/VolDetPro, and then uses Shapley's fair cost sharing to map this back to task-level weights. This unifies *geometric conflict quantification* and *game-theoretic fair allocation* within a single theoretical object, and thereby constitutes a conceptual advance over prior work along both perspectives.

