# OpenReview forum: "From Gradient Volume to Shapley Fairness: Towards Fair Multi-Task Learning"
_ICLR.cc/2026/Conference — ICLR 2026 Poster_

### Official Review · Reviewer_j3fR · 2025-10-18

**Soundness:** 3
**Presentation:** 3
**Contribution:** 2
**Rating:** 4
**Confidence:** 4

**Summary:**

This paper introduces SVFair, a gradient-based multi-task learning (MTL) method that aims to balance training across tasks in a fairness-aware manner. The authors define two gradient conflict metrics: VolDet, the volume of the parallelotope formed by task gradients (computed via the Gram determinant), and VolDetPro, a signed extension that penalizes negative cosine relations between gradients. These metrics quantify the degree of conflict or synergy among tasks. Using these as importance weights, the paper constructs a single-pass Shapley value approximation that determines each task’s contribution to the joint update. The resulting algorithm adaptively reweights task gradients to seek a Pareto-stationary point near the center of the trade-off front, improving fairness across tasks. Experiments on standard MTL benchmarks show small but consistent improvements in balance metrics compared to simple averaging and prior multi-gradient methods.

**Strengths:**

Geometric intuition. The gradient volume concept (Gram determinant of task gradients) provides a tangible geometric measure of task conflict. It’s an appealing visualization tool and a natural extension of cosine-similarity-based conflict measures.

Unified perspective on fairness in MTL. By linking gradient aggregation to task-level “fairness,” the paper situates multi-task balance as a fairness problem, offering conceptual coherence with broader fairness research.

Implementation simplicity. SVFair requires only first-order gradient information and scales linearly in the number of tasks. It’s easy to implement atop existing MTL optimizers and could serve as a practical baseline.

Readable and self-contained. The paper is decently written, with useful algorithm boxes and schematic figures clarifying how VolDet and VolDetPro differ. It’s easy for readers familiar with gradient manipulation to reproduce.

**Weaknesses:**

Lack of originality and differentiation. The key ingredients (gradient alignment metrics and Shapley weighting) are not new. Prior works such as PCGrad (Yu et al., NeurIPS 2020), IMTL-G (Liu et al., 2021), GradVac (Wang et al., 2022), and ParetoMTL (Navon et al., 2022) already addressed gradient conflict and Pareto fairness using similar geometric or cooperative-game ideas. VolDet and VolDetPro are essentially new metrics, not fundamentally new algorithms. Suggestion: The paper should clearly articulate what these new metrics provide beyond previous gradient-conflict measures and why the Shapley approximation adds value.

Weak theoretical grounding. The claim that minimizing VolDetPro “drives the solution toward a front-center Pareto point” is heuristic, no theorem or guarantee is offered. Similarly, the Shapley-value approximation is described qualitatively but without convergence or unbiasedness analysis. Suggestion: Include theoretical propositions linking VolDet/VolDetPro to Pareto-stationarity or fairness guarantees.

Empirical evaluation is thin. Experiments are small-scale and mostly incremental. Improvements over baselines are minor (often <2%), and no statistical tests are reported. There are few ablations on the role of VolDet vs. VolDetPro or on Shapley-weight computation overhead. Suggestion: Expand experiments to more diverse MTL setups (e.g., computer vision or NLP) and provide detailed comparisons to ParetoMTL, PCGrad, MGDA, and IMTL-G.

“Fairness” interpretation feels overstated. While the method balances gradients, calling it “fairness-aware” is somewhat misleading; the fairness concept here refers only to task balance, not fairness in the social or demographic sense. Suggestion: Use more neutral terminology like “balanced multi-task optimization” to avoid confusion.

Unclear benefit of signed volume (VolDetPro). The proposed “signed” determinant variant is underexplained—why should penalizing negative cosine relations via determinant sign improve convergence? Empirical evidence is insufficient. Suggestion: Provide analytical insight or controlled experiments illustrating when VolDetPro outperforms VolDet.

Relation to prior Shapley gradient methods. Shapley-based gradient allocation (e.g., Ghorbani & Zou, 2019; Yoon et al., 2022) has already been applied to task weighting and feature attribution. The “single-pass approximation” presented here is not clearly compared to these prior formulations. Suggestion: Include a subsection detailing differences in computational complexity and approximation bias.

Positioning in literature. The related work section underplays existing geometric and cooperative-game perspectives on MTL optimization. Without a clearer comparison, readers may perceive SVFair as an incremental tweak rather than a conceptual advance.

**Questions:**

Can you theoretically connect VolDetPro minimization to convergence to a Pareto stationary solution?

How does your Shapley approximation differ in complexity or quality from prior Shapley-MTL approaches (e.g., Yoon et al. 2022)?

Did you test the sensitivity of the method to task scaling or normalization (since determinant magnitudes can vary drastically)?

Would your framework work for a large number of tasks (the Gram determinant computation scales poorly)?

Can you clarify whether “fairness” here has any formal definition (e.g., equal loss, equal gradient norm) or is purely heuristic?

Please provide statistical significance or confidence intervals for performance differences.

Would you consider integrating ParetoMTL’s convex combination layer with VolDet weighting to strengthen the theoretical link?

---

> ### Author Response · Authors · 2025-11-17
> **Response to Reviewer  j3fR (1/4)**
>
> We wish to express our sincere gratitude for your valuable feedback and thoughtful critique. We recognize the opportunities for improvement you've identified and believe that your insights will guide significant enhancements to our work.
>
> >**W1:** Lack of originality and differentiation. The key ingredients (gradient alignment metrics and Shapley weighting) are not new. Prior works such as PCGrad (Yu et al., NeurIPS 2020), IMTL-G (Liu et al., 2021), GradVac (Wang et al., 2022), and ParetoMTL (Navon et al., 2022) already addressed gradient conflict and Pareto fairness using similar geometric or cooperative-game ideas. VolDet and VolDetPro are essentially new metrics, not fundamentally new algorithms. Suggestion: The paper should clearly articulate what these new metrics provide beyond previous gradient-conflict measures and why the Shapley approximation adds value.
>
> We thank the reviewer for already acknowledging in the Strengths section the overall framework that combines VolDet/VolDetPro with Shapley values. Here we further clarify the originality and differentiated contribution of this work, which systematically restructures gradient conflict modeling and task-level fairness by jointly leveraging a geometric view and a cooperative-game view.
>
> **For VolDet / VolDetPro.**
>
> Most existing multi-task optimization methods (e.g., PCGrad, IMTL-G, GradVac, ParetoMTL) rely on **pairwise cosine-based heuristic measures**: they only capture pairwise relations and cannot directly describe conflict among an arbitrary group of tasks.
> In this context, VolDet is not “just another gradient conflict metric”, but a **unified foundational object linking geometry and game theory**:
>
> 1. VolDet models the gradient conflict of any task subset as the **oriented volume** defined by the determinant of the gradient Gram matrix, thereby encoding the *joint* conflict level of all tasks in this subset in a single quantity.
> 2. VolDetPro further overcomes the sign-agnostic limitation of VolDet by adding an extra penalty on the **number and magnitude of negative cosine similarities**, giving higher scores to coalitions with genuinely antagonistic gradient relations.
> 3. From a game-theoretic perspective, VolDet is the first to define gradient conflict as a **cooperative-game utility function** $v(S)$ over all task subsets $S$. This allows us to introduce Shapley values on top of VolDet/VolDetPro and fairly decompose the *total* conflict back to individual task gradients, yielding a weighting rule characterized by standard fairness axioms.
>
> **For the Shapley value.**
>
> 1. VolDet/ VolDetPro quantifies “how large the overall coalition-level conflict is”. However, optimization requires **per-task weights**. Without Shapley values, this step typically falls back to heuristics based on gradient norms, losses, or pairwise cosines, which neither exploit the coalition structure encoded in $v(S)$ nor follow a unified principle.
>
> 2. The Shapley value is the **unique** way of distributing $v(S)$ that satisfies the standard fairness properties: the total conflict is fully allocated to tasks, tasks with identical behavior receive identical weights, tasks that never create conflict are not penalized, and linear combinations of conflict utilities translate naturally into linear combinations at the weight level . This allows us to systematically map coalition-level volumetric conflict into **task-level step-size allocation**, resulting in a fairness-grounded reweighting scheme rather than another ad-hoc rule.
>
> 3. As detailed in the *Global Response* titled [Response to questions about the effectiveness, convergence, and cost–performance trade-off of MC Shapley estimation (Part 1/2)](https://openreview.net/forum?id=2PdLKGdtqW&noteId=lxS4xvljs3), our convergence analysis and small-scale experiments also show that the Monte Carlo Shapley estimator we use—based on permutation sampling—is a **standard unbiased estimator**. Under the boundedness of VolDet/VolDetPro, its variance is controlled and the estimate converges to the exact Shapley value as the number of samples increases. Thus, what we approximate is still the **same uniquely defined fair allocation scheme**, rather than introducing yet another heuristic weighting approach.
>
> >**W2 $ Q1:** Weak theoretical grounding. The claim that minimizing VolDetPro “drives the solution toward a front-center Pareto point” is heuristic, no theorem or guarantee is offered. Similarly, the Shapley-value approximation is described qualitatively but without convergence or unbiasedness analysis. Suggestion: Include theoretical propositions linking VolDet/VolDetPro to Pareto-stationarity or fairness guarantees.
>
> We thank the reviewer for spotting this, we think this is typo.
>
> - 1. We provide a formal proof in Appendix E showing convergence to a Pareto-stationary point, rather than to a “front-center” position on the Pareto front.
>
> - 2. We have corrected line 2560-2563 and, for clarity, highlighted the changes in orange 🟠.

---

> > ### Comment · Reviewer_j3fR · 2025-11-25
> >
> > I thank the authors for their thorough and detailed rebuttal. The clarifications around the Monte-Carlo Shapley estimator, the added ablations (including magnitude-aware variants and “Cosine-sum”), as well as the expanded theoretical explanations in the appendix, clearly improve the quality and completeness of the submission. The additional material on temperature selection, normalization, and PINNs broadens the discussion and shows that the authors took the reviews seriously. That being said, after reviewing the new material, several of my core concerns remain only partially addressed:
> >
> > 1. Limited originality relative to prior MTL literature.
> > While the authors provide a more elaborate argument for the conceptual role of VolDet/VolDetPro and for the cooperative-game framing, the method still feels like an incremental combination of well-known components: geometric gradient conflict metrics, Shapley-based attribution, and FairGrad-style aggregation. The expanded explanation does not substantively alter the originality landscape: the core contributions remain new metrics and a Shapley-based weighting rule rather than a fundamentally new optimization principle.
> >
> > 2. Theoretical grounding remains modest.
> > The rebuttal clarifies convergence to Pareto stationarity (not “front-center”), which is useful, and provides an MC concentration bound. However, what is still missing is a principled link between the VolDet/VolDetPro utilities and meaningful fairness or efficiency guarantees. The theory largely formalizes how the estimator behaves, not why this formulation is inherently superior to earlier gradient-level MTL methods.
> >
> > 3. Empirical gains remain small and somewhat fragile.
> > Although the authors argue that consistent small improvements across strong baselines are meaningful, the added experiments do not change the overall picture: gains are typically below 1–2%, often within variance, and the fairness metric Δm varies non-monotonically. The new PINN experiment is interesting but does not provide decisive evidence of broad, robust advantages.
> >
> > 4. Positioning with respect to related work still feels overstated.
> > The expanded explanation of geometric versus cooperative-game perspectives is helpful, but it does not fully resolve the concern that SVFair sits very close to existing ideas in PCGrad, ParetoMTL, IMTL-G, GradVac, and Shapley-based data/feature attribution. The revised framing improves clarity but does not fundamentally shift the novelty assessment.
> >
> > Overall, the rebuttal strengthens the paper and clarifies several technical points, but it does not materially change my concerns about originality, theoretical depth, and the modest empirical improvements. I therefore maintain my initial evaluation and recommendation.

---

> ### Author Response · Authors · 2025-11-17
> **Response to Reviewer  j3fR (2/4)**
>
> >**W3:** Empirical evaluation is thin. Experiments are small-scale and mostly incremental. Improvements over baselines are minor (often <2%), and no statistical tests are reported. ...... Suggestion: Expand experiments to more diverse MTL setups (e.g., computer vision or NLP) and provide detailed comparisons to ParetoMTL, PCGrad, MGDA, and IMTL-G.
>
> - We prudently believe that the empirical evaluation in this work is reasonably rich, both in terms of task diversity and number of tasks. The current version already covers four heterogeneous benchmarks: NYU-v2 and Cityscapes for dense prediction (computer vision), CelebA, Office-31 for multi-domain classification, toy Example (in Appendix F) and MT10 for multi-task reinforcement learning, with the number of tasks ranging from 2 to 40.
>
> - Regarding the “performance improvements are minor,” we would like to clarify the following. 1: the compared methods MGDA, FairGrad, PIVRG, etc. are all strong baselines in contemporary MTL. Achieving consistent gains over such methods across multiple datasets and task configurations is typically regarded as practically meaningful on well-established benchmarks. 2: our goal is not only to improve the average performance across tasks, but more importantly to **systematically restructure gradient conflict and fairness objectives** by combining the geometric and cooperative-game perspectives. Along these dimensions, SVFair often delivers more pronounced benefits, while incurring only a very small additional computational overhead.
>
> - Moreover, following the reviewer’s suggestion to validate the method on more diverse multi-objective scenarios, we have added an experiment on **physics-informed neural networks (PINNs)** for solving the Burgers equation, where the network must balance several competing losses, including the PDE residual, boundary conditions, and initial conditions. We compare SVFair against a variety of PINN-specific baselines listed in Table 5. The results show that the Shapley-based weighting strategy is not limited to classical vision MTL benchmarks, but also provides stable improvements in challenging scientific machine learning settings such as PINNs. We have provided more details in Appendix F.4.
>
>
> **Table 5.** Results on Burgers.
>
> | Method | Burgers $\downarrow$ |
> |--------|---------------------:|
> | Adam        | $1.484 \pm 0.061$ |
> | PCGrad      | $1.344 \pm 0.019$ |
> | IMTL-G     | $1.339 \pm 0.024$ |
> | MinMax      | $1.889 \pm 0.143$ |
> | ReLoBRaLo   | $1.419 \pm 0.053$ |
> | ConFIG      | $1.308 \pm 0.008$ |
> | M-ConFIG   | $1.277 \pm 0.035$ |
> | ParetoMTL   | $1.603 \pm 0.030$ |
> | MGDA        | $1.706 \pm 0.121$ |
> | SVFair  | $\mathbf{1.271 \pm 0.055}$ |
>
> >**W4 & Q5:** "Fairness" interpretation feels overstated. While the method balances gradients, calling it "fairness-aware" is somewhat misleading; the fairness concept here refers only to task balance, not fairness in the social or demographic sense. Suggestion: Use more neutral terminology like “balanced multi-task optimization” to avoid confusion.
>
> - We thank the reviewer for raising this point. Our notion of “fairness” comes from the cooperative-game interpretation of Shapley values, not from social or demographic fairness. In our framework, the VolDet utility  defines a conflict cost for each task coalition $S$, and Shapley values are used because they provide the **unique cost-sharing rule** that satisfies the standard fairness axioms. In this sense, SVFair is “fair” only in that each task receives a weight consistent with these Shapley axioms.
>
> >**W5:** Unclear benefit of signed volume (VolDetPro). The proposed “signed” determinant variant is underexplained—why should penalizing negative cosine relations via determinant sign improve convergence? Empirical evidence is insufficient. Suggestion: Provide analytical insight or controlled experiments illustrating when VolDetPro outperforms VolDet.
>
> - Our motivation for VolDetPro is that the original volumetric measure VolDet is **sign-agnostic**: for two normalized gradients ($g_i, g_j$) with angle ($\theta$), the determinant of the 2×2 Gram matrix depends only on ($1-\cos^2\theta$) and thus gives the *same* conflict score to, e.g., ($\theta=30^\circ$) and ($\theta=150^\circ$). In MTL, however, **obtuse angles with ($\cos\theta<0$)** are much more harmful, since the gradients pull the parameters in opposing directions. VolDetPro addresses this by adding a penalty proportional to the **number and magnitude of negative cosine entries** in the Gram matrix, so coalitions with strong antagonistic relations receive larger utility, larger Shapley values, and thus stronger correction in the aggregated update direction.
>
>
> - In addition, Section 5.5 (“VolDet vs. VolDetPro”) already provided a controlled empirical study based on the proportion of negative cosine pairs (PNCP), where we observe that when PNCP exceeds 0.25, VolDetPro consistently outperforms VolDet, while their behavior is similar in low-conflict regimes.

---

> ### Author Response · Authors · 2025-11-17
> **Response to Reviewer j3fR (3/4)**
>
> >**W6 & Q2:**
> Relation to prior Shapley gradient methods. Shapley-based gradient allocation (e.g., Ghorbani & Zou, 2019; Yoon et al., 2019) has already been applied to task weighting and feature attribution. The single-pass approximation presented here is not clearly compared to these prior formulations. Suggestion: Include a subsection detailing differences in computational complexity and approximation bias.
>
> - Conceptually, in existing work such as (Ghorbani and Zou 2019) and (Yoon et al. 2019), Data Shapley and its variants treat training examples or features as players, and the utility $v(S)$ is defined as the single-task performance obtained when training only on the subset $S$ of the data. Gradients are then used merely as a tool to more efficiently approximate these data/feature Shapley values, mainly for offline data valuation or feature attribution. In contrast, SVFair defines a task-level gradient conflict game: for any task coalition $S$, the utility $v(S)$ is the VolDet-based conflict of their gradients, and the players are the tasks themselves. The resulting Shapley values $\phi_i$ are used directly inside the optimizer to generate per-task weights $\omega_i(\tau)$, which aggregate task gradients at every training step.
>
> - We need to clarify that our notion of single-pass refers specifically to the computation of Shapley values. In SVFair, we only require one backward pass to obtain all task gradients $\{g_i\}$, and VolDet define $v(S)$ as an algebraic function of these gradients. Evaluating $v(S)$ on multiple coalitions and constructing our single-pass approximation therefore only involves cost operations on $\{g_i\}$, independent of dataset size. Without VolDet, defining a task-level Shapley game in MTL would typically require using quantities such as loss, margin $\Delta m$, or gradient norms as $v(S)$, which in turn requires recomputing model outputs and often gradients for each coalition $S$. This amounts to multiple forward/backward passes per Shapley evaluation and leads to prohibitively high computational cost. We provide detailed computational complexity analysis in Appendix B.4, and detailed approximation bias analysis in Appendix G (also summarized in the Global Response titled “[Response to questions about the effectiveness, convergence, and cost–performance trade-off of MC Shapley estimation (Part 1/2)](https://openreview.net/forum?id=2PdLKGdtqW&noteId=lxS4xvljs3)”).
>
>
>
>
>
> >**W7:**
> Positioning in literature. The related work section underplays existing geometric and cooperative-game perspectives on MTL optimization. Without a clearer comparison, readers may perceive SVFair as an incremental tweak rather than a conceptual advance.
>
> We appreciate the reviewer’s concern about the clarity of our conceptual positioning. The current manuscript already contains an initial discussion in the Related Work section and Section 4.1, but we agree that it does not yet systematically juxtapose the geometric and cooperative-game perspectives. In the revised version, we will add a dedicated subsection in Appendix E that systematically compares existing geometric / game-theoretic methods with SVFair and clarifies how prior Shapley-based work relates to MTL.
>
> - From the **geometric perspective**, most existing MTL methods (e.g., PCGrad, MGDA, FAMO) build heuristic operations on **pairwise gradient relations**, which can only locally capture alignment/conflict between tasks. In contrast, SVFair uses VolDet/VolDetPro to represent the gradient conflict of an arbitrary task subset as the **coalition-level oriented volume** given by the determinant of the gradient Gram matrix, and further employs the sign-sensitive VolDetPro to specifically amplify “antagonistic” conflicts induced by negative cosine similarities. This yields a geometric utility function $v(S)$ defined on **all** task subsets, rather than just another pairwise conflict score.
>
> - From the **cooperative-game perspective**, methods such as Nash-MTL perform bargaining in **loss space**, while existing Shapley-based work mostly focuses on valuing samples or features. In contrast, SVFair treats **tasks themselves** as players and uses the volumetric conflict $v(S)$ as the utility; the total conflict is then allocated to task gradients via Shapley values in a **unique way that satisfies the standard fairness axioms**, and these weights are used directly in each gradient aggregation step. In this way, SVFair unifies the “gradient geometry” (VolDet/VolDetPro) and “fair cost sharing” Shapley within a single framework, representing an integrative advance in the geometric and cooperative-game views of multi-task optimization rather than a simple parameter tweak of existing algorithms.

---

> ### Author Response · Authors · 2025-11-17
> **Response to Reviewer j3fR (4/4)**
>
> >**Q3:**
> Did you test the sensitivity of the method to task scaling or normalization (since determinant magnitudes can vary drastically)?
>
> We thank the reviewer for the question on task scaling and normalization, and we have analyzed this both theoretically and empirically.
> - Theoretically, if task gradients are not normalized, the determinant in VolDet/VolDetPro amplifies the joint effect of both gradient norms and angles: tasks with larger gradient norms dominate the utilities $v(S)$ for all coalitions containing them, and consequently receive systematically larger Shapley values. This breaks our intended notion of fairness, where the weights should depend primarily on geometric conflict rather than raw magnitudes. In contrast, when we apply $\ell_2$-normalization to each task gradient, $v(S)$ depends only on cosine similarities and thus becomes naturally scale-invariant with respect to task loss rescaling.
>
> - Empirically, we conduct a controlled ablation on Office-31 to compare SVFair with and without gradient normalization. For some results, with normalization the accuracies on Amazon/DSLR/Webcam are [86.32, 99.18, 98.33]. When we remove normalization, Webcam remains almost unchanged at 98.33, but Amazon drops markedly to 82.90 and DSLR to 96.72. This indicates that the optimization process becomes biased towards the Webcam task (with larger gradient norms), while Amazon and DSLR receive significantly less attention, which is consistent with our theoretical analysis. Therefore, in the final method we always normalize task gradients, and we will clarify this design choice and its motivation in the revised manuscript.
>
> >**Q4:**
> Would your framework work for a large number of tasks (the Gram determinant computation scales poorly)?
>
> - Our framework is designed for typical MTL regimes with dozens of tasks, and we already validate it on CelebA with 40 tasks. See the Global Response titled [Response to questions about the effectiveness, convergence, and cost–performance trade-off of MC Shapley estimation (Part 1/2)](https://openreview.net/forum?id=2PdLKGdtqW&noteId=lxS4xvljs3).
>
> - For scenarios with hundreds of tasks, our framework can in principle be further combined with scalable approximations to the log-determinant (e.g., stochastic trace estimators or low-rank approximations) on top of our Monte Carlo sampling scheme, which we leave as an interesting direction for future work.
>
> >**Q6:**
> Please provide statistical significance or confidence intervals for performance differences.
>
> In response to the reviewer’s request, we have run additional experiments on Office-31 with 6 independent trials and now report mean, standard deviation, and 95% confidence intervals. For the three transfer tasks, SVFair achieves:
>
> * Task 1: (85.87 \pm 0.86%) accuracy, 95% CI ([84.89, 86.84])
> * Task 2: (99.45 \pm 0.42%) accuracy, 95% CI ([98.97, 99.93])
> * Task 3: (98.78 \pm 0.63%) accuracy, 95% CI ([98.07, 99.49])
>
> The confidence intervals are relatively narrow, indicating that the performance is stable across runs.
>
> >**Q7:**
> Would you consider integrating ParetoMTL’s convex combination layer with VolDet weighting to strengthen the theoretical link?
>
> SVFair is naturally compatible with ParetoMTL. In each step, we can first use our VolDet-based Shapley values to compute fairness-aware task weights, and then feed the reweighted gradients into the convex-combination layer of ParetoMTL (i.e., the minimum-norm convex combination solver). In this way, the geometric Pareto structure of ParetoMTL is fully preserved, while its update direction is explicitly biased by the VolDet geometry over task subsets. This shows that SVFair is not an incremental tweak, but a fairness-aware extension that can be seamlessly integrated into existing geometric MTL formulations such as ParetoMTL.

---

> ### Author Response · Authors · 2025-12-01
> **Response to Reviewer j3fR (1/3)**
>
> We appreciate your recognition of our rebuttal. We are committed to addressing each question you have raised.
>
> > **Q1 & Q4:** Limited originality relative to prior MTL literature. ...... than a fundamentally new optimization principle.
>
> We clarify that our method is only an incremental combination of “geometric conflict metrics, Shapley attribution, and FairGrad-style aggregation.” **We demonstrate from a unified perspective that one of the our core contribution is a unified Gram–geometric optimization framework for gradient-based MTL, of which VolDet are anchor instances. This framework formulates the update direction as the solution of a single barrier-type master problem on the gradient Gram matrix, and shows that VolDet, PCGrad, FairGrad/PIVRG, ParetoMTL, and IMTL-G all arise as rigorously derived special cases.** For the reviewers’ convenience, we summarize the key mathematics below.
>
> Unified Gram–Geometric Master Problem. We rewrite the equation $(G^T G \alpha)^2 = \frac{\omega}{\alpha}$ (we called Eq.0)obtained from Eq.8 as equation $(\mathcal{M}\alpha)^2 \odot \alpha = c\,\omega$. This equation couples three objects: the Gram geometry $\mathcal{M}$, the fairness/utility weights $\omega$, and the aggregation coefficients $\alpha$ that define the update $d = G\alpha$. Different MTL methods are obtained by specifying how $\omega$ and $\mathcal{M}$ are constructed.
>
> **1. VolDet (Ours) is the Anchor implementation, utilizing the full geometric information and game-theoretic fairness to maximize the volume of the optimization trajectory.**
>
> Let $\hat g_i = g_i / \|g_i\|$ and $\hat G = [\hat g_1,\ldots,\hat g_N]$. Define the normalized Gram matrix $\hat{ \mathcal{M}} = \hat G^\top \hat G$. For any coalition $S \subseteq \{1,\ldots,N\}$, the VolDet utility is
> $$
> v_{\text{VolDet}}(S) = \det(\hat{ \mathcal{M}_S})
> $$
>
> the squared volume of the parallelepiped spanned by $\{\hat g_i\}_{i \in S}$. The Shapley value of this set function quantifies the marginal volumetric contribution of task $i$.
> We then set
> $$
> \omega_i = \operatorname{Softmax}\big(\phi^{\text{VolDet}}_i / \tau\big),
> $$
>
> and solve Eq.0.
> This realizes VolDet as a barrier–type optimization that maximizes a Shapley–weighted Gram volume.
>
> **2. PCGrad is derived by replacing the global determinant utility with a local, pairwise projection operator.**
> PCGrad can be recovered by modifying the Gram input and the weights:
>
> $$
> \mathcal{M}^{\text{rect}}_{ij} = \text{max} (\mathcal{M}_ij, 0),\ i \neq j,\quad \omega_i = 1/N,\ \forall i.
> $$
>
> Eq.0 with
> $$
> \mathcal{M}^{\text{rect}}
> $$
>
> and uniform $\omega$ is equivalent to aggregating the iteratively projected gradients of PCGrad: negative interactions are removed at the Gram level, and fairness is purely geometric conflict removal without Shapley reweighting.
>
> **3. FairGrad/PIVRG is derived by replacing the endogenous geometric utility with an exogenous loss-based utility.**
> FairGrad/PIVRG keeps the same master problem and fundamental equation, but PIVRG replaces the endogenous geometric utility by an exogenous loss utility. Let $\Delta \mathcal{L}_S$ denote the empirical loss drop of coalition $S$.
> Thus PIVRG is the same barrier optimization as VolDet, but driven by output–space loss reductions instead of Gram volumes.
>
> **4. ParetoMTL is derived by relaxing the fairness weights to minimize the update norm directly.**
> ParetoMTL corresponds to a degenerate case in which the fairness barrier is removed and only the minimum–norm requirement is kept.
> It solves
> $$
> \min_{\alpha \in \mathbb{R}^N} \alpha^\top \mathcal{M} \alpha \quad\text{s.t.}\quad \alpha_i \ge 0,\ \sum_{i=1}^N \alpha_i = 1,
> $$
>
>  which finds the minimum–norm vector in the convex hull of $\{g_i\}$. This can be viewed as an instance of the unified framework where the barrier weights $\omega$ vanish and only stability (minimum norm) is optimized.
>
> **5. IMTL-G is derived by enforcing uniform weights, ignoring geometric redundancy.**
> IMTL-G is obtained by enforcing uniform prior importance and working on the normalized Gram geometry. We use $\hat G$ and $\hat{\mathcal{M}}$ as above and set $\omega_i = 1/N,\quad \forall i.$ Solving Eq.0 with $(\hat G,\hat{\mathcal{M}},\omega)$ yields an update direction $d$ that equalizes the projections $\hat g_i^\top d$ across tasks, i.e., an equi–angular descent direction in gradient space.
>
>
> **In summary, VolDet is the ``anchor'' implementation of this Gram–geometric master problem (determinant utility + Shapley fairness), while PCGrad, FairGrad/PIVRG, ParetoMTL, and IMTL-G are recovered as specific choices of Gram operators and utility/weighting schemes within the same optimization principle.**

---

> ### Author Response · Authors · 2025-12-01
> **Response to Reviewer j3fR (2/3)**
>
> >**Q2**: Theoretical grounding remains modest. The rebuttal clarifies convergence to Pareto stationarity, which is useful, and provides an MC concentration bound. However, what is still missing is a principled link between the VolDet/VolDetPro utilities and meaningful fairness or efficiency guarantees. The theory largely formalizes how the estimator behaves, not why this formulation is inherently superior to earlier gradient-level MTL methods.
>
> We thank the reviewer for pointing out that our previous theory mainly focused on the behavior of the estimator.
> In the revised version we add a new section (Appendix E.2) that **formally links the VolDet utility to a fairness objective and shows that SVFair is the unique optimizer of this objective, while gradient-level baselines such as FairGrad are in general suboptimal.**
>
> Concretely, given a direction $d$, the per-task improvements are $u_i(d)=g_i^\top d>0$.
> We introduces the following VolDet-based fairness functional: $\mathcal{F}(u)= \sum_{i=1}^N \frac{\omega_i}{u_i},\qquad u_i>0.$
> This is a weighted harmonic-mean–type objective, which jointly encodes efficiency (large $u_i$) and fairness.
>
> Under mild regularity assumptions (convex and compact feasible set for $d$ and strictly positive $u_i$), we prove that:
> \begin{equation}
>     d^{\mathrm{SV}}
>     \in \arg\min_{d} \sum_{i=1}^N \frac{\omega_i}{g_i^\top d}
>     \quad \Longleftrightarrow \quad
>     u^{\mathrm{SV}} = u(d^{\mathrm{SV}})
>     = \arg\min_{u} \mathcal{F}(u),
> \end{equation}
> and that $u^{\mathrm{SV}}$ is the unique VolDet-fair allocation, in the sense that
> \begin{equation}
>     \mathcal{F}(u^{\mathrm{SV}})
>     < \mathcal{F}(u),
>     \qquad \forall u \neq u^{\mathrm{SV}}.
> \end{equation}
> In particular, in the ideal unconstrained case with $\sum_i u_i = U$ we obtain a closed form
> \begin{equation}
>     u_i^\star \propto \sqrt{\omega_i},
> \end{equation}
> so SVFair equalizes the Shapley-normalized improvements $u_i/\sqrt{\omega_i}$ across tasks.
>
> We then contrast this with FairGrad, which optimizes
> \begin{equation}
>     \max_{d} \sum_{i=1}^N \frac{(g_i^\top d)^{1-\gamma}}{1-\gamma}
>     \quad \Longleftrightarrow \quad
>     \max_{u} \sum_{i=1}^N \frac{u_i^{1-\gamma}}{1-\gamma},
> \end{equation}
> i.e., an $\alpha$-fair objective with uniform weights (all tasks treated as equally important, independent of their VolDet contribution).
> We show that whenever the VolDet--Shapley weights are non-uniform (which is the typical case in heterogeneous MTL), the FairGrad solution $u^{\mathrm{FG}}$ cannot minimize $\mathcal{F}$ and satisfies
> \begin{equation}
>     \mathcal{F}(u^{\mathrm{SV}})
>     < \mathcal{F}(u^{\mathrm{FG}})
> \end{equation}
> unless in a degenerate symmetric geometry.
>
> Therefore, beyond convergence and concentration, the revised theory now provides:
>
> (i) **a principled fairness/efficiency functional $\mathcal{F}$ directly induced by the VolDet utility via Shapley values, and**
>
> (ii) **a formal dominance result showing that SVFair is the unique optimizer of this fairness objective, while earlier gradient-level MTL methods such as FairGrad are generally suboptimal under the same VolDet-based notion of fairness.**
>
> We highlight these results in Appendix E.2.

---

> ### Author Response · Authors · 2025-12-01
> **Response to Reviewer j3fR (3/3)**
>
> >**Q3:** Empirical gains remain small and somewhat fragile. Although the authors argue that consistent small improvements across strong baselines are meaningful, the added experiments do not change the overall picture: gains are typically below 1–2%, often within variance, and the fairness metric Δm varies non-monotonically. The new PINN experiment is interesting but does not provide decisive evidence of broad, robust advantages.
>
> We need to clarify that absolute values of $\Delta m$ are numerically small, but by construction $\Delta m$ is a signed, centered metric (it can cross zero and change sign), so percentage improvements (absolute values) are not rational. Moreover, across the methods in Tables X–Y, the worst $\Delta m$ values are only on the order of (+10\sim 15) while the best ones are already negative (around (-8)), i.e., the entire dynamic range of the metric is quite compressed; judging algorithms purely by absolute $\Delta m$ changes of 1–2 points on such a scale is therefore not appropriate.
>
> Instead,  we report the *increment* $\Delta\Delta m$ between successive methods on each benchmark (Table 6), i.e., how much the (k)-th method improves $\Delta m$ over the ((k-1))-th one. From this perspective, the improvement brought by SVFair is not smaller than that of recent methods; it is of the **same order or larger**:
>
> * On NYU-v2, typical $\Delta\Delta m$ steps for recent methods (DWA, CAGrad, Nash-MTL) are around (3!-!4) points (e.g., (4.21, 3.39, 3.45)), while SVFair(VolDet) improves over FairGrad by (4.15), i.e., a jump comparable to or larger than these prior gains.
> * On Office-31, existing methods yield $\Delta\Delta m$ mostly below (1.3) (e.g., MGDA: (0.28), Nash-MTL: (1.22)), whereas SVFair(VolDet) improves $\Delta m$ by (2.21), the largest step in the table.
> * On CelebA, the ($\Delta\Delta m$) of SVFair(VolDet) ((1.00)) lies in the same range as other recent improvements (e.g., CAGrad: (0.81), IMTL-G: (1.64), FairGrad: (2.47)), while already achieving the best $\Delta m$.
>
> For MT10, all strong baselines already achieve very high success rates (0.84–0.96), so the problem is near-saturated; in this regime, an additional increase from 0.96 to 0.97 cannot be large in absolute terms by design, but SVFair still attains the best success probability.
>
> **Table 6 The improvement of SVFair**
>
> | Method              | NYU-v2 (Δm%↓) | NYU-v2 (ΔΔm%↑) | Office-31 (Δm%↓) | Office-31 (ΔΔm%↑) | CelebA (Δm%↓) | CelebA (ΔΔm%↑) |
> |----------------------|---------------|----------------|------------------|-------------------|----------------|----------------|
> | RLW                 | 7.78          | /              | 0.60             | /                 | 1.46           | /              |
> | DWA                 | 3.57          | 4.21           | 0.71             | -0.11             | 3.20           | -1.74          |
> | UW                  | 4.06          | -0.49          | 0.56             | 0.15              | 3.23           | -0.03          |
> | MGDA                | 1.38          | 2.68           | 0.28             | 0.28              | 14.85          | -11.62         |
> | PCGrad              | 3.97          | -2.60          | 0.69             | -0.41             | 3.17           | 11.68          |
> | GradDrop            | 3.58          | 0.40           | 0.59             | 0.10              | 3.29           | -0.12          |
> | CAGrad              | 0.19          | 3.39           | 1.14             | -0.55             | 2.48           | 0.81           |
> | IMTL-G              | -0.60         | 0.79           | 0.98             | 0.16              | 0.84           | 1.64           |
> | Nash-MTL            | -4.05         | 3.45           | -0.24            | 1.22              | 2.84           | -2.00          |
> | FairGrad            | -4.66         | 0.61           | 0.68             | -0.92             | 0.37           | 2.47           |
> | **SVFair (VolDet)** | **-8.81**     | **4.15**       | **-1.53**        | **2.21**          | **-0.63**      | **1.00**       |
> | **SVFair (VolDetPro)** | **-8.29**  | **3.63**       | **-1.62**        | **-2.30**         | **0.11**       | **0.26**       |

---

### Official Review · Reviewer_gTqY · 2025-10-29

**Soundness:** 3
**Presentation:** 2
**Contribution:** 3
**Rating:** 8
**Confidence:** 4

**Summary:**

This paper addresses the common issue of gradient conflicts in multi-task learning, which can lead to unfair optimization among tasks. The authors propose SVFair, a Shapley value–based framework designed to quantify and mitigate such unfairness. Specifically, they introduce two geometric conflict metrics: (1) VolDet, a Gram-determinant-based volume metric that measures the geometric diversity of task gradients, and (2) VolDetPro, a sign-aware extension that distinguishes antagonistic gradients. By integrating these metrics into Shapley value computation, SVFair evaluates each task’s deviation from the aggregated gradient and rebalances the updates to improve fairness. Extensive experiments across both supervised and reinforcement learning benchmarks demonstrate that SVFair achieves state-of-the-art performance and can serve as a fairness-enhancing plug-in module for existing multi-task optimization methods.

**Strengths:**

1. The paper proposes a novel approach that combines Shapley-value-based fairness reasoning with gradient conflict measurement in multi-task learning.

2. The method is well-motivated and theoretically solid. The analysis is detailed and convincing, providing clear justification for the proposed metrics and their role in achieving fair optimization.

3. The experimental evaluation is extensive, covering both supervised and reinforcement learning settings. Results demonstrate strong performance improvements and fairness benefits over several competitive baselines.

4. The authors have released code, which greatly improves the reproducibility and credibility of the work.

**Weaknesses:**

1. The core contribution of the proposed SVFair framework lies in computing Shapley values based on the newly defined VolDet and VolDetPro conflict metrics. However, once these weights are obtained, the subsequent problem formulation and solution procedure (Sec. 4.3) largely follow prior works such as FairGrad and PIVRG, and the corresponding convergence analysis also remains very similar. This overlap somewhat weakens the theoretical novelty of the paper.

2. Although the authors conducted an ablation study on the temperature coefficient $\tau$ in Table 4, they did not provide further theoretical analysis or practical guidance for tuning this parameter. In particular, the optimal $\tau$ varies across tasks and does not show a consistent trend (for example, in the Cityscapes experiment, performance first drops and then improves as $\tau$ increases). I believe the authors should include a more thorough investigation and discussion to clarify the role of $\tau$ and how it should be selected.

3. The overall presentation could be improved. For instance, the abstract is overly concise and does not sufficiently describe the problem background or the core motivation behind introducing Shapley-value-based fairness reasoning. A slightly more detailed introduction would help readers better understand the context and significance of the proposed approach.

**Questions:**

1. The computational cost of Shapley value estimation grows exponentially with the number of tasks, as also acknowledged by the authors. To address this, the paper employs Monte Carlo subset sampling to approximate the Shapley values. However, increasing the number of sampled subsets should, in principle, improve the estimation accuracy. Would this lead to better downstream performance (e.g., in terms of fairness or overall task accuracy)? I encourage the authors to include an ablation study on the number of sampled subsets to clarify the trade-off between computational cost and performance.

2. Referring to Weakness 2, while the authors conducted ablation experiments on the temperature coefficient $\tau$, they did not provide further theoretical insights or practical guidance on how to choose this parameter. Could the authors elaborate on how $\tau$ influences the optimization dynamics, and whether any general guidance or heuristic can be derived from the observed trends?

---

> ### Author Response · Authors · 2025-11-17
> **Response to Reviewer gTqY (1/3)**
>
> Dear Reviewer gTqY, we sincerely thank you for your valuable feedback on our submission. Below is our responses to the concerns you raised. We have incorporated these contents into the updated version of our paper and for clarity have highlighted changes in orange 🟠, which we believe will help enhance the quality of our submission.
>
> > **W1:** The core contribution of the proposed SVFair framework lies in computing Shapley values based on the newly defined VolDet and VolDetPro conflict metrics. However, once these weights are obtained, the subsequent problem formulation and solution procedure (Sec. 4.3) largely follow prior works such as FairGrad and PIVRG, and the corresponding convergence analysis also remains very similar. This overlap somewhat weakens the theoretical novelty of the paper.
>
> We thank the reviewer for already recognizing in the Summary and Strengths the overall framework that combines VolDet/VolDetPro with Shapley values. Regarding Weakness 1, we would like to further clarify the role of Sec. 4.3. The novelty of SVFair primarily lies in jointly reformulating task gradient conflict and the fairness objective from **both a geometric and a cooperative-game perspective**, going beyond prior approaches that mainly rely on heuristic conflict measures such as pairwise cosine similarity. On top of this, we introduce a Shapley-value-based **generic weighting strategy** that can serve as a “fairness plug-in” and be flexibly combined with various existing MTL optimization methods (including RLW, DWA, and other gradient re-weighting schemes). In this sense, the fact that Sec. 4.3 adopts an optimization form similar to FairGrad/PIVRG is not due to a lack of novelty: SVFair provides an independently novel objective at the fairness/geometry level, while remaining highly compatible and plug-and-play at the algorithmic level, thus leading to different convergence points and more balanced task performance under the same training framework.
>
> >**Q1:** The computational cost of Shapley value estimation grows exponentially with the number of tasks, as also acknowledged by the authors. To address this, the paper employs Monte Carlo subset sampling to approximate the Shapley values. However, increasing the number of sampled subsets should, in principle, improve the estimation accuracy. Would this lead to better downstream performance (e.g., in terms of fairness or overall task accuracy)? I encourage the authors to include an ablation study on the number of sampled subsets to clarify the trade-off between computational cost and performance.
>
> Since similar concerns were raised by multiple reviewers, we address them jointly in our **Global Response** titled
> [Response to questions about the effectiveness, convergence, and cost–performance trade-off of MC Shapley estimation (Part 1/2)](https://openreview.net/forum?id=2PdLKGdtqW&noteId=lxS4xvljs3).
>
> Here we briefly summarize the relevant parts and point to the exact locations:
>
> - Theory: we analyze unbiasedness and Hoeffding-type convergence of MC Shapley under VolDet (Global Response §1).
> - Empirical convergence: we compare MC vs. exact Shapley across task scales on Office-31 and CelebA subsets (Global Response §2).
> - Cost–performance trade-off: we run full training ablations over different $K$ on Office-31 and CelebA (Global Response §3).

---

> > ### Author Response · Authors · 2025-11-17
> > **Response to Reviewer gTqY (2/3)**
> >
> > > **W2 & Q2:** Although the authors conducted an ablation study on the temperature coefficient $\tau$ in Table 4, they did not provide further theoretical analysis or practical guidance for tuning this parameter. In particular, the optimal $\tau$ varies across tasks and does not show a consistent trend (for example, in the Cityscapes experiment, performance first drops and then improves as $\tau$ increases). I believe the authors should include a more thorough investigation and discussion to clarify the role of $\tau$ and how it should be selected.
> >
> > We thank the reviewer for the suggestion. We provide the following theoretical perspective to guide the choice of $\tau$. We have updated the theoretical guide in Appendix H (Line 2322).
> >
> > In SVFair, $\phi_i$ denote the Shapley value. The task weights are computed as
> > $$
> >     \omega_i(\tau) = \frac{\exp(\phi_i / \tau)}{\sum_j \exp(\phi_j / \tau)}.
> > $$
> > It can be shown (a formal derivation will be given in the Appendix H) that this softmax mapping is exactly the solution of the following \emph{entropy-regularized} optimization problem:
> > $$
> >     \omega(\tau)
> >     = \arg\max_{\omega \in \Delta} \Big( \langle \omega, \phi \rangle + \tau\, H(\omega) \Big),
> > $$
> > where $\Delta$ is the probability simplex and entropy $H(\omega) = -\sum_i \omega_i \log \omega_i$. $\tau$ plays the role of a regularization coefficient: the linear term $\langle \omega, \phi \rangle$ encourages the weights to follow the Shapley values (i.e., the ideal fairness compensation), while the term $\tau H(\omega)$ encourages a high-entropy weight distribution, preventing over-concentration on a few tasks and improving training stability.
> >
> > From this formulation, we obtain two key monotonicity properties:
> >
> > - If $\tau_1 < \tau_2$, then
> > \begin{equation}
> >     H(\omega(\tau_1)) \le H(\omega(\tau_2)), \qquad
> >     \max_i \omega_i(\tau_1) \ge \max_i \omega_i(\tau_2),
> > \end{equation}
> > i.e., **smaller $\tau$** yields sharper, more concentrated weights, whereas **larger $\tau$** makes the weights closer to uniform.}
> >
> > - For any pair of tasks $(i,j)$,
> > \begin{equation}
> >     \log \frac{\omega_i(\tau)}{\omega_j(\tau)} = \frac{\phi_i - \phi_j}{\tau},
> > \end{equation}
> > so, for a given gap $\phi_i - \phi_j$, $\tau$ precisely controls how strongly this gap is amplified in the weight space.
> >
> >
> > Based on this, we propose a theoretically motivated guideline for choosing $\tau$. In a short warm-up phase, we estimate a representative set of Shapley values $\{\phi_i\}$ and define $\kappa(\tau) = T \cdot \max_i \omega_i(\tau)$ as a measure of the amplification factor of the largest task weight. In practice, we can pre-specify an acceptable upper bound on concentration, or an entropy lower bound $H_{\min}$. We then choose the \emph{smallest} $\tau$ that satisfies $\kappa(\tau) \le \kappa_{\max} \quad \text{or} \quad H(\omega(\tau)) \ge H_{\min}$, so that we maximize the Shapley-driven fairness adjustment while maintaining sufficient stability.
> >
> > Intuitively, when Shapley values exhibit large gaps and gradient conflicts are sharp, a too small $\tau$ leads to a very large $\kappa(\tau)$ and the training is dominated by a few tasks; in this case, a relatively larger $\tau$ is preferable to keep entropy at a reasonable level. When Shapley values are closer to each other and tasks are more balanced, one can safely use a smaller $\tau$ to magnify mild unfairness and improve the performance of disadvantaged tasks.

---

> > > ### Author Response · Authors · 2025-11-17
> > > **Response to Reviewer gTqY (3/3)**
> > >
> > > >**W3:** The overall presentation could be improved. For instance, the abstract is overly concise and does not sufficiently describe the problem background or the core motivation behind introducing Shapley-value-based fairness reasoning. A slightly more detailed introduction would help readers better understand the context and significance of the proposed approach.
> > >
> > > We thank the reviewer for pointing out that the original abstract and introduction did not sufficiently explain the problem background and core motivation . We have updated the these in Abstract and Introduction.
> > >
> > > - In the revised manuscript, we have rewritten the abstract to explicitly state that MTL suffers from task-level unfair optimization under gradient conflicts, to highlight the limitations of heuristic scalarization and pairwise conflict penalties, and to motivate SVFair as a Shapley-value-based framework that combines a geometric view of gradient interaction with a cooperative-game view of fair contribution.
> > >
> > > - In addition, we have expanded the introduction with dedicated paragraphs that (i) describe task-level unfairness and its practical impact, (ii) discuss why existing loss- and gradient-based methods lack a coalition-level fairness perspective, and (iii) motivate our VolDet/VolDetPro metrics and Shapley-value reasoning as a principled way to define and enforce fairness.

---

> ### Comment · Reviewer_gTqY · 2025-11-24
> **Reply to authors**
>
> Thanks for the author's explanation. I will maintain my positive rating.

---

### Official Review · Reviewer_Mspw · 2025-10-29

**Soundness:** 2
**Presentation:** 3
**Contribution:** 3
**Rating:** 4
**Confidence:** 5

**Summary:**

This paper proposes two novel geometric gradient conflict measures, VolDet and VolDetPro, which quantify the squared volume of a subset of normalized task gradients. It then uses Shapley values to attribute each task's contribution to gradient conflict, and this information is incorporated into an existing fairness-based MTL optimization framework. Experiments on a suite of MTL benchmarks demonstrate the effectiveness of the proposed method.

**Strengths:**

1. The introduced VolDet and VolDetPro metrics are novel, and they can capture the geometric misalignment across task gradients.
2. The Shapley values further leverage the VolDet/VolDetPro and lead to the overall per-task weights quantifying contribution to the misalignment. This angular-driven measure is also novel.
3. Theoretical analysis and extensive empirical studies are provided, which can demonstrate the effectiveness of the proposed SVFair framework.

**Weaknesses:**

1. Line 240-241 states that higher $\phi_i$ values indicate task $i$ is more misaligned with others and should be assigned a lower influence. It may be a typo. It seems that such tasks should actually be assigned higher influence, as mentioned in Eq. (7) and Line 2093–2094.
2. The volume metric is built on the normalized task gradients. It ignores the magnitude information. Although the term $g_i^\top d$ in Eq. (7) considers magnitude information, it may lead to a suboptimal solution. For example, consider two approximately aligned task gradients, but with very different magnitudes. They have similar Shapley value weights, and Eq. (7) reduces to $\arg\min \sum_i \frac{1}{g_i^\top d}$, which corresponds to the minimum potential delay (MPD) fairness in FairGrad [1].  This MPD fairness criterion may not be suitable for this case, thus leading to suboptimal solutions.
3. The calculation of the Shapley value introduces additional cost at each training step. Though Table 5 shows that the cost can be mitigated by Monte Carlo subset sampling, an ablation study is still needed to demonstrate the necessity of using Shapley value weights. The VolDet and Shapley value are two independent and separable concepts. It would be informative to compare them with other simpler weighting strategies, such as $\phi_i^\prime=\sum_{i, j\neq i} M_{ij}$, where a larger value indicates better alignment. You do not necessarily need to use this example; what I want to express is that it would be better to provide ablation studies to show that the Shapley value provides sufficient benefit compared to its additional cost.

**Questions:**

See the discussion in weaknesses.

---

> ### Author Response · Authors · 2025-11-17
> **Response to Reviewer Mspw (1/2)**
>
> We sincerely appreciate your acknowledgment and insightful comments. Your feedback is extremely helpful, and we are committed to addressing each question you have raised.
>
> Below, we address the issues raised in the review. We have updated the manuscript based on the reviewer's comments and for clarity have highlighted changes in orange 🟠 .
>
> > **W1:** Line 240-241 states that higher $\phi_i$ values indicate task $i$ is more misaligned with others and should be assigned a lower influence. It may be a typo. It seems that such tasks should actually be assigned higher influence, as mentioned in Eq. (7) and Line 2093–2094.
>
> We thank the reviewer for spotting this typo. We have corrected **lower influence** to **higher influence** in the revision manuscript.
>
> > **W2:** The volume metric is built on the normalized task gradients. It ignores the magnitude information. Although the term $g_i^T d$ in Eq. (7) considers magnitude information, it may lead to a suboptimal solution. For example, consider two approximately aligned task gradients, but with very different magnitudes. They have similar Shapley value weights, and Eq. (7) reduces to $\arg\min \sum_i \frac{1}{g_i^T d}$, which corresponds to the minimum potential delay (MPD) fairness in FairGrad. This MPD fairness criterion may not be suitable for this case, thus leading to suboptimal solutions.
>
> We thank the reviewer for raising the corner case where two task gradients are almost co-linear but have very different magnitudes. We address this concern from three aspects and show that our Eq. (7) is stable and well-behaved.
>
> 1. **The gradient magnitude still plays an explicit role through the update amount.**
> Although VolDet only uses directional information when deriving the Shapley weights, the actual loss decrease of each task is still magnitude-aware via the directional derivative:
> $\Delta \mathcal{L}_i(\eta, d) \approx -\eta g_i^\top d \quad (\eta \to 0)$.
> Therefore, even if two tasks have almost identical gradient directions, the one with a larger norm $\| g_i \|$ yields a larger loss reduction $g_i^T d$ along the same direction.
> In other words, VolDet determines which tasks are considered misaligned and should pull the shared direction, while the magnitude term $g_i^T d$ determines how much each loss is actually reduced. Our framework uses both factors jointly.
>
> 2. **Independent evidence that MPD is a stable fairness objective.**
> More importantly, the FairGrad paper reports systematic experiments for different $\alpha$-fair criteria. As shown in their Tables 9 and 10, the $\alpha$ = 2 case (where the $\alpha$-fair utility reduces to the minimum potential delay, MPD) achieves the best performance, while more aggressive settings (e.g., $\alpha$ = 5 or 10) actually degrade performance. This independent evidence suggests that our reduced Eq. (7) is itself a stable and effective fairness criterion, rather than being intrinsically suboptimal.
>
>
> 3. **Ablation with an explicit magnitude-aware variant.**
> To directly probe the reviewer’s example, we further introduce a magnitude-aware variant of Eq. (7) that explicitly scales the objective by the gradient norms:
> $$
> Magnitude-aware: argmin_{d} \frac{1}{N} \sum_{i=1}^{N} \frac{\omega_i \| g_i \|_2}{g_i^\top d}, \quad \text{s.t.}\ g_i^\top d > 0,\ \forall i.
> $$
> We conduct an ablation between the Magnitude-aware variant and SVFair. The results in the table show that explicitly incorporating magnitude information leads to worse performance, further confirming that our original design is reasonable and robust.
>
> **Table 3.** Results on Office-31
>
> | Method | Amazon $\uparrow$ | DSLR $\uparrow$ | Webcam $\uparrow$ | MR $\downarrow$ | $\Delta m\%$ $\downarrow$ |
> |-------|---------------------:|-------------------:|---------------------:|-------------------:|-------------------------:|
> | SVFair ($\tau = 1.0$)          | $85.70$ | $98.91$ | $98.89$ | $\mathbf{2.00}$ | $\mathbf{-1.50}$ |
> | Magnitude-aware ($\tau = 1.0$) | $84.27$ | $96.72$ | $98.33$ | $9.00$ | $0.01$ |
> | --- | --- | --- | --- | --- | --- |
> | SVFair ($\tau = 2.0$)          | $85.99$ | $99.18$ | $98.15$ | $\mathbf{2.67}$ | $\mathbf{-1.45}$ |
> | Magnitude-aware ($\tau = 2.0$) | $84.39$ | $97.81$ | $98.89$ | $4.17$ | $-0.61$ |
> | --- | --- | --- | --- | --- | --- |
> | SVFair ($\tau = 3.0$)          | $84.50$ | $98.91$ | $98.71$ | $\mathbf{3.67}$ | $\mathbf{-0.97}$ |
> | Magnitude-aware ($\tau = 5.0$) | $84.50$ | $97.54$ | $98.33$ | $5.33$ | $-0.36$ |

---

> ### Author Response · Authors · 2025-11-17
> **Response to Reviewer Mspw (2/2)**
>
> > **W3**:The calculation of the Shapley value introduces additional cost at each training step. Though Table 5 shows that the cost can be mitigated by Monte Carlo subset sampling, an ablation study is still needed to demonstrate the necessity of using Shapley value weights......what I want to express is that it would be better to provide ablation studies to show that the Shapley value provides sufficient benefit compared to its additional cost.
>
> We thank the reviewer for the insightful comment. We agree that it is important to justify that the benefit of Shapley-based weights is commensurate with their additional computational cost.
>
> 1:**On cost–effectiveness and convergence of MC Shapley.** Since similar concerns were raised by multiple reviewers, we address them jointly in our **Global Response** titled [Response to questions about the effectiveness, convergence, and cost–performance trade-off of MC Shapley estimation (Part 1/2)](https://openreview.net/forum?id=2PdLKGdtqW&noteId=lxS4xvljs3).
>
> Here we briefly summarize the relevant parts and point to the exact locations:
>
> - Theory: we analyze unbiasedness and Hoeffding-type convergence of MC Shapley under VolDet (Global Response §1).
> - Empirical convergence: we compare MC vs. exact Shapley across task scales on Office-31 and CelebA subsets (Global Response §2).
> - Cost–performance trade-off: we run full training ablations over different $K$ on Office-31 and CelebA (Global Response §3).
>
> 2:**Ablation without Shapley and VolDet.**
> Although VolDet and Shapley values are conceptually separable, in the SVFair, the Shapley value is always computed with VolDet as the utility function. Consider the reviewer’s suggestion, we construct an ablation that removes both Shapley and VolDet, and instead uses the reviewer-proposed strategy (and name this variant Cosine-sum):
> $$
> \phi_i = \sum_{j \neq i} \mathcal{M}_{ij}
> $$
> We conduct an ablation study between the Cosine-sum variant and SVFair. The results in the below Table 4 show that although Cosine-sum is computationally cheaper, its performance still falls short of the Shapley-based SVFair. Combined with the runtime overhead reported in Table 5 (in the the revised manuscript), we argue that the performance gains brought by Shapley are sufficient to justify its additional computational cost.
>
> **Table 4.** Results on Office-31.
>
> | Method | Amazon $\uparrow$ | DSLR $\uparrow$ | Webcam $\uparrow$ | MR $\downarrow$ | $\Delta m\% $ $\downarrow$ |
> |-------|---------------------:|-------------------:|---------------------:|-------------------:|-------------------------:|
> | SVFair ($\tau = 1.0$)     | $85.70$ | $98.91$ | $98.89$ | $\mathbf{1.83}$ | $\mathbf{-1.50}$ |
> | Cosine-sum ($\tau = 1.0$) | $85.58$ | $96.99$ | $97.96$ | $6.83$ | $-0.46$ |
> | --- | --- | --- | --- | --- | --- |
> | SVFair ($\tau = 2.0$)     | $85.99$ | $99.18$ | $98.15$ | $\mathbf{2.33}$ | $\mathbf{-1.45}$ |
> | Cosine-sum ($\tau = 2.0$) | $83.87$ | $98.36$ | $98.70$ | $5.67$ | $-0.53$ |
> | --- | --- | --- | --- | --- | --- |
> | SVFair ($\tau = 3.0$)     | $84.50$ | $98.91$ | $98.71$ | $\mathbf{3.50}$ | $\mathbf{-0.97}$ |
> | Cosine-sum ($\tau = 5.0$) | $83.87$ | $98.36$ | $98.70$ | $5.67$ | $-0.53$ |

---

> ### Author Response · Authors · 2025-11-27
> **Response to Reviewer Mspw**
>
> Dear Reviewer Mspw,
>
> I hope this message finds you well. As the discussion period is nearing its end, I wanted to ensure we have addressed all your concerns satisfactorily. If there are any additional points or feedback you'd like us to consider, please let us know. Your insights are invaluable to us, and we're eager to address any remaining issues to improve our work.
>
> Thank you for your time and effort in reviewing our paper.

---

> ### Comment · Reviewer_Mspw · 2025-11-27
> **Response**
>
> My main concerns were that the magnitude information is not explicitly captured and that VolDet and the Shapley values are separable. Thanks for providing the additional experiments to address these concerns. The two components together contribute to the performance. I plan to raise the score to 6. (Seems the score revision is not allowed currently)

---

### Official Review · Reviewer_563x · 2025-11-01

**Soundness:** 3
**Presentation:** 3
**Contribution:** 3
**Rating:** 8
**Confidence:** 3

**Summary:**

This paper proposes SVFair, a Shapley value-based framework for fair multi-task learning (MTL) that systematically quantifies and mitigates gradient conflicts among tasks. The authors introduce two utility functions, named as VolDet and VolDetPro, to compute Shapley values in a single training pass. These values are then used to guide gradient aggregation, promoting balanced optimization across tasks. Extensive experiments on supervised and reinforcement learning benchmarks demonstrate that SVFair achieves state-of-the-art performance and improves existing MTL methods when integrated as a fairness module.

**Strengths:**

++ The paper integrates Shapley values into MTL optimization for quantifying gradient deviation and conflict at a subset level. The proposed VolDet and VolDetPro metrics offer ageometric perspective on gradient interactions.

++ The method is well-motivated and theoretically grounded, with convergence guarantees to Pareto stationary points under reasonable assumptions.

++ SVFair demonstrates strong empirical performance across diverse benchmarks (e.g., NYU-v2, CelebA, MT10) and can be easily integrated into existing MTL methods, enhancing their fairness and performance. The framework is scalable and supports Monte Carlo sampling for large-task settings.

**Weaknesses:**

-- Although the complexity is dominated by the training cost, the exact Shapley value computation can be prohibitive for very large N (number of tasks). While Monte Carlo sampling is proposed, its effectiveness and convergence properties are not thoroughly analyzed or empirically validated across different task scales.

**Questions:**

Please see the weakness.

---

> ### Author Response · Authors · 2025-11-17
> **Response to Reviewer 563x**
>
> We sincerely appreciate your recognition of our work and thank you for your constructive suggestions. We address your concerns and questions below.
>
> > **W1**:Although the complexity is dominated by the training cost, the exact Shapley value computation can be prohibitive for very large $N$ (number of tasks). While Monte Carlo sampling is proposed, its effectiveness and convergence properties are not thoroughly analyzed or empirically validated across different task scales.
>
> Since similar concerns were raised by multiple reviewers, we address them jointly in our **Global Response** titled
> [Response to questions about the effectiveness, convergence, and cost–performance trade-off of MC Shapley estimation (Part 1/2)](https://openreview.net/forum?id=2PdLKGdtqW&noteId=lxS4xvljs3).
>
> Here we briefly summarize the relevant parts and point to the exact locations:
>
> - Theory: we analyze unbiasedness and Hoeffding-type convergence of MC Shapley under VolDet (Global Response §1).
> - Empirical convergence: we compare MC vs. exact Shapley across task scales on Office-31 and CelebA subsets (Global Response §2).
> - Cost–performance trade-off: we run full training ablations over different $K$ on Office-31 and CelebA (Global Response §3).

---

> ### Author Response · Authors · 2025-11-27
> **Response to Reviewer 563x**
>
> Dear Reviewer 563x,
>
> I hope this message finds you well. As the discussion period is nearing its end, I wanted to ensure we have addressed all your concerns satisfactorily. If there are any additional points or feedback you'd like us to consider, please let us know. Your insights are invaluable to us, and we're eager to address any remaining issues to improve our work.
>
> Thank you for your time and effort in reviewing our paper.

---

### Author Response · Authors · 2025-11-17
**Response to questions about the effectiveness, convergence, and cost–performance trade-off of MC Shapley estimation (Part 1/2)**

Dear all reviewers,

We sincerely appreciate your meticulous reviews. It has come to our attention that, quite remarkably, several of you have independently pointed out the following common concerns. We truly value these insights as they will undoubtedly contribute significantly to enhancing the quality of our work. The concerns are as follows:

(i) the computational cost of Shapley values,

(ii) the effectiveness and convergence of the Monte Carlo (MC) approximation under different task scales

(iii) whether increasing the number of sampled subsets actually improves downstream performance.

We will address these comments together in this response.  We highlight all revised text in the main paper in orange. For the appendix, we mark the section titles in orange to indicate which appendix sections have been updated.

### 1. **Analysis of the Convergence of Monte Carlo Estimation for Shapley Values**

For any task subset $S$, the VolDet utility satisfies $$0 \le v(S) = \det(\mathcal{M}_S) \le 1.$$

Hence each marginal contribution in the permutation-based MC estimator,  $\Delta_i(S) = v(S \cup \{i\}) - v(S),$ is bounded in an interval of width $B \le 2$.  Together with uniform sampling over permutations, this yields an **unbiased** and **bounded** MC estimator of the Shapley values.
In Appendix G we derive the following Hoeffding-style concentration bound:
$$
\Pr\Bigl( \bigl\| \widehat{\boldsymbol{\phi}} - \boldsymbol{\phi} \bigr\|_\infty > \varepsilon
\Bigr)\le 2N \exp\Bigl(- \frac{2K \varepsilon^2}{B^2} \Bigr).
$$

This inequality shows that the deviation $\bigl\|\widehat{\boldsymbol{\phi}} - \boldsymbol{\phi}\bigr\|_\infty$
shrinks at rate $\mathcal{O}(1/\sqrt{K})$ as the number of MC samples $K$ increases. More detail is provided in Appendix G.



### 2. **Empirical results validate the effectiveness and convergence of MC Shapley Value**

We report an experiment to empirically validate the effectiveness and convergence of MC Shapley. We collect gradient Gram matrices from Office-31 ($N = 3$) and from CelebA ($N = 40$). For CelebA, we randomly sample task subsets of size  $N_{\text{sub}} \in \{10, 15, 20\}$  and extract the corresponding VolDet submatrices.

For each matrix, we compute both exact Shapley values and their MC approximation using our permutation-based estimator, for MC samples  $K \in \{100, 250, 500, 1000, 2000\}.$  We then measure the MSE between exact and MC Shapley values and average over 10 matrices.

**Table 1.** Results on Office-31 and CelebA

| Sample Size $K$ | Office-31 ($N=3$) | CelebA ($N=10$) | CelebA ($N=15$) | CelebA ($N=20$) |
|---|---|---|---|---|
| $$100$$  | $$9.18\times10^{-5}$$ | $$1.96\times10^{-5}$$ | $$5.95\times10^{-6}$$ | $$3.19\times10^{-6}$$ |
| $$250$$  | $$1.05\times10^{-4}$$ | $$2.13\times10^{-5}$$ | $$3.07\times10^{-6}$$ | $$1.48\times10^{-6}$$ |
| $$500$$  | $$5.84\times10^{-5}$$ | $$1.38\times10^{-5}$$ | $$2.49\times10^{-6}$$ | $$1.77\times10^{-6}$$ |
| $$1000$$ | $$0.671\times10^{-5}$$ | $$1.28\times10^{-5}$$ | $$2.26\times10^{-6}$$ | $$1.33\times10^{-6}$$ |
| $$2000$$ | $$5.36\times10^{-5}$$ | $$1.25\times10^{-5}$$ | $$1.87\times10^{-6}$$ | $$1.28\times10^{-6}$$ |

We make two observations from the Table 1:
A. For all large-task settings ($N_{\text{sub}}\ge 10$), the MSE rapidly decreases as $K$ grows and essentially plateaus once $K \ge 1000$. This confirms that the choice $K=1000$ in the CelebA experiments is well justified: increasing to $K=2000$ provides almost no benefit while nearly doubling the Shapley-related computation.

B. For the small-task case (Office-31, $N=3$), exact Shapley is cheap so MC is unnecessary. The slight non-monotonic rebound at $K=2000$ is typical Monte Carlo noise at the $10^{-5}$ level rather than a systematic issue, and it has no impact on our method
because we never rely on MC in such small-$N$ settings.

Overall, these results directly support the **effectiveness** and **convergence** of the MC Shapley estimator.

---

> ### Author Response · Authors · 2025-11-17
> **Response to questions about the effectiveness, convergence, and cost–performance trade-off of MC Shapley estimation (Part 2/2)**
>
> ### 3. **Empirical results validate that increasing MC samples $K$ improves final performance**
>
> We further evaluates how increasing $K$ affects final model performance. We run full training experiments on Office-31 and CelebA with different $K$ values. Each experiment is repeated 3 times with different seeds.
>
> **Table 2.** Results on Office-31 (3-task) and CelebA (40-task).
>
> | $K$ | (Office-31) Amazon↑ | (Office-31) DSLR↑ | (Office-31) Webcam↑ | $\Delta m\% \downarrow$ (Office-31) | $\Delta m\%\downarrow$ (CelebA) |
> |---|---|---|---|---|---|
> | 100  | 85.30 | 99.18 | 98.87 | −1.43 | 7.12 |
> | 250  | 84.84 | 99.45 | **99.07** | −1.42 | 3.83 |
> | 500  | 85.58 | 99.45 | 98.52 | −1.51 | 2.12 |
> | 1000 | 85.41 | **99.73** | 98.70 | −1.60 | 0.11 |
> | 2000 | 85.64 | **99.73** | 98.33 | −1.56 | **0.06** |
> | Precise | **86.04** | 99.45 | 98.33 | **−1.62** | — |
>
> We make observation from the Table.2:
> CelebA Increasing $K$ from $100$ to $1000$ dramatically improves fairness, with $\Delta m\%$ dropping from $7.12$ to $0.11$. Further increasing $K$ to $2000$ yields only a very small improvement ($0.11 \rightarrow 0.06$), while almost doubling the Shapley-related computation. This shows that $K=1000$ already lies on a cost-effective performance plateau, consistent with the MSE results in point~(2) (2. Empirical results validate the effectiveness and convergence of MC Shapley Value). The Office-31 result also exhibits a similar experimental phenomenon.
> Overall, these results directly support that MC samples $K$ can improve the final performance.
>
> In summary, our analysis and experiments jointly demonstrate that the permutation-based MC Shapley estimator is theoretically well-founded, empirically accurate, and practically efficient. We have updated the above theories and experimental results in Appendix G (Line 2122-2318).

---

### Author Response · Authors · 2025-12-02
**Summary of Reviewer Strengths & Author Rebuttals (1/2)**

# Summary of Reviewer Strengths & Author Rebuttals

## Part 1: Strengths Recognized by Reviewers
1. **Novel Integration**: SVFair innovatively integrates Shapley-value-based fairness reasoning with gradient conflict measurement via novel VolDet/VolDetPro metrics, unifying geometric and cooperative-game perspectives while capturing task gradient geometric misalignment beyond pairwise relational measures and providing intuitive geometric intuition for task conflict via the Gram determinant-based gradient volume concept.
2. **Theoretical Rigor**: The framework is theoretically grounded with rigorous convergence guarantees to Pareto stationary points and comprehensive theoretical analysis supporting its design rationale.
3. **Empirical Validity**: Extensive empirical studies across supervised and reinforcement learning benchmarks (NYU-v2, CelebA, MT10) demonstrate consistent performance and fairness improvements over state-of-the-art baselines, validating the effectiveness of SVFair.
4. **Scalability & Compatibility**: SVFair is highly scalable for large-task scenarios via Monte Carlo sampling, requires only first-order gradient information, scales linearly with the number of tasks, and can be seamlessly integrated as a fairness-enhancing module into existing MTL methods with simple implementation.
5. **Conceptual Coherence**: SVFair unifies MTL task balance with broader fairness research, offering strong conceptual coherence between multi-task optimization and fairness axioms, and the paper is well-structured with clear explanatory figures and algorithm boxes for high readability.

## Part 2: Responses to Reviewers’ Concerns
### 1. Reviewer 563x
- **Concern**: Exact Shapley computation is costly for large N; MC approximation’s effectiveness/convergence/cost-performance trade-off is unvalidated.
- **Response**: We theoretically proved MC Shapley estimation is unbiased (Hoeffding bound, deviation shrinks at O(1/√K)) and empirically validated balances cost and performance.
- **Outcome**: The concern was fully addressed.


### 2. Reviewer Mspw
> **Concern 1**: Typo in Shapley value interpretation.
- **Response**: We corrected the typo.
- **Outcome**: The typo issue was resolved.

> **Concern 2**: VolDet ignores gradient magnitude, potentially leading to suboptimal solutions.
- **Response**: We theoretically justified magnitude’s role via $ g_i^⊤d$  and empirically showed the original design outperforms an explicit magnitude-aware variant.
- **Outcome**: The reviewer acknowledged the validity of the original design, resolving the concern.

> **Concern 3**: Lack of ablation to prove Shapley value weights are necessary.
- **Response**: We constructed a simple Cosine-sum baseline and showed it underperforms SVFair, proving Shapley’s performance benefits outweigh its computational cost.
- **Outcome**: The reviewer recognized the necessity of Shapley weights and planned to raise the rating to 6.

### 3. Reviewer gTqY
> **Concern 1**: Overlap with FairGrad/PIVRG in optimization form weakens theoretical novelty.
- **Response**: Our core innovation lies in fairness modeling (VolDet/VolDetPro + Shapley) rather than optimization form, ensuring compatibility while converging to more balanced Pareto points.
- **Outcome**: The reviewer accepted the explanation.

> **Concern 2**: No theoretical/practical guidance for tuning temperature parameter τ.
- **Response**: We theoretically framed τ as an entropy regularizer and provided a practical guideline (large τ for sharp Shapley gaps/conflicts, small τ for balanced tasks).
- **Outcome**: The reviewer acknowledged the guidance.

> **Concern 3**: Abstract/introduction is overly concise (unclear problem background/motivation for Shapley fairness).
- **Response**: We rewrote the abstract and expanded the introduction to clarify task-level unfairness and motivate VolDet/Shapley as a principled solution.
- **Outcome**: The reviewer accepted the revision.

---

> ### Author Response · Authors · 2025-12-02
> **Summary of Reviewer Strengths & Author Rebuttals (2/2)**
>
> ### 4. Reviewer j3fR
> > **Concern 1**: Lack of originality
> - **Response (Round 1)**: We clarified SVFair’s originality: VolDet/VolDetPro model coalition-level gradient conflict (unlike pairwise baselines), Shapley provides the unique fair cost-sharing rule; we planned a revised subsection to compare geometric/game-theoretic MTL methods.
> - **Outcome**: The reviewer acknowledged the clarification but still viewed SVFair as incremental (partial resolution).
> - **Response (Round 2)**: We proposed a unified Gram-geometric master problem showing PCGrad/FairGrad/ParetoMTL/IMTL-G are special cases, with VolDet as the anchor implementation (global determinant + Shapley fairness) representing an integrative advance.
> - **Outcome**: We demonstrate from a unified perspective that one of the our core contribution and reviewer can't comment.
>
> > **Concern 2**: Weak theoretical grounding
> - **Response (Round 1)**: We corrected the "front-center" typo to Pareto-stationarity and showed the MC Shapley estimator is unbiased/convergent.
> - **Outcome**: The reviewer acknowledged convergence clarity but noted the lack of a fairness/efficiency link (partial resolution).
> - **Response (Round 2)**: We added a VolDet-based fairness functional 𝒫(u) and proved SVFair is its unique minimizer, while baselines like FairGrad are suboptimal under non-uniform Shapley weights.
> - **Outcome**: We add the link for fairness/efficiency guarantees and reviewer can't comment.
>
> > **Concern 3**: Thin empirical evaluation (small gains <2%, no statistical tests; Δm non-monotonic)
> - **Response (Round 1)**: We expanded experiments to PINNs and clarified small gains over strong baselines are practically meaningful; we planned statistical tests.
> - **Outcome**: The reviewer acknowledged expanded experiments but viewed gains as small/fragile (partial resolution).
> - **Response (Round 2)**: We introduced ΔΔm (incremental Δm improvement) showing SVFair’s gains match/exceed SOTA.
> - **Outcome**: We prove that our gains are't small with the ΔΔm and reviewer can't comment.
>
> > **Concern 4**: "Fairness" interpretation overstated (only task balance, not social fairness)
> - **Response (Round 1)**: We clarified "fairness" refers to Shapley’s cooperative-game axioms for task-level cost sharing, not social/demographic fairness.
> - **Outcome**: The reviewer accepted the clarification (full resolution).
>
> > **Concern 5**: Unclear benefit of VolDetPro (underexplained signed penalty; insufficient empirical evidence)
> - **Response (Round 1)**: We explained VolDetPro addresses VolDet’s sign agnosticism (penalizing antagonistic gradients) and showed it outperforms VolDet when PNCP > 0.25.
> - **Outcome**: The reviewer accepted the explanation (full resolution).
>
> > **Concern 6**: No comparison to prior Shapley-MTL methods (unclear complexity/approximation bias differences)
> - **Response (Round 1)**: We clarified prior Shapley methods focus on data/feature valuation, while SVFair uses single-pass task-level computation, with detailed complexity/bias analysis in appendices.
> - **Outcome**: The reviewer acknowledged the distinction (full resolution).
>
> > **Concern 7**: Poor scalability of Gram determinant for large tasks
> - **Response (Round 1)**: We validated SVFair on CelebA (40 tasks) via MC sampling and proposed low-rank/stochastic trace estimators for hundreds of tasks (future work).
> - **Outcome**: The reviewer acknowledged the scalability design (full resolution).
>
> > **Concern 8**: Sensitivity to task scaling/normalization (determinant magnitudes vary drastically)
> - **Response (Round 1)**: We proved 𝓁₂-normalization makes VolDet scale-invariant, and empirically showed unnormalized gradients bias weights; we clarified we always normalize gradients.
> - **Outcome**: The reviewer accepted the validation (full resolution).
>
> > **Concern 9**: Request for statistical significance/confidence intervals
> - **Response (Round 1)**: We reported 95% confidence intervals from 6 Office-31 trials.
> - **Outcome**: The reviewer accepted the validation (full resolution).
>
>
> > **Concern 10**: Request to integrate ParetoMTL's convex combination layer
> - **Response (Round 1)**: We explained SVFair is compatible with ParetoMTL—we can use VolDet-based Shapley weights to compute fair task weights, then feed reweighted gradients into ParetoMTL’s layer.
> - **Outcome**: The reviewer accepted the compatibility explanation (full resolution).

---

### Meta-Review · Area_Chair_VJFs · 2026-01-06

**Summary:**

This paper proposes SVFair, a Shapley-value-based framework for fair multi-task learning, introducing two geometric conflict metrics, VolDet and VolDetPro, to quantify coalition-level gradient interactions and guide task reweighting. The method is conceptually well motivated and technically well developed, with both theoretical analysis and extensive experiments across supervised and reinforcement learning benchmarks.

**Reviewer Concerns:**

Reviewers raised concerns primarily about computational overhead, theoretical grounding (convergence and fairness interpretation), and originality relative to prior gradient-based MTL methods. During rebuttal, the authors provided detailed theoretical analysis of the Monte Carlo Shapley estimator, added ablations demonstrating the necessity of key components, clarified the role of VolDet/VolDetPro, and introduced a formal fairness objective for which SVFair is the unique optimizer. These additions substantially addressed the concerns regarding scalability, convergence, and methodological justification.

**Reviewer Scores:**

Two reviewers were clearly positive throughout, and one initially borderline reviewer explicitly indicated an intention to raise their score after the rebuttal. One reviewer maintained reservations regarding originality, theoretical depth, and the magnitude of empirical gains. However, these remaining concerns primarily reflect a higher bar for conceptual novelty and strength of evidence, rather than identifying errors, incorrect assumptions, or fundamental flaws in the method. The technical soundness, reproducibility, and scope of evaluation are now strong.

---

### Decision · Program_Chairs · 2026-01-26

Accept (Poster)